# TAME THE BALROG:
# TASK-ADAPTIVE MODULAR EMERGENCE FRAMEWORK FOR GAME AGENTS

## ABSTRACT

Interactive games have proven to be key benchmarks for advancing Artificial Intelligence (AI), requiring capabilities like long-term planning, exploration, and adaptation to stochastic environments. While Large Language Models (LLMs) have achieved notable results across many domains, they struggle in complex gaming environments like those in the BALROG benchmark. The absence of adaptive frameworks that can dynamically configure themselves based on environmental characteristics, limits the progress of AI in games. To this end, we introduce the Task-Adaptive Modular Emergence (TAME) framework, which employs genetic algorithms to evolve environment-specific structures from modular components, enabling significant performance improvements of LLMs across diverse domains. TAME discovers high-performing configurations by selecting between baseline and hierarchical structures, selectively incorporating specialised modules, and fine-tuning each component through systematic mutations. Evaluating TAME across the BALROG benchmark, TAME discovers high-performing architectures that deliver substantial gains: Gemini-2.0-Flash improves from 27.16% to 35.05%, while GPT4.1-nano rises from 9.91% to 17.20%. Moreover, these structures demonstrate good transferability for larger models of the same family. Transfering these architectures to Gemini-2.5-Pro, we achieve new state-of-art performance on BALROG.

## 1 INTRODUCTION

Large language models (LLMs) have achieved remarkable growth across a wide range of tasks, from general language understanding (Hendrycks et al., 2020) and code generation (Wang et al., 2024a; Pan et al., 2025; Hong et al., 2024), to recent breakthroughs including mastering the ARC reasoning benchmark (Chollet, 2024; Chollet et al., 2024) and performing at gold-medal level on International Mathematical Olympiad (Chervonyi et al., 2025). However, these models struggle significantly in interactive decision-making environments that require sequential actions, state awareness, and long-term planning (Liu et al., 2024; Klissarov et al., 2025).

Interactive games have historically served as major testbeds for artificial intelligence, with examples including Atari (Mnih et al., 2013), Starcraft (Team, 2019), or GrantTurismo (León et al., 2024). Those successes predominantly emerged from reinforcement learning (RL) approaches specifically engineered for each domain, often requiring millions of training episodes and domain-specific reward shaping. While LLMs hold considerable promise on the possibility of zero-shot generalisation across games through their vast pretraining experience, e.g., game wikis, strategy guides, and gameplay discussions, they fail to translate this latent knowledge effectively. This performance gap is clearly illustrated in the BALROG benchmark (Paglieri et al., 2025), a suite of diverse games traditionally employed in RL research, where even state-of-the-art LLMs achieve only partial success in the simpler games and barely progress with more challenging ones.

Notably, agentic frameworks have emerged as a dominant approach to enhancing LLM capabilities in other complex domains, including software development (Yang et al., 2024; Hong et al., 2024; Pan et al., 2025) or scientific research (Bran et al., 2023; Wang et al., 2024a; Lu et al., 2024). There are also recurrent efforts to improve long-term memory management in agentic frameworks,

with solutions that prioritise either speed and cost efficiency, such as Jarvis-1 (Wang et al., 2024b), or performance, such as A-mem (Xu et al., 2025). Yet, despite presenting similar challenges, no agentic framework has been applied to games, which requires to handle challenges such as partially observability or exploration that are not as critical in most math or coding applications.

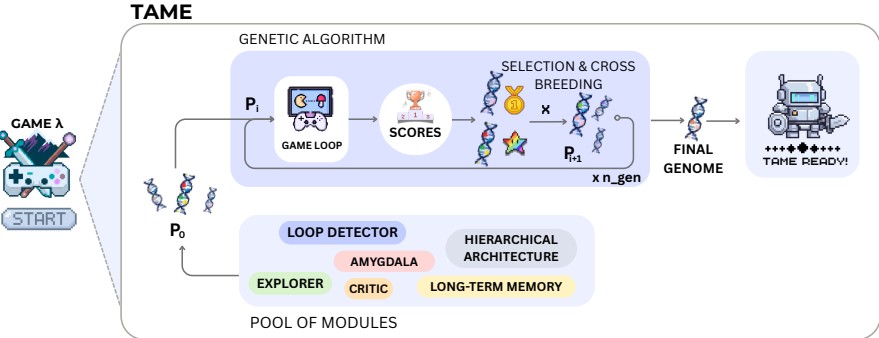

Figure 1: TAME evolutionary framework overview. TAME runs a genetic algorithm to generate a population of genomes representing different module combinations, hyperparameters, and prompts, tune them through a series of genetic operation, and selects the genome achieving the highest score. Icons were generated with Google Gemini Pro 2.5, 2025.

To address this gap, we introduce the TAME (Task-Adaptive Modular Emergence) framework, a genetic framework for LLMs to evolve agent architectures for diverse games. TAME consists of a series of handcrafted game skill modules that enable different capabilities that might be relevant in games, e.g., hierarchical planning, memory. However, unlike in other domains, games can encode very diverse dynamics, e.g., Nethack is a game where exploration, long-term planning and memory are of paramount importance, while none of those skills help in a game like TextWorld. Thus, TAME undergoes an evolutionary process, iteratively exploring the best configuration for a specific game. Figure 1 illustrates this process. Each candidate structure is encoded through a vector that represents which modules are activated, along with the hyperparameters and the prompt discovered for this candidate. Through successive generations, TAME employs mutation and cross-over operations on the genomes of the selected candidates to discover increasingly effective structures, hyperparameters, prompts and inputs, balancing performance and diversity in their selection.

We validate TAME through extensive experiments on the BALROG benchmark. We find that TAME-discovered configurations achieve relative score improvement of 29% and 74% compared to the baseline LLM performance, Gemini Flash 2.0 and GPT-4.1-nano, respectively. Moreover, we show that architectures discovered by TAME exhibit strong transferability across the same family of models: genomes evolved using Gemini-2.0-Flash directly enhance the performance of Gemini-2.5-Flash-Lite and Gemini 2.5-Pro without additional adaptation, and same pattern repeats on the GPT models. In the case of Gemini-2.5-Pro, we achieve new state-of-the-art results on BALROG.

Thus, our contributions can be summarised as follows: (1) We introduce TAME, the first emergent game-agentic framework that enables LLMs to evolve modular structures tailored to gaming environments and achieve state-of-the-art performance in the BALROG benchmark; (2) We present a genetic approach key for the functioning of this framework, introducing a set of modules for general gaming capabilities and a genome that captures modules, hyperparameters and prompts enabling TAME to adapt the agent to the game with a low evolutionary budget (4 generations and 5 children); (3) We demonstrate that TAME can be used to find effective agentic configurations with smaller models that directly transfer and improve the performance of larger and reasoning models of the same family; (4) We propose a novel and effective long-term memory system that combines embedding-based retrieval with LLM-augmented semantic memory matching the performance of state-of-the-art memory methods requiring three times less LLM calls.

## 2 RELATED WORK

**Prompting and Memory.** Hallucinations remain a key challenge for LLMs (Kalai et al., 2025), which can be mitigated through prompting techniques like chain-of-thought (Wei et al., 2022) and

step-by-step reasoning. Limited context windows (Brown et al., 2020) is another limitation, driving development of memory systems. Retrieval-Augmented Generation (RAG) (Lewis et al., 2021) combines LLMs with external document retrieval to reduce hallucinations without retraining. HiAgent (Hu et al., 2024a) manages hierarchical memory using subgoals, dividing into "working memory" and "cross-trial memory" with LLM-based observation summarisation, while Park et al. (2023) balance memory retrieval using recency, importance, and relevance scores. Jarvis1 (Wang et al., 2024b) stores task names, plans, and observation sequences using embedding (CLIP) for encoding and retrieval. A-mem (Xu et al., 2025) introduces structured memory notes with timestamps, keywords, and embeddings, establishing inter-memory connections through LLM calls.

**LLMs as hierarchical planners.** In TWOSOME (Tan et al., 2024), LLMs score actions based on observations, allowing RL agents to leverage world knowledge for improved decisions. MaestroMotif (Klissarov et al., 2024) uses LLMs to generate reward functions for skills, while LLM-Augmented Hierarchical Agents (Prakash et al., 2023) use LLMs to inject commonsense priors for more efficient policy learning. Jarvis1 (Wang et al., 2024b), consists of planner and controller, enhanced by multimodal memory system. An important limitation of Jarvis-1 is the necessity of human-crafted goals based on specific skills, limiting its application in games with emerging tasks.

**Agentic frameworks.** Agentic frameworks are systems that enable Large Language Models to act as autonomous agents capable of reasoning, planning, and interacting with external tools and environments. Recent work, such as AGENTBREEDER Rosser & Foerster (2025), shows that optimising frameworks provides superior multi-agent performance on reasoning, mathematics, and safety benchmarks. Moreover, multiple works show improvements in scientific discovery (Lu et al., 2024) and software development (Yang et al., 2024) through these structured frameworks. Related research like AFLOW (Zhang et al., 2024) and ADAS (Hu et al., 2024b) focus on optimising agent workflows, i.e. the sequential flow and coordination of processing steps. Other frameworks (Yuan et al., 2024; Zhang et al., 2025) use evolutionary search to construct multi-agent systems, where multiple distinct agents are coordinated with specialised roles. Different from these works, we present a single-agent system operating on a fixed workflow architecture and evolves the internal implementations. Closer to our work, AgentSquare (Shang et al., 2024) proposes a single-agent framework but with several key shortcomings over our approach when applied to games. Unlike AgentSquare, which relies on manual tool definitions and a restricted module set, TAME does not require additional user input to adapt to different games. TAME also expands the search space to include critical gaming skills—such as exploration, loop detection, and survival (see Figure 15) and introduces a novel, more effective long-term memory system.

**Evolutionary Strategies.** While earlier work in Evolutionary Hyperparameter Optimization (EHO) focused on optimising over numerical hyperparameters Vincent & Jidesh (2023) and neural network topologies Lu et al. (2019); Stanley et al. (2019), recent literature has also explored combining LLMs and evolutionary frameworks. Genetic algorithms are a common choice due to easy parallelisation across hardware (Ma et al., 2024; Rosser & Foerster, 2025; Sarkar et al., 2025).(Lehman et al., 2022) propose "evolution through large models" using LLMs as evolutionary operators. EvoPrompt (Guo et al., 2025) employs LLMs for crossover and mutation in genetic algorithms to discover diverse prompts, while Rainbow Teaming (Samvelyan et al., 2024) mutates adversarial prompts to populate MAP-Elites archives systematically. DOMiNO (Zahavy et al., 2022) balances quality-diversity trade-offs using Lagrange multipliers. Eureka (Ma et al., 2024) shows evolutionary optimisation over reward code benefits from human initialisation.

**Options.** We propose a hierarchical structure for decision-making, a method common in robotics (Wohlke et al., 2021), autonomous driving (Duan et al., 2020), and games (Lin et al., 2021). Following the Options Framework (Sutton et al., 1999), we use the term options to denote manageable subtasks that decompose the main objective.

## 3 TAME FRAMEWORK

We introduce TAME (Task-Adaptive Modular Emergence framework), a novel agentic framework designed for dynamic LLM adaptation across diverse gaming environments. Inspired by Eureka (Ma et al., 2024), which shows that human priors significantly improve LLM-based evolutionary optimisation performance, TAME begins with an initial population $\mathcal{P}_0$ comprising diverse modular structure configurations, each encoding different combinations of human-crafted modules and hy-

perparameters. The framework's modular architecture consists of six core components illustrated in Figure 2: hierarchical goal decomposition (comprising a Meta-Controller, Low-Level Executor, and Completion Validator), Long-Term Memory, Critic, Loop Detector, Amygdala, and Explorer. TAME's evolutionary process operates iteratively: in each generation, the framework evaluates all new members $p \in \mathcal{P}_i$ on the target game and selects candidates for the next generation based on two criteria: (1) the top $N$ performers by absolute score, and (2) $M$ additional diverse solutions that achieve at least a fraction $\alpha$ of the best performer's score. This dual selection strategy balances exploitation of successful structures with exploration of the relevant solution space. Each candidate's genome is represented as a vector encoding active modules, hyperparameters (e.g., memory decay rates, exploration-exploitation trade-offs), and module-specific prompts. After every iteration of TAME, the genomes of the selected candidates undergo mutation and crossover operations to generate the new members of the subsequent population $\mathcal{P}_{i+1}$. Through successive generations, TAME discovers increasingly effective structures tailored to each game's requirements. The remaining of this section details the genetic algorithm and the design and functionality of each modular component in TAME.

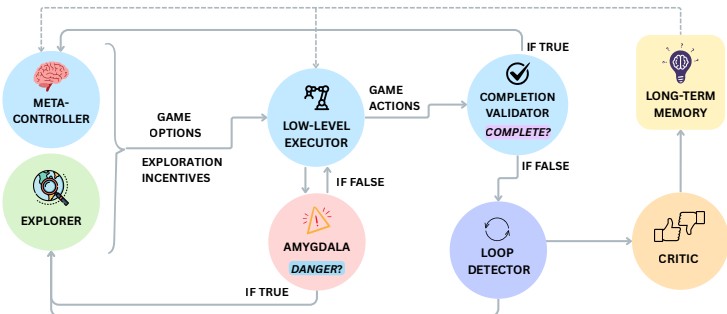

Figure 2: Fully enabled TAME modular architecture: an epsilon-greedy mechanism selects between Meta-controller (providing options toward game objectives) and Explorer (options towards exploring the environment). Low-level Executor proposes actions, while Amygdala checks for danger and prioritises survival. Completion Validator checks option completion while the Loop Detector identifies stuck states, and the Critic summarises key actions that led to the outcome of the option. Long-term memory saves successful and failed trajectories and adds them to LLM context.

### 3.1 NOTATION AND FUNCTIONAL INPUTS

Let $\mathcal{G} = \{g_1, \ldots, g_n\}$ the set of decomposed options towards the game objective, $A$ the action space, $O$ the observation space, $S$ the state space, $H$ the set of option summaries, and $\mathcal{T}$, space of natural text. Each state $s_t \in S$ at time $t$ is defined as:

$$s_t = (o_t, I, h_{i-1}, g_i, \tau_i, M_i^{\pm}, F_{\text{recent}}, F_i, F_{i,act}, F_{i,obs})$$

where $o_t \in O$ is the current agent's observation and $I$ provides game context (objective and available actions), $h_{i-1} \in H$ summarises the previous option's outcome, $g_i \in \mathcal{G}$ is the active option with termination condition $\tau_i$, $M_i^{\pm}$ contains the top successful $k_{succ}$ and failed $k_{fail}$ options. $F_{\text{recent}}$ stores the last $m$ action-observation pairs, while $F_i$, $F_{i,actions}$, and $F_{i,obs}$ maintain action-observation, action-only, and observation-only trajectories for the current option respectively.

Each component within $s_t$ can serve as a functional input to any module within the framework. Through early experimentation we noted that the selection of appropriate functional inputs has a significant impact on performance, and it is up to the evolutionary process to find the most appropriate inputs for each module. For the initial population generation, we hand-crafted inputs that we consider most relevant for each module, providing the genetic algorithm with informative human priors. This is further detailed in Section 3.2.

### 3.2 GAME ADAPTATION

Given the diverse dynamics and requirements across different gaming environments, we propose a genetic optimisation approach that automatically explores diverse agentic architectures to adapt

the underlying LLM to the specific game characteristics. To that end, we first encode the agentic structure descriptors into genomes. Each genome is constructed from three core components:

- **Modules**: TAME modules (See Section 3.3) are parametrised as a set of binary values. Such values indicate if a module is active (1) or not (0). The set of modules include: *Hierarchy* (enables or disables hierarchical goal decomposition, includes Meta-controller, Low-level executor and Completion Validator), *Long-Term Memory* (stores past experiences), *Critic* (summarises key actions toward the current option), *Amygdala* (activates survival mode), *Loop Detector* (detects looping behaviour), *Explorer* (controls exploration strategies).

- **Hyperparameters**: A set of continuous values encoding: *Long-Term Memory Time Decay Factor* $\lambda$, which sets the option priority decay rate; *Long-Term Memory Similarity Threshold* $\tau_{cos}$, which specifies the cosine similarity cutoff for storing memories; *Exploration Parameter* $\epsilon$, which controls the epsilon-greedy exploration; and *Language Model Temperature*.

- **Prompts**: The last component of the genome includes the prompts used by the modules of the agentic structure.

For any game $G$ the initial population $\mathcal{P}_0$ consists of four predefined genomes:

$$\mathcal{P}_0 = \left\{ \mathbf{g}_{\text{Baseline}}, \mathbf{g}_{\text{Hierarchical[Hand-Crafted]}}, \mathbf{g}_{\text{Full[Claude]}}, \mathbf{g}_{\text{Full[Hand-Crafted]}} \right\}$$

Where $\mathbf{g}_{\text{Baseline}}$ is genome corresponding to a baseline structure (no modules activated) with a single prompt as in Paglieri et al. (2025) (BALROG). $\mathbf{g}_{\text{Hierarchical[Hand-Crafted]}}$ corresponds to TAME[Hierarchical+Long-Term Memory], a genome with the hierarchical and long-term memory modules activated. $\mathbf{g}_{\text{Full[Claude]}}$ - refers to a genome with all the modules activated that employs prompts proposed by Claude-Sonnet-4. Finally, $\mathbf{g}_{\text{Full[Hand-Crafted]}}$ is also a genome with all the modules active, but with our own engineered prompts. See Appendix Q for full detail.

**Genetic Operations.** Given the distinct nature of the components within the genome (i.e., binary or continuous values, or prompts) TAME employs distinct crossover and mutation strategies depending on the type of variable or representation that is handled. For binary variables, we use a parent-based probabilistic flipping mechanism that incorporates an inheritance bias. Continuous variables are handled through Gaussian perturbation for mutation and linear interpolation for crossover. Finally, for prompt optimisation, we adopt the EvoPrompt methodology by Guo et al. (2025), which enables crossover and mutation tailored to LLM-based prompts (availabe in Appendix P).

**Genome Evaluation.** Each genome is evaluated using the average game progression across $n_{ep}$ episodes as a fitness function. Moreover, we embed the genome representation and calculate the minimum distance to genomes already existing in the population as a score of diversity, allowing for keeping population of best scoring and most diverse genomes.

**Genetic Algorithm.** Our genetic algorithm iterates through four steps: 1) *Parent Population Selection*: TAME chooses parents based on roulette wheel selection (a probabilistic parent-picking method where each individual's chance of being chosen is proportional to its fitness score) 2) *Reproduction*: Parents for reproduction are chosen from parent population with $p_{single}$ and $1 - p_{single}$ referring to the probability of single-parent or two-parent, respectively. Genetic operations are applied, represented by mutation+crossover or mutation alone (based on the number of parents) 3) *Fitness Evaluation:* calculates genome's performance score and diversity based on average game progression and embeded genome representation, respectively. 4) *Population Pruning*: the population is trimmed to maintain a maximum of $N + M$ individuals — $N$ highest-performing plus $M$ most diverse genomes (subject to achieving a factor $\alpha$ of the performance of the best genome within the population). Most diverse genomes are those with largest minimum distance of their embedding with respect to the already existing embeddings in the population. For comprehensive information on the genetic algorithm, refer to the appendices: Appendix H provides further detail and discussion, while Appendix H.3 and Appendix D contain the pseudocode and hyperparameters, respectively.

## 3.3 MODULAR BLUEPRINTS

As anticipated through Section 3, inspired by how Eureka improved its ability to find better reward functions by starting from a human-crafted set of prior, TAME incorporates a set of human-crafted

modules that target essential capabilities for agents in interactive games. The remaining of this section details such components. We remind the reader than $s_t^{mc}, s_t^{lc}, s_t^{cv}, s_t^c, s_t^e, s_t^a, s_t^{ld}$ used in sections below, represent subsets of $s_t$ selected by the genetic algorithm for corresponding modules.

### 3.3.1 HIERARCHICAL PLANNING

The hierarchical module consists of three main components illustrated in Figure 2: Meta-Controller suggests sequence of options, Low-Level Executor performs a sequence of actions towards each option, and Completion Validator judges if an option has been completed successfully or failed.

**Meta-Controller.** The Meta-Controller decomposes the game objective into a more manageable sequence of options. Specifically, it implements $\pi_{high} : S \to \mathcal{G}$, mapping the current state to an ordered sequence of options:

$$g = \pi_{high}(s_t^{mc}) = \text{LLMprompt}_{high}(s_t^{mc}) = (g_1, g_2, \dots) \tag{1}$$

Each option $g_i$ consists of the fields: *name*, *description*, *prerequisites*, *success conditions*, *penalty component*, *progress indicators*, *estimated priority*.

**Low-level Executor.** This system implements $\pi_{low} : S \to A$, producing an action sequence based on the current state information provided $s_t^{le}$:

$$a = \pi_{low}(s_t^{le}) = \text{LLMprompt}_{low}(s_t^{le}) = (a_1, a_2, \dots) \tag{2}$$

where the length of the sequence is decided by the Low-level Executor.

**Completion Validator.** The Completion Validator implements the binary classifier $\varphi : S \to \{0, 1\}$, determining whether an option has been completed:

$$C_i = \varphi(s_t^{cv}) = \text{LLMprompt}_{\varphi}(s_t^{cv}) \in \{0, 1\}. \tag{3}$$

Here $C_i = 1$ indicates successful termination. For details on the hand-crafted LLM prompts used as initial seeds we refer the reader to Appendix Q.

### 3.4 LONG-TERM MEMORY

TAME implements a novel memory system that seeks to leverage the cost and speed efficiency of embedding-based systems like Jarvis-1 (Wang et al., 2023) while achieving a performance closer to more complex systems like A-mem (Xu et al., 2025). To that end, our system adopts Jarvis-1's storage framework, maintaining option information including name, description, prerequisites, success conditions, progress indicators, penalty components, and observation sequences. We extend this with two key additions: (1) Critic llm-generated summaries highlighting key success/failure actions, and (2) success/failure classification labels obtained from Completion Validator. This enhancement provides actionable guidance for future tasks requiring a single LLM call while avoiding A-mem's computational overhead of three LLM calls for memory and link creations, and evolutions. We now provide further detail of how TAME's long-term memory works:

**Critic.** The Critic module is a function $\rho : S \times \{0, 1\} \to \mathcal{T}$, mapping the state and recent option outcome from Completion Validator to text:

$$h_i = \rho(s_t^c, C_i) = \text{LLMprompt}_{\rho}(s_t^c) \tag{4}$$

where the text aims to summarise the key factor that led to the success or failure of the option.

**Creation of the memory.** Each memory entry is defined as:

$$M_i = \{g_i, C_i, o, h_i\} \tag{5}$$

where $g_i$ is the option, $C_i \in \{0, 1\}$ is the output of the Completion Validator, $o$ is the observation sequence towards current option, and $h_i$ is the option summary from the Critic. Then, the memory structure is implemented as follows: each memory entry is stored as a vector embedding of the above, enabling efficient similarity-based retrieval. The embedding function $\phi$ transforms each entry:

$$\mathbf{e}_i = \phi(M_i) \in \mathbb{R}^d \tag{6}$$

**Similarity-Based Filtering.** Following the "importance" scoring approach from the generative agents framework (Park et al., 2023), we prevent storage of repetitive experiences. A new memory $M_{new}$ is stored only if:

$$\max_{M_i \in \mathcal{M}} \frac{\mathbf{e}_{new} \cdot \mathbf{e}_i}{||\mathbf{e}_{new}|| \cdot ||\mathbf{e}_i||} \leq 1 - \tau_{cos} \tag{7}$$

where $\tau_{cos}$ is a set constant. This method effectively filters out frequently repeated actions (e.g., "chop wood" in Crafter) that provide limited learning value.

**Long-Term Memory Retrieval Mechanism.** In order to address the limited context window size, we only extract $k_{succ} + k_{fail}$ best scoring memories at each option execution. Inspired by Retrieval-Augmented Generation (RAG) (Lewis et al., 2021), we enable access to past experiences through the following steps:

- *Query Encoding*: The new option name ($n_{g_i}$) and description ($d_{g_i}$) are embedded into the same vector space as stored memories using sentence embeddings:

$$\mathbf{q}_i = \phi(n_{g_i}, d_{g_i}) \in \mathbb{R}^d \tag{8}$$

- *Temporal Decay*: Following A-mem (Xu et al., 2025), we prioritise recent experiences using exponential decay:

$$w(e_t) = \exp(-\lambda \cdot t) \tag{9}$$

where $t$ is the time elapsed since memory creation.

- *Memory score*: We combine similarity and recency through weighted sum:

$$\text{score}(\mathbf{q_i}, \mathbf{e_j}) = w_{similarity} \cdot \text{sim}(\mathbf{q_i}, \mathbf{e_j}) + w_{recency} \cdot w(\mathbf{e_j}) \tag{10}$$

- *Stratified Retrieval:* The system retrieves top-k successful and failed memories:

$$M_i^+ = \text{Top-}k_{succ}(\text{score}(\mathbf{q_i}, \mathbf{e}_j) : \varphi_i = 1) \tag{11}$$

$$M_i^- = \text{Top-}k_{fail}(\text{score}(\mathbf{q_i}, \mathbf{e}_j) : \varphi_i = 0) \tag{12}$$

where $\varphi_j$ indicates success/failure of memory $j$.

- *Context Integration*: All $k_{succ} + k_{fail}$ retrieved memories are integrated into the modules prompts (modules including memory are decided by genetic algorithm).

This process gives access to both effective strategies and failure patterns, allowing for informed decision-making. Visualisations of retrieval patterns are shown in Appendix F.

## 3.5 SKILL-SPECIFIC MODULES

On top of the hierarchical structure, we identify *survival* and *exploration* as two key components in many video games. Moreover, we identify looping behaviour as a significant LLM limitation. All three modules are illustrated in Figure 2.

**Explorer.** Let explorer be defined as a function $\pi_{explorer} : S \to \mathcal{G}_{\text{explore}}$, where $\mathcal{G}_{\text{explore}}$ is the set of exploration-oriented options, and:

$$g^{exp} = \pi_{\text{explorer}}(s_t^e) = \text{LLM}_{\text{explorer}}(s_t^e) \tag{13}$$

where $g^{exp} = \{g_1^{\text{exp}}, g_2^{\text{exp}}, \ldots, g_k^{\text{exp}}\}$ is the sequential exploration plan, and each $g_i^{\text{exp}}$ is structured identically to regular options but prompted for discovery rather than game goal completion.

- **Exploration Strategy** We implement an $\epsilon$-greedy exploration strategy where the Meta-controller selection becomes:

$$\text{Controller}(s_t) = \begin{cases} \pi_{\text{explorer}}(s_t^e) & \text{with probability } \epsilon_t \\ \pi_{\text{high}}(s_t^{mc}) & \text{with probability } 1 - \epsilon_t \end{cases} \tag{14}$$

with $\epsilon_0 = 0.1, \epsilon_t = 0.99 \times \epsilon_{t-1} = 0.99^t \times \epsilon_0$

**Amygdala.** Let $\sigma : S \to \{0, 1\}$ be the amygdala function mapping observations to binary classification of danger assessment:

$$D_i = \sigma(s_t^a) = \text{LLM}_{\text{amygdala}}(s_t^a) \tag{15}$$

At each Low-Lever Executor step, if $D_i = 1$, the system immediately activates a "survival option" (see Appendix G.1 for details); otherwise, normal execution continues.

**Loop Detector.** The loop detector implements $\psi : S \to \{0, 1\}$, detecting repetitive behavior in recent execution history:

$$L_i = \psi(s_i^{ld}) = \text{LLMprompt}_{\psi}(s_i^{ld}) \in \{0, 1\}. \tag{16}$$

Where $L_i = 1$ means looping behaviour is detected.

## 4 EMPIRICAL EVALUATION

We evaluate our method through three key experiments. First, we benchmark our genetic algorithm with Gemini-2.0-Flash and GPT4.1-nano against the SOTA systems on the BALROG benchmark. Second, we demonstrate the transferability of TAME's selected genomes across different Gemini and GPT models without additional training. Third, we compare our memory system against Jarvis-1 and A-mem baselines. The detailed experiments on Gemini family can be found in the Appendix K, and detailed experiments on GPT can be found in the Appendix L .

### 4.1 TAME RESULTS

This section compares the baseline and TAME's performance on the BALROG benchmark. Baseline scores are obtained by evaluating the BALROG repository with Gemini-2.0-Flash and GPT4.1-nano following the original author's methodology. TAME scores represent the best performance achieved selected by our genetic algorithm (Section 3.2). We run genetic algorithm through $n_{gen} = 4$ iterations, with each iteration producing $n_{child} = 5$ children. We then repeat the genetic algorithm 3 times and average the results (see detailed results from independent runs in the Appendix K.2 and Appendix L.1). Through empirical evaluation we notice that gives sufficient performance gains. Number of episodes per each child evaluation is adapted from BALROG.

| Environment | Gemini-2.0-Flash | | GPT4.1-nano | | Episodes |
| --- | --- | --- | --- | --- | --- |
| | Baseline ($\uparrow$) | TAME ($\uparrow$) | Baseline ($\uparrow$) | TAME ($\uparrow$) | |
| **Average** | $27.16\% \pm 2.24\%$ | $\mathbf{35.05\% \pm 2.18\%}$ | $9.90\% \pm 1.33\%$ | $\mathbf{17.20\% \pm 1.47\%}$ | - |
| babayi | $58.00\% \pm 6.98\%$ | $\mathbf{72.00\% \pm 6.65\%}$ | $32.00\% \pm 6.60\%$ | $\mathbf{48.67\% \pm 7.07\%}$ | 50 |
| babaisai | $30.83\% \pm 4.22\%$ | $\mathbf{42.50\% \pm 6.51\%}$ | $12.50\% \pm 3.02\%$ | $\mathbf{21.55\% \pm 3.76\%}$ | 120 |
| textworld | $32.55\% \pm 6.95\%$ | $\mathbf{33.40\% \pm 7.23\%}$ | $0.59\% \pm 0.58\%$ | $\mathbf{2.88\% \pm 0.94\%}$ | 30 |
| crafter | $29.09\% \pm 4.51\%$ | $\mathbf{38.18\% \pm 4.25\%}$ | $11.82\% \pm 2.15\%$ | $\mathbf{19.78\% \pm 2.33\%}$ | 10 |
| minihack | $12.50\% \pm 5.23\%$ | $\mathbf{23.33\% \pm 6.69\%}$ | $2.50\% \pm 2.47\%$ | $\mathbf{10.00\% \pm 2.78\%}$ | 40 |
| nle | $0.00\% \pm 0.00\%$ | $\mathbf{0.91\% \pm 0.44\%}$ | $0.00\% \pm 0.00\%$ | $\mathbf{0.22\% \pm 0.20\%}$ | 5 |

Table 1: Baseline vs. TAME progression across three runs of genetic algorithm using Gemini-2.0-Flash and GPT-4.1-nano. For full details across independent runs see Appendix K.2 and L.1. *Note: Values are absolute scores, not relative improvements.*

As shown in Table1, TAME consistently outperforms the baseline achieving relative gain of $\sim 29\%$ in the case of Gemini-2.0-Flash and $\sim 74\%$ in the case of GPT4.1-nano. Moreover, TAME improves performance in all the games for both models. Notably, while the baseline models cannot achieve any noticeable progress on Nethack (the hardest game) TAME achieves 0.91% and 0.22% as an average scores (note that the best model on BALROG benchmark scores 1.8% on Nethack). Per-task details over one run are provided in Appendix J along with an analysis of module activations in Appendix I. The examples of final genomes returned by one of the genetic algorithm runs are available in Appendix R. Further results of the performance of initial population $\mathcal{P}_0$ in Appendix K.3 and L.2.

## 4.2 TRANSFERABILITY OF TAME STRUCTURES

Next, we evaluate whether architectures evolved with Gemini-2.0-Flash and GPT4.1-nano can be effective when transferred to other models. Thus, we use TAME selection to evaluate populations of Gemini-2.5-Flash-Lite and Gemini-2.5-Pro models to exclusively choose between the base configuration from BALROG or the best-performing structure discovered with Gemini-2.0-Flash for each game (see details in Appendix K.6). Similarly, we check for transferability between GPT4.1-nano and GPT4.1-mini (see details in Appendix L.4).

| Method | Score (↑) | BALROG Rank (↓) |
|---|---|---|
| Gemini-2.5-Pro[Transferred] | **47.57% ± 2.72%** | (1) ↑ 1 |
| Grok-4 | 43.60% ± 2.20% | 1 |
| Gemini-2.5-Pro[Baseline] | 43.35% ± 2.3% | 2 |
| Gemini-2.0-Flash[TAME] | 35.05 % ± 2.24% | (3) ↑ 9 |
| Gemini-2.0-Flash[Baseline] | 27.16% ± 2.12% | (12) |
| GPT-4.1-mini[Transferred] | 26.80% ± 1.92% | (12) |
| GPT-4.1-mini[Baseline] | 24.43% ± 1.89% | (12) |
| Gemini-2.5-Flash-Lite[Transferred] | 20.48% ± 0.91% | (14) ↑ 9 |
| GPT-4.1-nano[TAME] | 17.20% ± 1.47% | (18) ↑ 7 |
| Gemini-2.5-Flash-Lite[Baseline] | 11.87% ± 1.32% | (23) |
| GPT-4.1-nano[Baseline] | 9.91% ± 1.33% | (25) |

Table 2: Comparison of TAME against top scoring models in BALROG leaderboard (September 2025). We show how they would rank (in parenthesis) relative to the current leaderboard. Rank improvements are indicated with ↑. *Note: Values are absolute scores, not relative improvements.*

From Table 2 presenting the results, we observe that the Transferred Gemini-2.5-Flash-Lite achieves ∼73% relative improvement. Detailed analysis in Table 10 in Appendix K.6 demonstrates that TAME's discovered structures successfully transfers in five out of six environments, with only TextWorld achieving baseline performance. For Gemini-2.5-Pro, we also observe gains although more moderate. Table 11 in Appendix K.6 shows that the transferred structures significant improvements in the BabyAI and BabaIsAI environments, which require extensive planning, highlighting the framework's strengths in this domain. We also observe improvements in Textworld and MiniHack, however, improvements are not shown in Crafter and NetHack. We hypothesise that Gemini-2.5-Flash-Lite benefits more substantially because it is a non-reasoning model similar to Gemini-2.0-Flash, where we carried the optimisation, whereas Gemini-2.5-Pro is a reasoning-based models. Notably, transferring TAME's discovered genomes to Gemini-2.5-Pro we achieve state-of-art performance above the best model on the BALROG leaderboard - Grok-4. Similarly, we see large improvements on the leaderboard for Gemini-2.5-Flash-Lite and Gemini-2.0-Flash with TAME, now occupying rank 14 and 3 from 23 and 12 respectively. Our results extend beyond the Gemini model family, demonstrating that the TAME framework generalises across architectures. We obtain an ∼74% relative improvement when comparing TAME against the GPT-4.1-nano baseline (for more details see Appendix L.1). Moreover, genomes transferred from GPT4.1-nano to GPT4.1-mini improve the relative score by ∼10% (for more details see Appendix L.4), further strengthening our transferability claims.

## 4.3 ABLATION: MEMORY TYPES

We also include ablations to demonstrate the effectiveness of our long-term memory system. In order to test memory, we use the hierarchical structure described in Section 3.3.1, combined with three different memory architectures: Jarvis, TAME-Memory[ours] and A-mem. Both **Jarvis** memory and **A-mem** store the same core elements: $g_i$ (the option, including all information associated with it), $C_i \in \{0, 1\}$ (the status indicator), and $o$ (the observation sequence corresponding to the current option). TAME extends Jarvis framework by introducing a critic, as well as a filter for successful and failed trajectories (but does not create links between memories). This requires one additional LLM call compared to Jarvis, but two fewer LLM calls per generation compared to A-mem. Thus, our approach explores a trade-off between the simplicity of Jarvis and the more complex and expensive structure of A-mem.

We notice an improvement compared to Jarvis and A-mem as shown on Table 3, motivating the integration of critic module for memory storage. Moreover, we achieve this while requiring a third of the LLM calls that A-mem employs. Thus allowing our system to iterate faster and with a reduced compute cost. Further details are included in Appendix N.

| Environment | Jarivs (↑) | TAME-Memory[ours] (↑) | A-mem (↑) |
|---|---|---|---|
| Average | 17.52% ± 1.73% | **23.11% ± 1.75%** | 21.45% ± 1.80% |

Table 3: Comparison of average game progression across 6 games using different memory types.

## 5 DISCUSSION AND CONCLUSION

We presented TAME, a genetic framework for evolving LLM-based agents that is both game-agnostic and adaptive. Through genetic mutations and in-game evaluation, TAME configures human-crafted modules for core gaming skills such as exploration, survival, long-term memory, and loop detection. TAME explores diverse modular configurations, inputs, prompts, and hyper-parameters. We also introduced a novel memory system combining the efficiency of embedding retrieval with the contextual depth of LLM-augmented memory, matching the performance of state-of-art LLM-augmented memory systems in games while reducing the number of LLM calls required to achieve that performance.

We evaluated TAME on the well-established BALROG benchmark and find that it consistently enhances the underlying LLMs. Gemini-2.0-Flash improves from 27.16% to 35.05%, and GPT-4.1-nano from 9.91% to 17.20%,with a limited evolutionary budget of 4 generations and 5 children. Moreover solutions discovered on smaller models transfer training-free to larger models of the same family improving their performance. Transferring genomes to Gemini-2.5-Pro we reach 47.57% overall performance in BALROG and outperform the state-of-the-art. These results showcase both the generalisability of the core modules and the effectiveness of our genetic approach. We further confirm the importance of long-term memory and adaptive architecture, with our proposed memory system outperforming two existing baselines while remaining more cost-efficient than complex agentic systems.

We note some limitations. We find that TAME provides greater benefits to some games than others, where it defaults to the baseline architecture. We also observed that while TAME improves complex reasoning tasks overall, spatial reasoning remains a weakness. This suggests the potential not only for expanding the set but for genetic discovery of entirely new modules and capabilities, beyond those hand-crafted in this work. Moreover, while transferability proved effective, gains were less pronounced for reasoning models, motivating further study of transfer and emergence across different architectures. Finally, future work could explore alternatives for the genetic algorithms, like MAP-Elites, or different approaches such as Bayesian optimisation.

Overall, TAME establishes a new state of the art in game-playing LLM agents, laying the foundation for more better gaming agents.

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

## A    LLM USAGE DECLARATION

We employed LLMs to assist us in the writing of this paper. Our writing pipeline consisted of one of the authors first writing a draft paragraph, then using an LLM to assist in polishing the writing and grammar, and finally having other authors review and provide the final version to the text. Additionally, we employed foundational models to assist us in creating illustrations.

Finally, throughout the course of this research we used LLM-powered search engines like Perplexity in addition to traditional alternatives such as Google Scholar and conference proceedings while gathering relevant literature.

## B    BALROG GAME DETAILS

The BALROG framework incorporates six distinct gaming environments, each designed to evaluate specific aspects of agentic reasoning (Figure 3):

**BabyAI**    BabyAI is a grid-based environment with different difficulty levels. The agent is presented wth five different tasks.

**TextWorld**    TextWorld offers a text-based exploration environment where agents interact exclusively through natural language commands. There are three different tasks.

**Crafter**    Crafter simulates a Minecraft-inspired survival environment where progression is measured through 22 distinct achievements.

**BabaIsAI**    BabaIsAI presents a rule-based puzzle environment where agents must navigate grid-based scenarios. There are 40 different tasks.

**MiniHack**    MiniHack represents a task-oriented version of the classic NetHack (Küttler et al., 2020) game, evaluating agents across eight challenges testing different skills.

**NetHack Learning Environment (NLE)**    NLE implements the complete NetHack roguelike game, presenting the most comprehensive challenge within the benchmark. This environment simultaneously evaluates navigation, survival instincts, long-term strategic planning, resource management, and exploration skills within an unpredictable, dynamically evolving game state.

| Skills | BabyAI | TextWorld | Crafter | Baba Is AI | MiniHack | NLE |
|---|---|---|---|---|---|---|
| Navigation | ✔ | ✔ | ✔ | ✔ | ✔ | ✔ |
| Exploration | ✔ | ✔ | ✔ | ✔ | ✔ | ✔ |
| Resource Management | ✗ | ✔ | ✔ | ✗ | ✔ | ✔ |
| Complex Credit Assignment | ✗ | ✗ | ✔ | ✔ | ✔ | ✔ |
| Deducing Env. Dynamics | ✗ | ✗ | ✗ | ✔ | ✔ | ✔ |
| Long-term Planning | ✗ | ✗ | ✗ | ✔ | ✔ | ✔ |
| Turns to Complete | $10^1$ | $10^2$ | $10^3$ | $10^2$ | $10^2$ | $10^4$–$10^5$ |
| Time to Master for Humans | Seconds | Minutes | Hours | Hours | Hours | Years |

Figure 3: Game environments overview. Adapted from BALROG (Paglieri et al., 2025)

## C    METRIC SCORES

We adapt metric scores from BALROG to quantify how close an agent is to completing each task. All scores are normalised to a range of 0–100. The scoring scheme varies by environment. In MiniHack, BabyAI, and BabaIsAI, tasks give binary scores: either 0 (failure) or 100 (success). In contrast, TextWorld, Crafter, and NetHack return continuous scores between 0 and 100, where the

score represents the proportion of achievements completed. For NetHack specifically, the authors of BALROG introduced a novel scoring system based on data-informed metrics. Authors derived scores from the probability of a human player winning the game after reaching a particular dungeon level or experience level. The authors argue that this metric better captures meaningful progression than previous metrics in that game. We adopt the same procedure for NetHack in our work.

## D    HYPERPARAMETERS SELECTED

This Appendix details the choice of hyperparameters in our methodology. Table 4 details the values and descriptions.

The genetic algorithm optimises four hyperparameters: $\tau_{\cos}$, $\lambda$, $\epsilon_0$, and $T$. We establish lower bounds of 0 for $\tau_{\cos}$, $\lambda$, and $\epsilon_0$, effectively disabling these components when not beneficial to performance. Upper bounds were determined in order to maintaini sufficient search space for optimisation.

The language model temperature $T$ follows standard practice with a default value of 1.0, allowing the genetic algorithm to explore a range of solutions. We implement exponential decay for $\epsilon_t$ following established reinforcement learning approaches, enabling the transition from exploration to exploitation as the system learns optimal behaviours. We follow the short term memory length in BALROG and set it to $m = 16$. We set $k_{succ} = 5$ and $k_{fail} = 5$, limiting the number of information added to the prompt, but also adding significant amount of past experiences; through empirical evaluation we notice that a higher number of memories added is not beneficial.

Following DOMiNO methodology, we set $\alpha = 0.7$ to ensure meaningful population diversity whilst maintaining performance standards. Our similarity-recency weighting ($w_{\text{similarity}} = 0.7$, $w_{\text{recency}} = 0.3$) prioritises semantic relevance over temporal proximity, reflecting the hypothesis that content similarity is more beneficial than recency.

The genetic algorithm parameters balance computational efficiency with solution quality. We set probability of selecting single parent in genetic algorithm to be 70% (vs two parents to be 30%), allowing for more mutations without crossover operations. We set $n = 4$ iterations as empirical evaluation demonstrated satisfactory performance is achieved at this point, providing an effective balance between solution quality and computational cost. Population management parameters $N = M = 5$ maintain an optimal balance between preserving high-performing solutions and promoting genetic diversity, following established evolutionary computation principles that prevent premature convergence whilst ensuring computational tractability.

## E    BALROG BASELINE CONFIGURATIONS

This section details number of episodes and their length per each BALROG game. Moreover, we show the BALROG prompt that we use as initial seed for the baseline architecture.

**Episode details**    Table 5 details the episode specifications per game. The table shows time needed for each episode completion, as well as details on number of tasks per different environments.

**Baseline Prompt (BALROG)**    Below we present a prompt from BALROG (Paglieri et al., 2025) paper, used for Baseline evaluation.

> **Baseline BALROG Prompt**
>
> ```
> """You always have to output one of the above actions at a time and no other text. You always have
>      to output an action until the episode terminates."""
> ```

## F    TESTING THE RETRIEVAL MECHANISM LONG-TERM MEMORY SYSTEM

In this section we test the retrieval of saved memories and the abilities to act upon them. Due to stochasticity, we need to have a reliable comparison. We focus our evaluation on Crafter, as it is

| Parameter | Value | Description |
|---|---|---|
| Embedding | `all-MiniLM-L6-v2` | Pre-trained sentence embedding model used for semantic similarity calculations |
| d | 384 | Dimension of the embedding |
| $\tau_{\cos}$ | $[0, 0.1]$ | Cosine similarity threshold parameter |
| $\lambda$ | $[0, 0.1]$ | Long-Term Memory decay factor |
| $\epsilon_0$ | $[0, 0.1]$ | Initial exploration parameter |
| $T$ | $[0.1, 2]$ | Language Model Temperature |
| $\epsilon_t$ | $0.99^t \times \epsilon_0$ | Time-decayed parameter following exponential decay |
| $m$ | 16 | Short-term memory length (most recent action-observation pairs) |
| $k_{success}$ | 5 | Number of top scoring successful long-term memories added to the LLM prompt |
| $k_{fail}$ | 5 | Number of top scoring failed long-term memories added to the LLM prompt |
| $\alpha$ | 0.7 | Minimum fraction of highest scoring genome for diversity |
| $w_{similarity}$ | 0.7 | Weight assigned to similarity component in scoring |
| $w_{recency}$ | 0.3 | Weight assigned to recency component in scoring |
| $p_{single}$ | 70% | Percentage chance to choose single parent for reproduction |
| $1 - p_{single}$ | 30% | Percentage chance to choose two parents for reproduction |
| $n_{gen}$ | 4 | Number of iterations (parent population creation) of genetic algorithm |
| $n_{child}$ | 5 | Number of children created for each population of parents in genetic algorithm |
| $N$ | 5 | Number of best scoring genomes saved at each step of genetic algorithm |
| $M$ | 5 | Number of most diverse genomes saved at each step of genetic algorithm (scoring at least $\alpha$ fraction of top performing genome) |
| $n_{ep}$ | dependent on the game | Number of episodes for each child evaluation. Details in the Table 5 |

Table 4: Hyperparameter values used in the TAME framework

an environment requiring long-term planning, giving motivation to log-term memory approach. We disable life hazards such as zombies and skeletons, as they are not relevant to the testing subject. We set random seed to 32 for all episodes.

To evaluate our information-retrieval mechanism we test a long-horizon "craft iron sword" task in Crafter. We replace the environment's default objective with the production of an iron sword (see Appendix F.1.1). This task is intentionally complex: it requires chopping wood, crafting and placing a crafting table, crafting a wooden pickaxe, collecting stone, crafting a stone pickaxe, placing a

| Environment | Evals | Tasks per Eval | Total | Episode Length |
|---|---|---|---|---|
| BabyAI | 10 | 5 | 50 | $10^1$ |
| BabaIsAI | 3 | 40 | 120 | $10^2$ |
| Crafter | 10 | 1 | 10 | $10^3$ |
| TextWorld | 10 | 3 | 30 | $10^2$ |
| MiniHack | 5 | 8 | 40 | $10^2$ |
| NetHack | 5 | 1 | 5 | $10^4 - 10^5$ |

Table 5: Episode details per BALROG game

furnace adjacent to the crafting table, collecting iron, and finally crafting an iron sword. We selected this objective because, in prior baseline runs without our memory system, the agent never completed the task. Through the episode we would like to check if memories are activated at relevant time steps, showing retrieval ability. Moreover, successful completion under our system provides strong evidence that the memory-critic architecture supports multi-step planning and sequential options.

In order to track the memories activated, we inject five task-oriented memories at the start of each episode: "craft wooden pickaxe", "craft stone pickaxe", "mine iron", "place furnace", and "craft iron sword" (see Appendix F.1.2). Each memory is paired with a human-crafted critic that summarises the steps needed to achieve particular option. During each episode we log when each memory activates. An example progression through episode is provided in Figure 4 and the corresponding memories activated can be seen in Figure 5. More examples on activation timelines are provided in Appendix F.2.

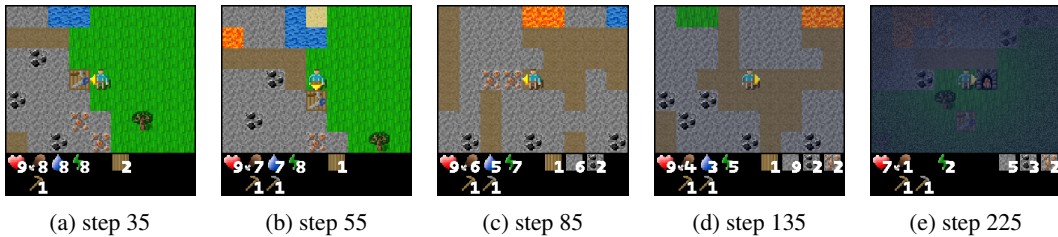

| (a) step 35 | (b) step 55 | (c) step 85 | (d) step 135 | (e) step 225 |

Figure 4: Testing memory retrieval: Task progression across episode: (a) agent crafts wood pickaxe, (b) agent crafts stone pickaxe, (c) agent collects iron, (d) agent collects iron, (e) agents attempts to craft iron sword

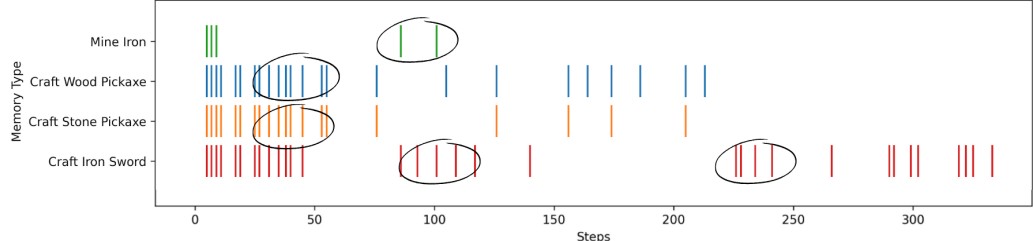

Figure 5: Testing memory retrieval: memories activated based on the step

**Discussion of the example:** From Figure 5 we can notice that all memories are activated at the beggining. This is due to the fact that very early in the game, those are they only memories present. We can then see that "Craft Wood Pickaxe" and "Craft Stone Pickaxe" are heavily retrieved until around 25th-60th step. This is when agent completed "Craft Stone Pickaxe task" (agent completes "Craft Wood Pickaxe" task earlier, but due to similarity of those two tasks, it it activated when focusing on stone version). "Mine Iron" memory activates two times between 80-100 steps, when agent is mining two pieces of iron. "Craft Iron Sword" appears often around step number 100 which is when agent first attempts to complete it, but realises that it needs to place a furnace and table first.

Then the memory is activated later as well, which is after placing table and furnace and attempting to craft iron sword. Unfortunately, agent is unsuccessful because it didn't place furnace close enough to the table. This experiment demonstrates that relevant memories are activated at the right time, and that agent is able to act upon them. Also, when not needed (i.e. the task is completed), memories are activated far less often. It is also important to notice that the "Craft Iron Sword" memory continues to be retrieved even towards the end of the episode when the agent is actively attempting this task. This indicates that the memory system maintains access to relevant historical experiences throughout the entire episode, regardless of when they were initially formed.

**Results**   Across 10 independent runs the agent succeeded in producing an iron sword in 1/10 episodes (baseline: 0% from all previous runs that we did with Gemini-2.0-Flash). This is a substantial improvement, which suggests that when memories are stored, agent is able to retrieve them, and act upon them. This is a simplified case, as we provided human crafted memories, but with the right prompting we believe that the critic module will be able to reproduce those. Additional observations:

- Memories reliably activated when their prerequisites were satisfied and were deactivated immediately after the corresponding option was completed.
- The specificity of the critic strongly affected performance. For example, phrasing a critic as "place the furnace next to the crafting table" versus "place the furnace adjacent to the table" produced different success scores. This highlights the value of precise, action-oriented critic definitions that focus on the key state features leading to success or failure. Following on that we prompted the critic accordingly.
- This experiment also demonstrates the difficulty of the "Craft Iron Sword" task, even when given with clear instructions agent fails 90% of the time.

### F.1   TESTING MEMORY RETRIEVAL PROMPTS

This subsection details prompts used in order to test long-term memory retrieval. First we show the prompt detailing the goal of iron sword creation, then we show memories added at the begging of the episode: *Craft Wood Pickaxe, Craft Stone Pickaxe, Craft Stone Sword, Mine Iron, Place Furnace, Create Iron Sword*. Those are hand-crafted memories, designed in order to track memory retrieval.

#### F.1.1   IRON SWORD GOAL PROMPT

This subsection details the prompt for Craft Iron Sword goal that the agent is tasked with during the long-term memory retrieval experiment.

---

**Craft Iron Sword Goal**

```
    """ You are playing Crafter. The following are the only valid actions you can take in the game,
    ↪ followed by a short description of each action:

{action_strings}.
Your goal is to craft an iron sword. """
```

---

#### F.1.2   INJECTED MEMORIES

This subsection details the memories added at the beggining of the episode, in order to track memory retrieval.

---

**Craft Wood Pickaxe Memory**

```
craft_wood_pickaxe = {
"name": "Craft Wood Pickaxe",
"description": "Craft Wood Pickaxe for gathering stone",
"subgoal_prerequisites": "Agent has 1 piece of wood in inventory and table is placed",
"success_condition": "Wood Pickaxe is in inventory",
"subgoal_progress_indicators": "Agent is gathering wood near table",
"subgoal_penalty_component": "Agent crafts pickaxe without enough wood",
```

---

```
"status": 'successful',
"summary of the run": "Agent collects three pieces of wood and places a table in a clear spot. Then
    ↪ agent crafts a wood pickaxe at the table.",
}
```

### Craft Stone Pickaxe Memory

```
stone_pickaxe_memory = {
"name": "Craft Stone Pickaxe",
"description": "Craft Stone Pickaxe",
"subgoal_prerequisites": "Agent has 1 piece of stone and 1 piece of wood in inventory and table is
    ↪ placed",
"success_condition": "Stone Pickaxe is in inventory",
"subgoal_progress_indicators": "Agent is gathering stone near table",
"subgoal_penalty_component": "Agent crafts pickaxe without enough resources",
"status": 'successful',
"summary of the run": "Agent collects four pieces of wood and places a table in a clear spot. Then
    ↪ agent collects one piece of stone using wood pickaxe. Then agent crafts a stone pickaxe at
    ↪ the table.",
}
```

### Craft Stone Sword Memory

```
stone_sword_memory = {
"name": "Craft Stone Sword",
"description": "Craft Stone Sword for combat",
"subgoal_prerequisites": "Agent has 1 piece of stone and 1 piece of wood in inventory and table is
    ↪ placed",
"success_condition": "Agent has stone sword in inventory",
"subgoal_progress_indicators": "Agent has 1 pieces of stone and 1 piece of wood",
"subgoal_penalty_component": "Agent crafts sword without enough resources",
"status": 'successful',
"summary of the run": "Agent collects 4 pieces of wood. Then agent places a table and crafts a wood
    ↪ pickaxe. Lastly, agent uses wood pickaxe to craft 2 pieces of stone and crafts a stone sword
    ↪ at the table.",
}
```

### Mine Iron Memory

```
mine_iron = {
"name": "Mine Iron",
"description": "Mine Iron",
"subgoal_prerequisites": "Agent has 1 wood pickaxe in inventory",
"success_condition": "Wood Pickaxe is in inventory",
"subgoal_progress_indicators": "Agent is gathering iron near table or furnace",
"subgoal_penalty_component": "Agent iron sword without enough resources",
"status": 'failed',
"summary of the run": "Agent repeatedly tried 'Do' action using pickaxe near iron but fails to
    ↪ collect iron. It is recommended agent tries using different tool.",
}
```

### Place Furnace Memory

```
furnace_memory = {
    "name": "Place Furnace next to Table",
    "description": "Place furnace next to the Table for crafing iron tools",
    "subgoal_prerequisites": "Agent has 4 pieces of stone in inventory. Table is placed. ",
    "success_condition": "Furnace is placed",
    "subgoal_progress_indicators": "Agent is gathering stone",
    "subgoal_penalty_component": "Agent places furnace in unsuitable location or without enough
        ↪ stone",
    "status": 'successful',
    "summary of the run": "Agent placed a furnace next to the table using 4 pieces of stone. ",
}
```

**Craft Iron Sword Memory**

```
iron_sword_memory = {
"name": "Craft Iron Sword",
"description": "Craft Iron Sword",
"subgoal_prerequisites": "Agent has 1 piece of stone, 1 piece of wood, 1 piece of coal in inventory
    ↪ and table and furnace is placed next to each other",
"success_condition": "Iron Pickaxe is in inventory",
"subgoal_progress_indicators": "Agent is gathering iron near table and furnace",
"subgoal_penalty_component": "Agent iron sword without enough resources",
"status": 'successful',
"summary of the run": "Agent collects 1 piece of stone and one piece of wood using wood pickaxe.
    ↪ Then agent collects one piece of iron using stone pickaxe. Then agent crafts an iron sword
    ↪ next to the table and furnace.",
}
```

### F.2 ADDITIONAL LONG-TERM MEMORY RETRIEVAL EXPERIMENTS

This section shows additional experiemnts carried out in order to test memory retrieval, when tasked the agent with iron sword task. Eachof the experiments consists of images showing agent progression, as well as memory activation across the episode.

#### F.2.1 EPISODE 1

Illustrations from the game available in Figure 6 and memory activations in 7. The reason why agent didn't succeed in completing the task is because the agent didn't place table close enough to furnace.

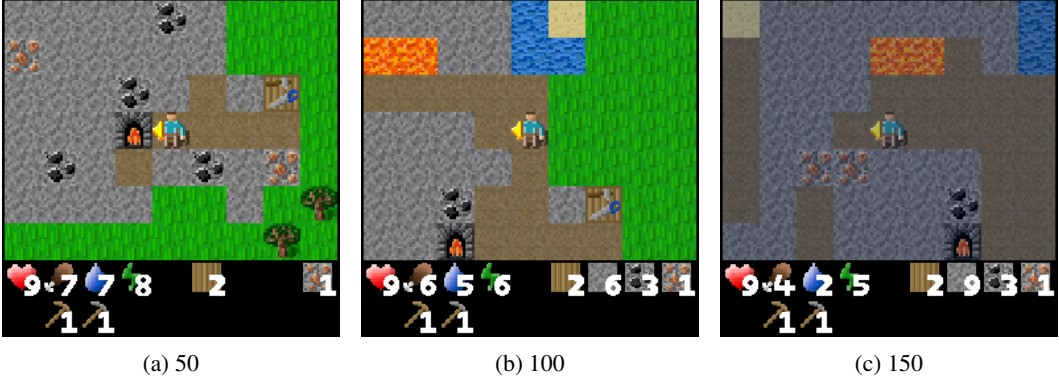

(a) 50          (b) 100          (c) 150

Figure 6: Episode 1: Task progression across episode

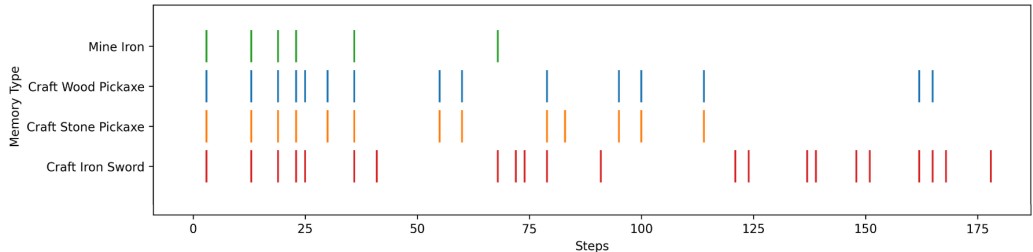

Figure 7: Episode 1: Memory retrieval along the episode

#### F.2.2 EPISODE 2

Illustrations from the game available in Figure 8 and memory activations in 9. The reason why agent didn't succeed in completing the task is because the agent focuses on placing furnaces a few times.

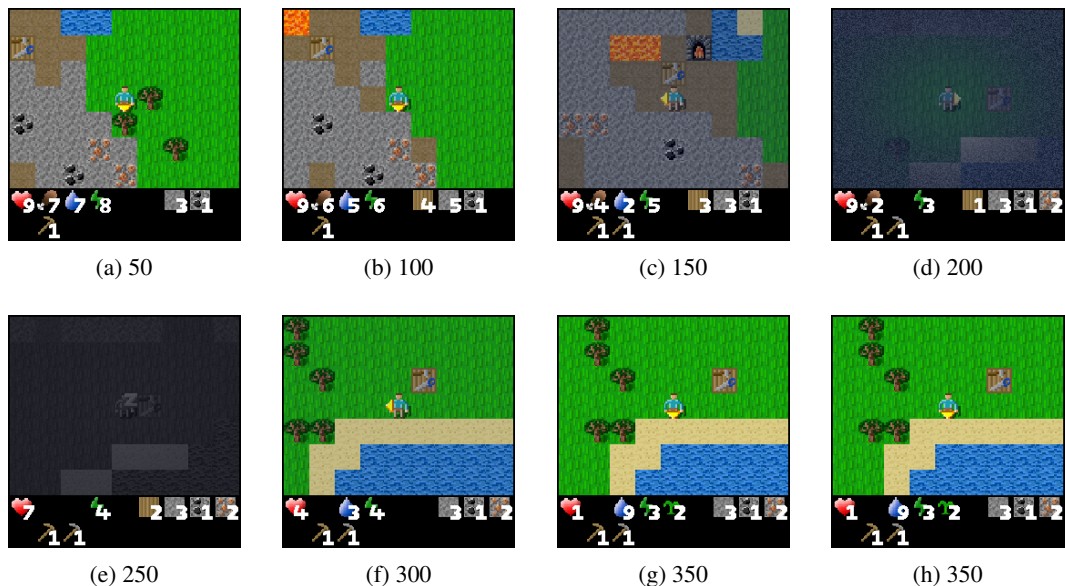

| (a) 50 | (b) 100 | (c) 150 | (d) 200 |

| (e) 250 | (f) 300 | (g) 350 | (h) 350 |

Figure 8: Episode 2: Task progression across episode

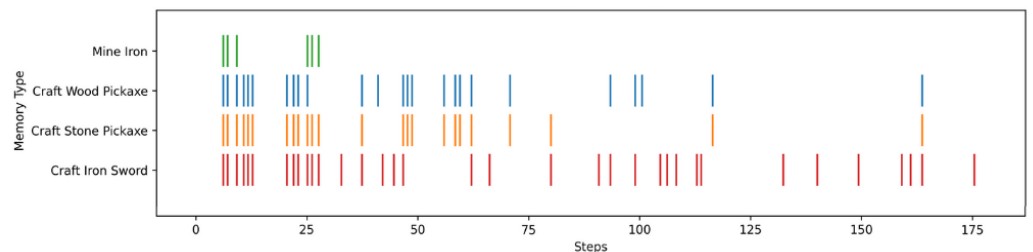

Figure 9: Episode 2: Memory retrieval along the episode

### F.2.3   EPISODE 3

Illustrations from the game available in Figure 10 and memory activations in 11. The reason why agent did not succeed in completing the task is because agent does not have enough wood (also crafts multiple tables and furnaces).

### F.2.4   EPISODE 4

Illustrations from the game available in Figure 12 and memory activations in 13. The reason why agent did not succeed in completing the task is because agent does nott have enough wood (also crafts multiple tables and furnaces).

## G   EXAMPLES OF MODULES OUTPUTS

This section shows examples of amygdala and explorer modules behaviours.

### G.1   AMYGDALA

This section details Amygdala submodule. First, we show defualt survival option, then we illustrate an example of amygdala in the episode (using Crafter).

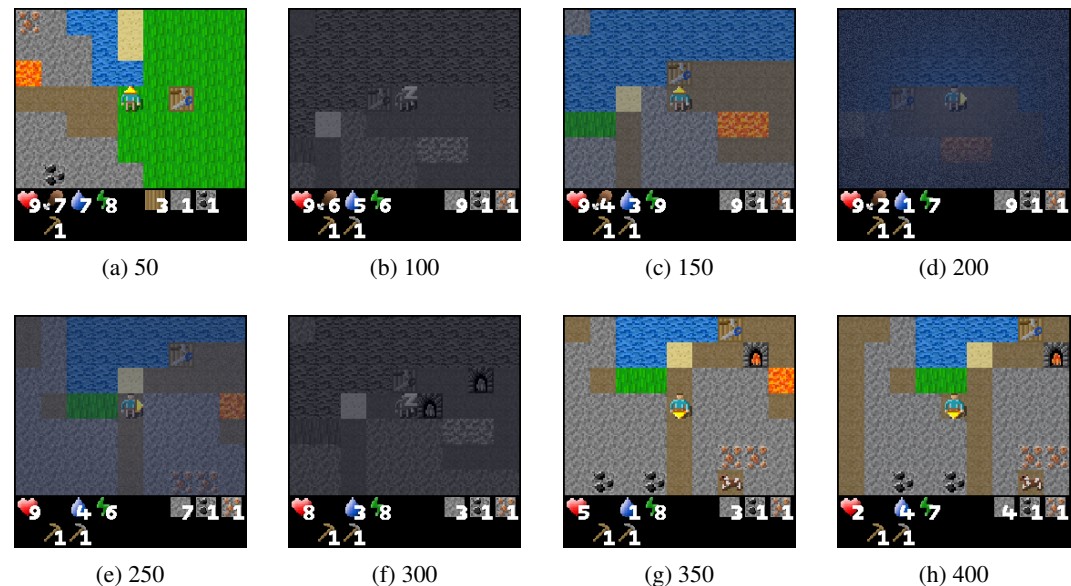

Figure 10: Episode 3: Task progression across episode

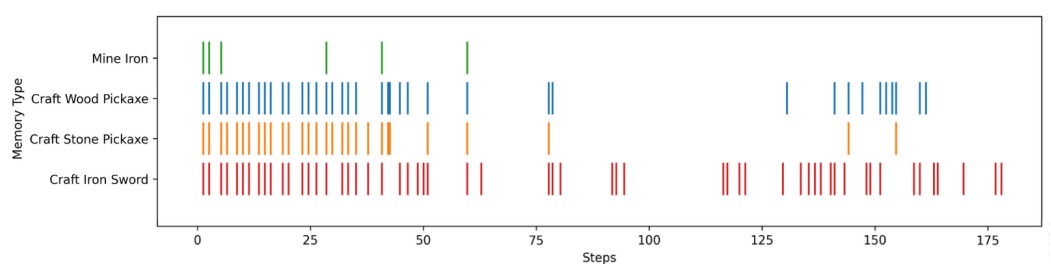

Figure 11: Episode 3: Memory retrieval along the episode

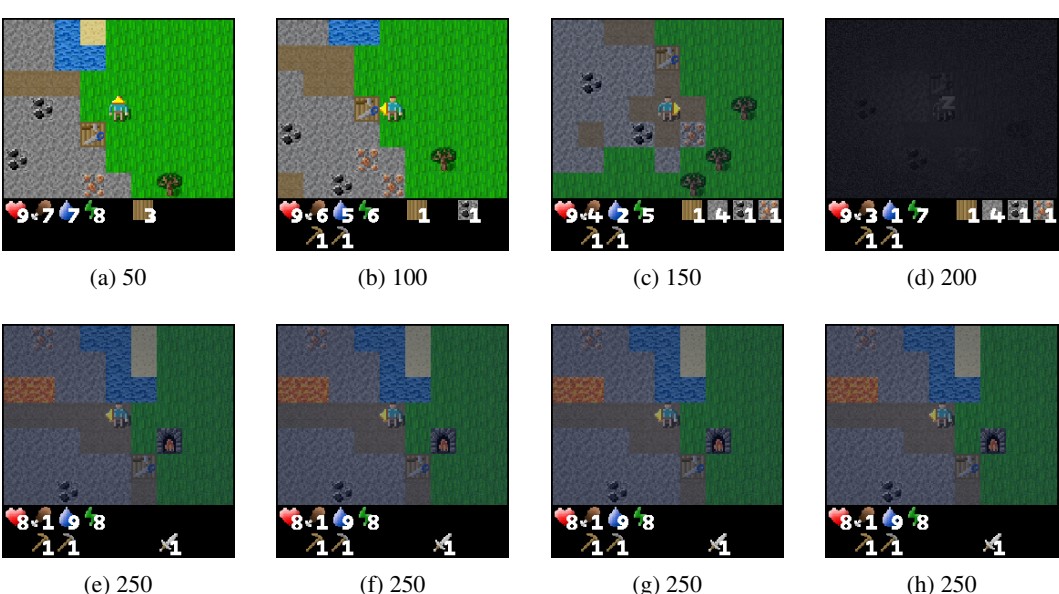

Figure 12: Episode 4: Task progression across episode

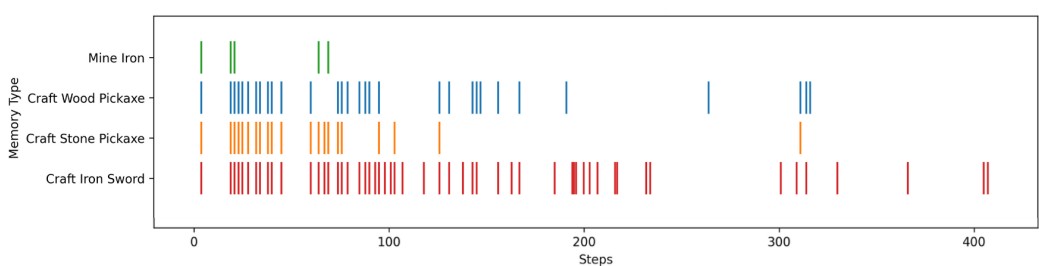

Figure 13: Episode 4: Memory retrieval along the episode

**Default survival option**    The prompt below shows default survival option, that agent is switching to, whenever danger is encountered.

---
**Default Survival Option**

```
    subgoal =  {
    "name" : "respond to danger",
    "description": "Respond to danger.",
    "prerequisites": "Agent is in immediate danger",
    "success_condition": "Agent has eliminated the danger",
    "penalty_component": "Agent is not responsing to danger.",
    "progress_indicators": "Agent is closer to eliminating the danger",
    "estimated_priority": "high"
}
```
---

**Amygdala module activation.**    Here we present an example from Crafter when amygdala is activated. The agent is initially focused on exploring the environment and fulfilling subgoals such as placing plants. However, in the early stages of the game, a skeleton appears. As soon as the agent observes the skeleton within its field of view, it activates survival mode.

After survival mode is activated, the agent begins gathering resources for combat:

1. Chops down wood
2. Places a crafting table
3. Creates a wooden sword necessary for the fight

The agent then chases the skeleton, and once it is adjacent to the enemy, it initiates combat. During the chase and fight, the agent loses health but successfully manages to defeat the skeleton. Shortly afterwards, the amygdala is deactivated and the agent returns to working towards general game objective.

### G.2    EXPLORER

In this section we show an example of exploration plan proposed by Explorer.

```
{'reasoning': 'The agent needs to break out of its repetitive westward
movement and explore new areas, prioritizing resource gathering and
different directions. Sand has been identified as a new area to explore.',
'subgoals': [
  {'name': 'Move Towards Sand',
   'description': 'Move towards the sand to the south-east to explore
   new terrain.',
   'prerequisites': '',
   'success_condition': 'Agent is standing on sand.',
   'penalty_component': 'Moving in the opposite direction (North or West)
   for more than 3 steps without collecting resources.',
   'progress_indicators': 'Decreasing distance to sand in long_term_context.',
   'estimated_priority': 'high'},
```

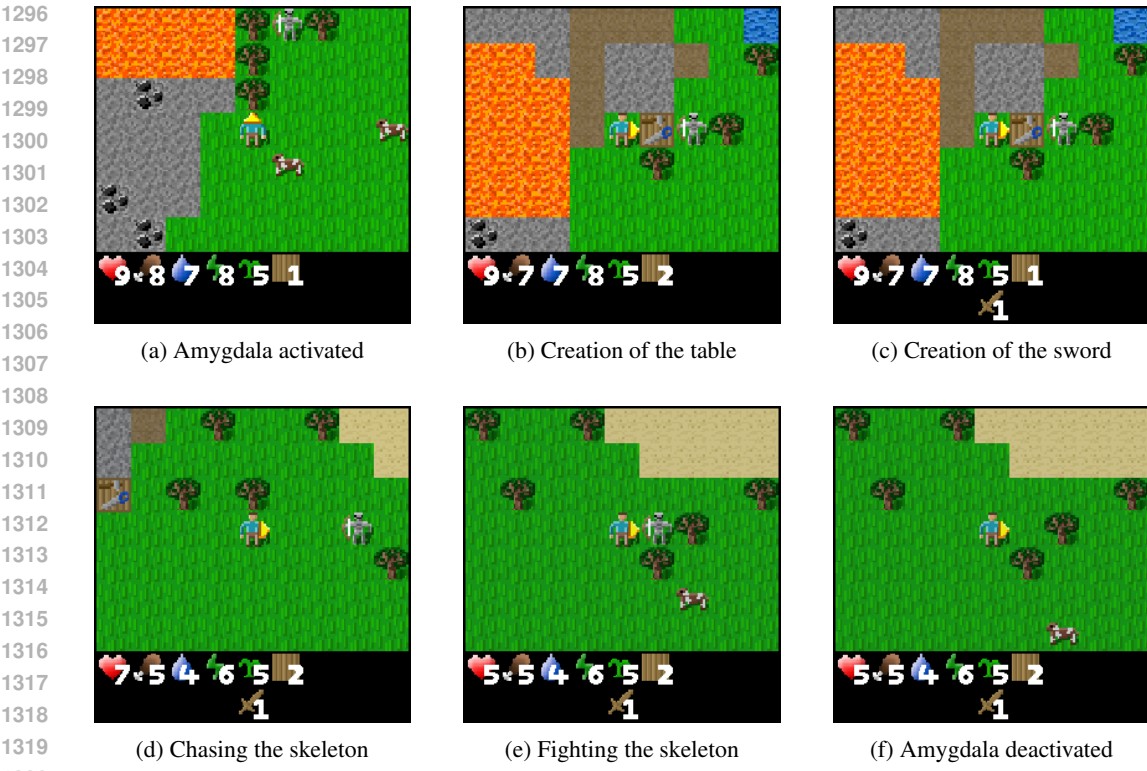

(a) Amygdala activated      (b) Creation of the table      (c) Creation of the sword

(d) Chasing the skeleton      (e) Fighting the skeleton      (f) Amygdala deactivated

Figure 14: Amygdala survival response sequence: (a) activation, (b)&(c) resource gathering, (d)&(e) combat, (f) deactivation.

```
{'name': 'Explore Area Around Sand',
 'description': 'Once on the sand, explore in all directions (North, South,
 East, and West) to discover resources and new landmarks.',
 'prerequisites': 'Agent is standing on sand.',
 'success_condition': 'Agent has moved at least 3 steps in each cardinal
 direction from the sand.',
 'penalty_component': 'Staying within the same 3x3 area on the sand for
 more than 5 steps.',
 'progress_indicators': 'Number of unique tiles visited around the sand.',
 'estimated_priority': 'medium'}
]}
```

## H  DETAILED GENETIC ALGORITHM

In this Appendix we present the detailed specifications of the genetic algorithm.
Parameter selection:

$$\text{Generations} = 4 \tag{17}$$

$$\text{Population size} = |\mathcal{P}_{\text{diverse}}| + |\mathcal{P}_{\text{best}}| = 5 + 5 = 10 \tag{18}$$

$$\text{Children per generation} = 5 \tag{19}$$

$$p_{\text{binary}} = p_{\text{continuous}} = p_{\text{prompt}} = 0.5 \tag{20}$$

where:

- **Generations** – the number of evolutionary iterations performed
- **Population size** – the total number of genomes maintained across diverse and best-performing subpopulations

- **Children per generation** – the number of new offspring genomes created through mutation in each generation

- $p_{\text{binary}}, p_{\text{continuous}}, p_{\text{prompt}}$ – the probabilities of applying binary, continuous, or prompt mutation operations, respectively, when creating offspring.

The number of generations was limited to four due to the significant computational cost and long runtimes associated with evaluating each genome, particularly in environments like NetHack. However, this was sufficient to demonstrate a clear performance improvement and allow for the discovery of specialised architectures.

**Parent Selection** Parent selection follows roulette wheel selection (a probabilistic parent-picking method where each individual's chance of being chosen is proportional to its fitness score) with fitness-proportionate probabilities:

$$P(\mathbf{g}_i) = \frac{f(\mathbf{g}_i)}{\sum_{j=1}^{|\mathcal{P}|} f(\mathbf{g}_j)}$$

where $f(\mathbf{g}_i)$ is the fitness score of genome $\mathbf{g}_i$.

**Offspring Generation** For each child $\mathbf{g}_{\text{child}}$:

- With probability $p_{single}$: select one parent
- With probability $1 - p_{single}$: select two parents

For each genome component, evolutionary operations are applied with probability $p_{binary}, p_{continuous}, p_{prompt}$, depending on the component. Otherwise parent attributes are copied directly. In the two-parent case, the parent from which to copy each attribute is chosen with Bernoulli(0.5) probability.

### H.1 GENETIC OPERATIONS

This subsection focuses on methodology of crossover and mutations operations.

**Modules Operations** Single parent:

$$b_i^{(child)} = \begin{cases} b_i^{(parent)} & \text{with probability } 0.8 \\ 1 - b_i^{(parent)} & \text{with probability } 0.2 \end{cases}$$

Two parents:

$$b_i^{(child)} = \begin{cases} b_i^{(p1)} = b_i^{(p2)} & \text{with probability } 0.9 \text{ if } b_i^{(p1)} = b_i^{(p2)} \\ 1 - b_i^{(p1)} & \text{with probability } 0.1 \text{ if } b_i^{(p1)} = b_i^{(p2)} \\ \text{Bernoulli}(0.5) & \text{if } b_i^{(p1)} \neq b_i^{(p2)} \end{cases}$$

**Hyperparameter Operations** The continuous value inheritance depends on parent activity states. Let $A_i^{(pk)}$ indicate if feature $i$ is active in parent $k$:

For single parent:

$$c_i^{(child)} = \text{clip}(c_i^{(parent)} + \mathcal{N}(0, \sigma^2), c_{i,\text{min}}, c_{i,\text{max}})$$

For two parents:

$$c_i^{(child)} = \begin{cases} c_i^{(p1)} + \mathcal{N}(0, \sigma^2) & \text{if } A_i^{(p1)} = 1, A_i^{(p2)} = 0 \\ c_i^{(p2)} + \mathcal{N}(0, \sigma^2) & \text{if } A_i^{(p1)} = 0, A_i^{(p2)} = 1 \\ \alpha c_i^{(p1)} + (1 - \alpha)c_i^{(p2)} + \mathcal{N}(0, \sigma^2) & \text{if } A_i^{(p1)} = A_i^{(p2)} = 1 \\ c_{i,\text{default}} & \text{if } A_i^{(p1)} = A_i^{(p2)} = 0 \end{cases}$$

where $\alpha \sim U(0, 1)$. $c_{i,min}, c_{i,max}, c_{i,default}$ are detailed in Appendix D.

**Prompt Operations** Prompt evolution utilises the EvoPrompt prompt methodology Guo et al. (2025). Using a similar approach we use LLM as a crossover and mutation operator.

- **Single parent**: $p_i^{(child)} = \text{LLMprompt}_{mutate}(p_i^{(parent)})$
- **Two parents**: $p_i^{(child)} = \text{LLMprompt}_{mutate}(\text{LLMprompt}_{crossover}(p_i^{(p1)}, p_i^{(p2)}))$

If a parent has module $i$ disabled ($b_i = 0$), the corresponding prompt reverts to default: $p_i = p_{i,\text{default}}$ (default prompts in Appendix Q.1).

LLM mutation and crossover prompts available in Appendix P. It is important to notice that this approach enables functional mutation: LLM is prompted with all possible functional inputs to be used in any prompt (described in Section 3.1).

### H.2 POPULATION MANAGEMENT

This subsection focuses on population management: details about fitness function, diversity measure and population pruning.

**Fitness Evaluation** Each genome is evaluated using the fitness function:

$$f(\mathbf{g}) = \frac{1}{n_{ep}} \sum_{i=1}^{n_{ep}} \text{GameProgression}_i(\mathbf{g})$$

where $n_{ep}$ is the number of episodes for specific game (see Table 5).

**Diversity measure** The genome distance function uses an embedding-based approach where each genome is represented as a single embedding vector (here we use sentence-transformers/all-MiniLM-L6-v2 embedding). The distance between two genomes is calculated using cosine similarity:

$$d(\mathbf{g}_1, \mathbf{g}_2) = 1 - \cos(\mathbf{e}_1, \mathbf{e}_2) \tag{21}$$

where $\mathbf{e}_i$ is the embedding vector representation of genome $\mathbf{g}_i$.

The cosine similarity between two embedding vectors is computed as:

$$\cos(\mathbf{e}_1, \mathbf{e}_2) = \frac{\mathbf{e}_1 \cdot \mathbf{e}_2}{\|\mathbf{e}_1\|_2 \|\mathbf{e}_2\|_2}$$

Each genome $\mathbf{g} = \{\mathbf{b}, \mathbf{c}, \mathbf{p}\}$ is transformed into a unified embedding vector $\mathbf{e} \in \mathbb{R}^d$ that captures the semantic representation of all genome components (binary variables, continuous parameters, and prompts) in a single high-dimensional space. The way we measure diversity, is the minimum distance to the genomes already existing in the archive.

**Population Pruning** After children evaluation, population pruning maintains diversity using the following algorithm:

1. Initialise $\mathcal{P}_{\text{new}} = \emptyset$
2. Add top-N scoring genomes: $\mathcal{P}_{\text{new}} \leftarrow \text{top}_5(\mathcal{P})$
3. For remaining genomes $\mathcal{G}_{\text{remaining}}$:
   (a) Calculate minimum distance to current population:
   $$d_{\min}(\mathbf{g}) = \min_{\mathbf{g}' \in \mathcal{P}_{\text{new}}} d(\mathbf{g}, \mathbf{g}')$$
   (b) Select genome maximising diversity with performance constraint:
   $$\mathbf{g}^* = \arg \max_{\mathbf{g} \in \mathcal{G}_{\text{remaining}}} d_{\min}(\mathbf{g}) \quad \text{s.t. } f(\mathbf{g}) \geq 0.7 \cdot f(\mathbf{g}_{\text{best}})$$
   (c) Add $\mathbf{g}^*$ to $\mathcal{P}_{\text{new}}$ and remove from $\mathcal{G}_{\text{remaining}}$
4. Repeat step 3 until the desired population size reached ($N + M$)

### H.3 GENETIC APPROACH: PSEUDO CODE

---

**Algorithm 1** TAME: Genetic Algorithm

---

**Require:** Game environment
**Ensure:** Optimised genome $g^*$
  1: Initialize $P_0 = \{g_{\text{basic}}, g_{\text{hierarchical}}, g_{\text{default}}, g_{\text{full}}\}$
  2: **for** each $g \in P_0$ **do**
  3:    $g_{\text{fitness}} \leftarrow \text{EvaluateFitness}(g)$
  4: **end for**
  5: $P \leftarrow P_0$
  6: **for** generation $= 1$ to GENERATIONS **do**
  7:    $C \leftarrow \emptyset$ {Children population}
  8:    **for** $i = 1$ to CHILDREN_PER_GENERATION **do**
  9:      **if** rand() $< 0.7$ **then**
 10:        $p_1 \leftarrow \text{RouletteWheelSelection}(P)$
 11:        $c \leftarrow \text{SingleParentOperations}(p_1)$
 12:      **else**
 13:        $p_1, p_2 \leftarrow \text{RouletteWheelSelection}(P, 2)$
 14:        $c \leftarrow \text{TwoParentOperations}(p_1, p_2)$
 15:      **end if**
 16:      $c_{\text{fitness}} \leftarrow \text{EvaluateFitness}(c)$
 17:      $C \leftarrow C \cup \{c\}$
 18:    **end for**
 19:    $P \leftarrow \text{PopulationPruning}(P \cup C)$
 20: **end for**
 21:
 22: **return** $\arg\max_{g \in P} g_{\text{fitness}} = 0$

---

## I MODULES ACTIVATED

In this Appendix we discuss module activation based on the game. The results are taken from the first one of genetic algorithm using Gemini-2.0-Flash. Module activation is based on final genomes returned by genetic algorithm, available in Appendix R. Module activation plot is demonstrated in Figure 15. When selecting the baseline configuration, no additional modules apart from Long-Term Memory can be activated. We notice that 4 out of 6 environments selected hierarchical module, highlighting the effectiveness of complex goal decomposition. TextWorld is a text-based environment where it is difficult to predict next actions due to their dependence on current observation, therefore hierarchical structure and memory are not adding value. Moreover, MiniHack has relatively short length (100 steps), which might be also why baseline structure was favoured.

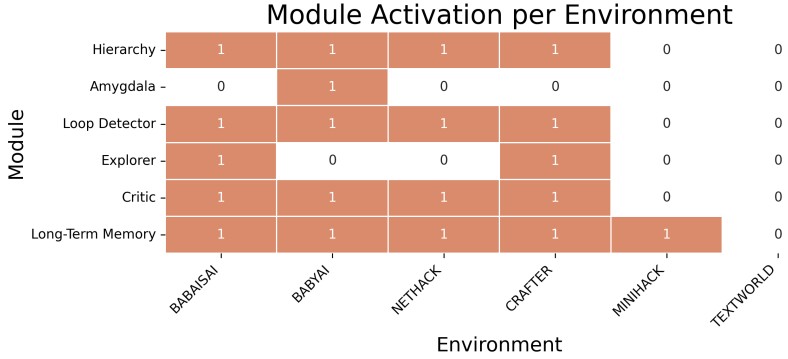

Figure 15: Module activation in TAME across environments.

## J TASK PERFORMANCE

In this Appendix, we compare different tasks performance across BabyAI, BabaIsAI, and Minihack for baseline versus TAME (first run using Gemini-2.0-Flash). Crafter and Nethack are excluded because they each have only one default task. TextWorld is also excluded since its genetic output matches the baseline.

**MiniHack** Interestingly, five out of eight tasks are never solved by any method, showcasing the difficulty (see Figure 16). The Corridor-R3 task, which is never completed by the baseline, nevertheless shows 40% progress with TAME. Corridor-R3 is an exploration problem in which the goal is to find the staircase Team (2024), illustrating TAME agent's improved exploration ability. In both CorridorBattle-Dark and MazeWalk-9×9, TAME achieves higher performance. CorridorBattle-Dark requires the agent to fight monsters, thereby testing planning and memory MiniHack Team, whereas MazeWalk-9×9 is a maze in which the agent must reach a terminal goal, testing exploration and memory Samvelyan et al. (2021).

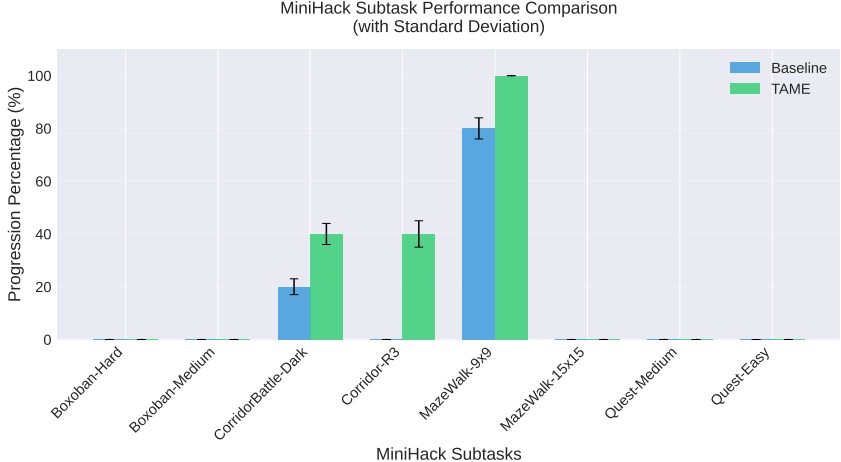

Figure 16: MiniHack tasks progression, Baseline vs TAME using Gemini-2.0-Flash.

**BabyAI** We observe clear performance improvements across all tasks except "putnext" where both agents achieve 0% success rate (see Figure 17). The putnext task presents significant challenges due to its complex spatial reasoning requirements. Through empirical analysis, we identified that agents fail to understand the necessary positioning strategy: they must navigate to a location one step away from the target position (which is adjacent to the object) before dropping the item. The persistence of this failure in our improved method highlights fundamental limitations in the agent's spatial reasoning capabilities. In all the other tasks, we notice an improvement when comparing TAME with baseline.

**BabaIsAI** This environment consists of 40 distinct tasks that can be categorised into four main types: make_win, make_you, goto_win, and make_wall_win. Our analysis reveals substantial improvements in the goto_win category and notable progress in make_win tasks, where performance increased from 0% baseline (see Figure 18). When examining performance across different room configurations (two_room versus single_room layouts), we observe consistent improvements in both settings. For difficulty categorisation, we define three levels based on task complexity: simple tasks have no modifiers or distractors, medium tasks contain 1-2 modifiers/distractors, and complex tasks have more than 2 modifiers/distractors. Most notably, the greatest performance gains occur in medium and complex categories, demonstrating that our method is particularly effective for challenging scenarios that require sophisticated reasoning capabilities.

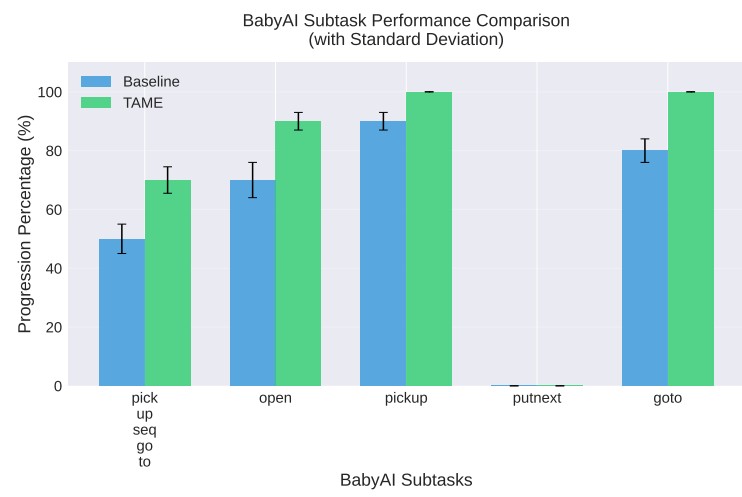

Figure 17: BabyAI tasks progression, Baseline vs TAME using Gemini-2.0-Flash.

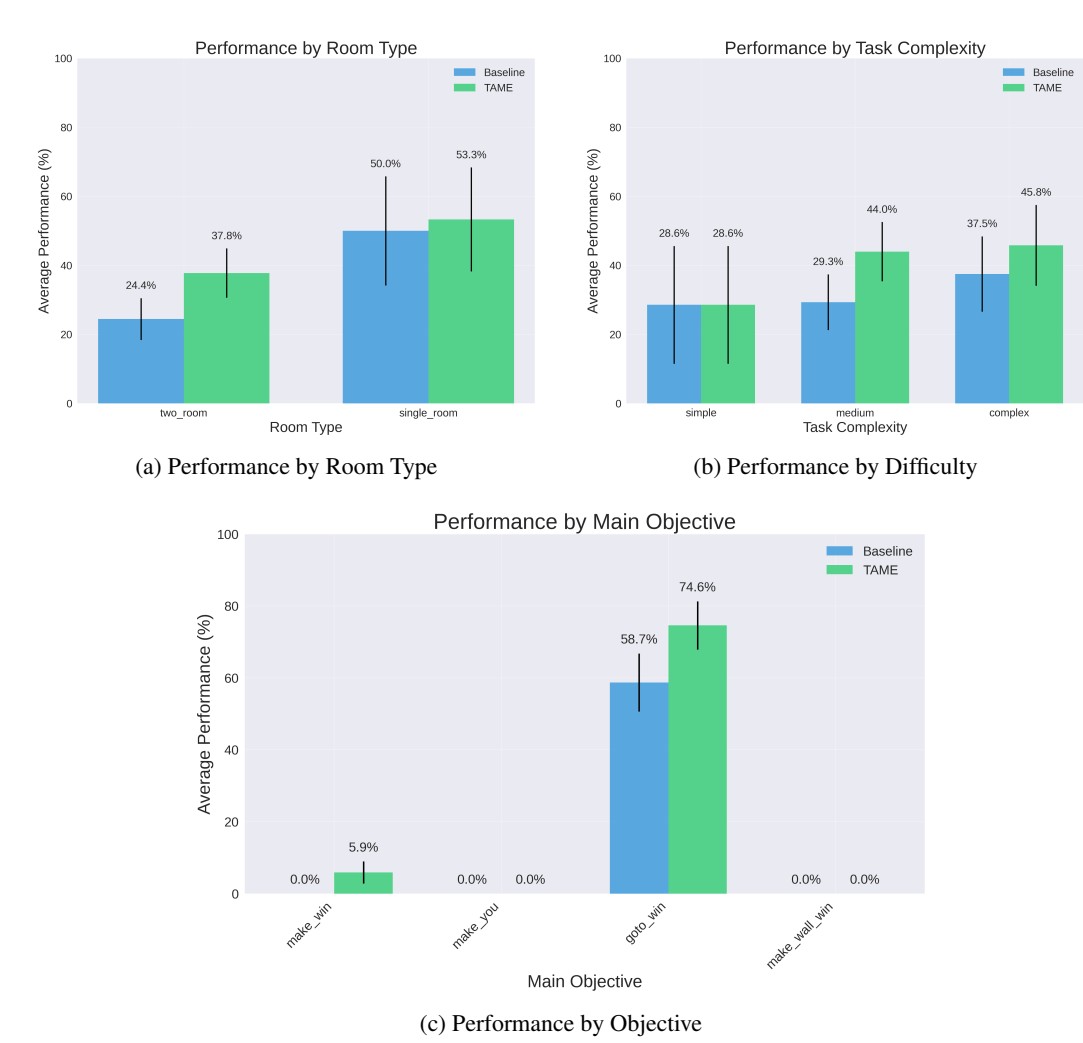

(a) Performance by Room Type

(b) Performance by Difficulty

(c) Performance by Objective

Figure 18: BabaIsAI task progression, Baseline vs TAME using Gemini-2.0-Flash

# K    DETAILS OF GEMINI EXPERIMENTS

This section details the setup and exact result of our Gemini model family. In our experimentation we repeat the genetic algorithm 3 times in order to access its variance across runs. The number of runs per episode remains the same as in BALROG benchmark.

## K.1    'IMPROVED PROMPT' GEMINI RESULTS

First, we compare Baseline together with "Improved Prompt", and TAME. "Improved Prompt" (see Appendix O) is created by prompting Claude-Sonnet-4.5 for a prompt that improves the Baseline prompt and allowing it for functional inputs as described in 3.1 . This experiment demonstrates the difficulty of crafting a prompt that works well for every environment, motivating our genetic approach, adapting the framework based on the game.

| Environment | Baseline (↑) | ImprovedPrompt[Claude] (↑) | TAME (↑) | Episodes |
|---|---|---|---|---|
| **Average** | 27.16% ± 2.12% | 23.36% ± 1.44% | **35.05% ± 2.24%** | - |
| babyai | 58.00% ± 6.98% | **78.00% ± 5.86%** | 72.00% ± 6.65% | 50 |
| babaisai | 30.83% ± 4.22% | 27.50% ± 4.08% | **42.50% ± 4.51%** | 120 |
| textworld | 32.55% ± 6.95% | 0.78% ± 0.77% | **33.40% ± 7.23%** | 30 |
| crafter | 29.09% ± 4.51% | 26.36% ± 2.47% | **38.18% ± 4.25%** | 10 |
| minihack | 12.50% ± 5.23% | 7.50% ± 4.16% | **23.33% ± 6.69%** | 40 |
| nle | 0.00% ± 0.00% | 0.00% ± 0.00% | **0.91% ± 0.44%** | 5 |

Table 6: "Improved Prompt" results with Gemini-2.0-flash

## K.2    SCORES ACROSS 3 GENETIC ALGORITHM RUNS WITH GEMINI-2.0-FLASH

In Table 7 we see detailed scores across all environments and across all 3 independent genetic algorithm runs. The main results in the Table 1 are averaged across those 3 runs.

| Environment | Baseline (↑) | TAME[Run1] (↑) | TAME[Run2] (↑) | TAME[Run3] (↑) |
|---|---|---|---|---|
| **Average** | 27.16% ± 2.12% | 34.78% ± 2.22% | 34.87% ± 2.20% | 35.52% ± 2.24% |
| babyai | 58.00% ± 6.98% | 72.0% ± 6.35% | 68.0% ± 6.60% | 76.0% ± 6.04% |
| babaisai | 30.83% ± 4.22% | 41.67% ± 4.50% | 43.33% ± 4.52% | 42.5% ± 4.51% |
| textworld | 32.55% ± 6.95% | 32.55% ± 6.95% | 32.55% ± 6.95% | 35.10% ± 7.78% |
| crafter | 29.09% ± 4.51% | 39.09% ± 4.9% | 39.55% ± 3.91% | 35.91% ± 4.33% |
| minihack | 12.50% ± 5.23% | 22.5% ± 6.6% | 25.0% ± 6.85% | 22.50% ± 6.61% |
| nle | 0.00% ± 0.00% | 0.85% ± 0.47% | 0.79% ± 0.44% | 1.10% ± 0.41% |

Table 7: Detailed scores across 3 genetic algorithm runs with Gemini-2.0-Flash.

## K.3    INITIAL POPULATION SCORES

This section details the scores of initial population used during genetic algorithm. Scores per environment can be seen in Table 8. The details of each components of initial population can be found in the Appendix Q.2.

## K.4    INITIAL POPULATION IMPROVEMENT

In order to access the performance gain of genetic algorithm, we compare it against initial population. In the Table 9 we present the percentage improvement of final TAME score when compared to the best score from initial population (see Appendix K.3). Those results show that even with relatively small budget, TAME structure is able to improve on average of 25.27% when compared to initial population, showcasing the effectiveness of the genetic algorithm over human-crafted initial population.

| Environment | Baseline (↑) | Hierarchical[Hand-Crafted] (↑) | Full[Claude] (↑) | Full[Hand-Crafted] (↑) |
|---|---|---|---|---|
| **Average** | $38.80\% \pm 5.32\%$ | $38.48\% \pm 4.06\%$ | $34.53\% \pm 3.99\%$ | $33.93\% \pm 4.07\%$ |
| babyai | $58.00\% \pm 6.98\%$ | $62.00\% \pm 6.86\%$ | $58.00\% \pm 6.98\%$ | $65.31\% \pm 6.80\%$ |
| babaisai | $30.83\% \pm 4.22\%$ | $29.17\% \pm 4.15\%$ | $40.83\% \pm 4.49\%$ | $31.67\% \pm 4.25\%$ |
| textworld | $32.55\% \pm 6.95\%$ | $8.24\% \pm 2.25\%$ | $4.67\% \pm 1.50\%$ | $6.08\% \pm 1.70\%$ |
| crafter | $29.09\% \pm 4.51$ | $33.64\% \pm 4.64$ | $31.36 \% \pm 3.55 \%$ | $20.00\% \pm 1.72\%$ |
| minihack | $12.50\% \pm 5.23\%$ | $5.00\% \pm 3.45\%$ | $12.50\% \pm 5.23\%$ | $12.50\% \pm 5.23\%$ |
| nle | $0.00\% \pm 0.00\%$ | $0.37\% \pm 0.33\%$ | $0.68\% \pm 0.37\%$ | $0.00\% \pm 0.00\%$ |

Table 8: Initial population for Gemini-2.0-Flash genetic algorithm.

| Environment | Relative Improvement (↑) |
|---|---|
| **Average** | $\mathbf{25.27\% \pm 3.05\%}$ |
| TAME[Run 1] | 22.24% |
| TAME[Run 2] | 24.00% |
| TAME[Run 3] | 29.47% |

Table 9: Relative percentage improvement of TAME over the best-performing initial population scores across environments using Gemini-2.0-Flash. *Note: The values indicate the percentage progression relative to the best performing initial genome, not the raw scores themselves.*

### K.5   CONVERGENCE PROPERTIES OF GENETIC ALGORITHM

We also show in Figure 19 the graphical representation of average game progression (% completion) over generations in genetic algorithm with Gemini-2.0-Flash. In each generation we consider 5 top scoring genomes. We present average and standard deviation of the score across all 6 games over 3 runs, showcasing that there is a clear improvement in terms of average progress, which shows promise for convergence when algorithm is run for longer.

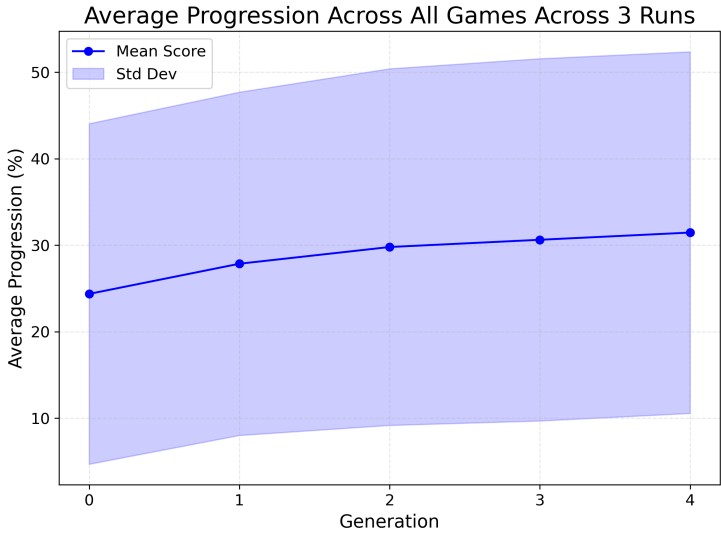

Figure 19: Average progression across all games (3 runs) with ± standard deviation. *Note: Values are absolute scores, not relative improvements.*

### K.5.1   CONVERGENCE PROPERTIES ACROSS GAMES

Figure 20 expands on Figure 19, presenting the convergence patterns accross all games. We measure the mean and standard deviation of the top 5 scoring genomes in the population at each generation. The results show a clear convergence trend: the standard deviation decreases in all environments

except for MiniHack. This demonstrates that increasing the computational budget enables convergence. Note that due to the scale of the y-axis, NetHack's standard deviation appears negligible in the figure. However, Table 1 confirms that measurable variance exists and that TAME continues to show improvement in this environment.

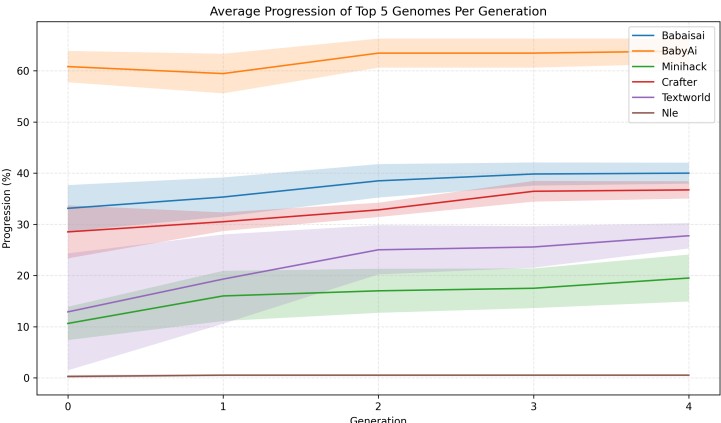

Figure 20: Average progression of top 5 scoring genomes in each generation over environments using Gemini-2.0-Flash. *Note: Values are absolute scores, not relative improvements*.

### K.6 GENOME TRANSFERABILITY TO OTHER GEMINI MODELS

In this Appendix we detail transerability of genomes obtained through genetic algorithm using Gemini-2.0-Flash, to Gemini-2.5-Flash-Lite and Gemini-2.5-Pro without additional training. In the Table 10 we compare performance of Baseline using Gemini-2.5-Flash-Lite vs TAME transferred to the same model. In the Table 11 we compare performance of Baseline using Gemini-2.5-Pro vs TAME transferred to the same model. Lastly, we show combined results in the Figure 21. The Transferred results are averaged over 3 independent runs of genetic algorithm.

| Environment | Baseline [2.5-Flash-Lite] | TAME[Transferred] | Episodes |
|---|---|---|---|
| **Average** | **11.87% $\pm$ 1.32%** | **20.48% $\pm$ 0.91%** | - |
| babyai | 46.00% $\pm$ 7.05% | **58.00% $\pm$ 3.06%** | 50 |
| babaisai | 9.17% $\pm$ 2.63 | **38.33% $\pm$ 0.48%** | 120 |
| textworld | **7.45% $\pm$ 2.30%** | **7.45% $\pm$ 0.00%** | 30 |
| crafter | 8.64% $\pm$ 1.00% | **13.01% $\pm$ 4.15%** | 10 |
| minihack | 0.00% $\pm$ 0.00% | **5.83% $\pm$ 1.67%** | 40 |
| nle | 0.00% $\pm$ 0.00% | **0.27% $\pm$ 0.15%** | 5 |

Table 10: Comparison of Baseline vs. TAME[Transferred] using Gemini-2.5-Flash-Lite. TAME[Transferred] averaged across 3 genetic algorithm runs.

#### K.6.1 TRANSFER SCORES ACROSS INDIVIDUAL GENETIC RUNS

In the Table 12 we present detailed results of genetic algorithm transferability to Gemini-2.5-Pro. Similarly, in Table 13 we show results of genetic algorithm transferability to Gemini-2.5-Flash-Lite across 3 independent genetic runs.

## L DETAILS OF GPT EXPERIMENTS

In this section we aim to test if we can get similar improvement when applying TAME to another family of Language Models. We test GPT4.1-nano, as it provides a fast inference, and we also test transferability to GPT4.1-mini. We run genetic algorithm 3 times and average results.

| Game | Baseline [2.5-Pro] | Top Model | TAME[Transferred] | Episodes |
|---|---|---|---|---|
| **Average** | **43.35% ± 2.31%** | **43.60% ± 2.17%** | **47.57% ± 2.72%** | - |
| babyai | 80.0% ± 5.70% | 76.00% ± 6.00% | **92.00% ± 3.46%** | 50 |
| babaisai | 56.70% ± 4.50% | 45.80% ± 4.50% | **67.50% ± 5.00%** | 120 |
| textworld | 49.20% ± 8.20% | **62.90% ± 7.90%** | 52.73% ± 6.12% | 30 |
| crafter | 55.0% ± 6.0% | **57.30% ± 3.90%** | 55.00% ± 0.00% | 10 |
| minihack | 17.50% ± 6.00% | 17.5% ± 6.00% | **18.33% ± 1.44%** | 40 |
| nle | 1.70% ± 0.20% | **1.80% ± 0.8%** | 1.70% ± 0.00% | 5 |

Table 11: Comparison of Baseline vs. TAME[Transferred] using Gemini-2.5-Pro. TAME[Transferred] averaged across 3 genetic algorithm runs.

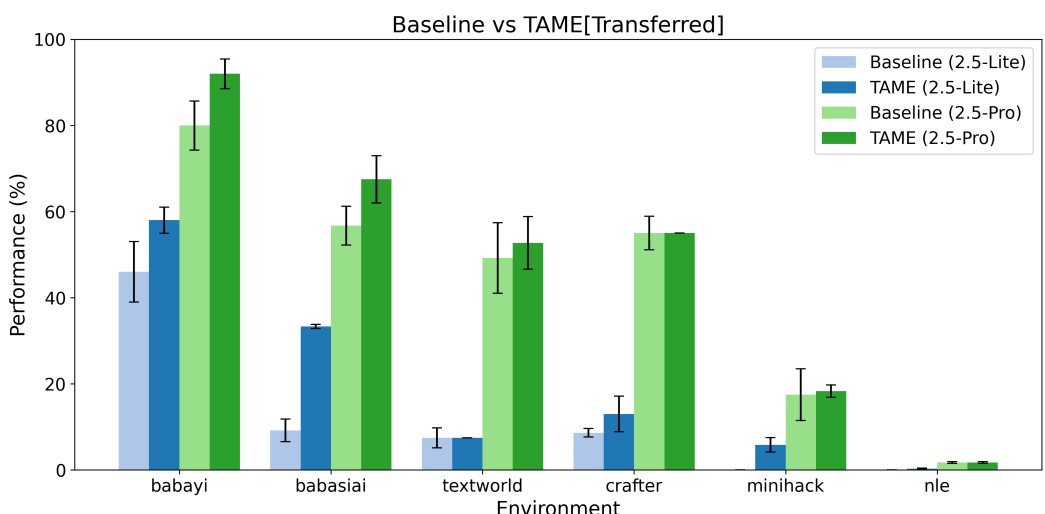

Figure 21: Evaluation of TAME Structure Transferability: Identical performance to the baseline indicates that the genetic algorithm favored the baseline architecture over the transferred TAME genome.

## L.1 TAME RESULTS OVER GPT MODEL FAMILY

We repeat the same procedure as described in section 3.2 applied to the initial population described in Section L.2. Table 14 shows the final TAME result, illustrating the relative gain of 74% when comparing average result over Baseline (see Table 15 for detailed results from independent runs). Additionally, we include results of applying the "Improved Prompt" proposed by Claude-Sonnet-4.5 (see Appendix O ). We conclude that TAME outperforms both baseline and ImprovedPrompt results. Those results demonstrate that TAME architecture are transferrable on more than onefamily of Language Models.

## L.2 INITIAL POPULATION OF GPT FAMILY MODELS

Table 16 shows initial population for GPT4.1-nano genetic algorithm. It follows the same structure as Gemini experiment and each component is described more in detail in the Appendix Q.2.

## L.3 IMPROVEMENT OVER INITIAL POPULATION

Table 17 shows the relative improvement when TAME is contrasted with best scoring genome from initial population described in Appendix L.2. The results demonstrate average relative gain of around 37.10%, showing the effectiveness of genetic algorithm over human-crafted inital population. Note that since NetHack baseline performance is 0%, even though we notice an improvement

| Environment | Baseline[2.5-Pro] | TAME[Run1] | TAME[Run2] | TAME[Run3] | Episodes |
|---|---|---|---|---|---|
| **Average** | $43.35\% \pm 2.31\%$ | $47.65\% \pm 2.20\%$ | $47.82\% \pm 2.14\%$ | $48.17\% \pm 2.25\%$ | |
| babyai | $80.00 \pm 5.70\%$ | $90.00 \pm 4.24$ | $96.0 \pm 2.77\%$ | $90\% \pm 4.24\%$ | 50 |
| babaisai | $56.70\% \pm 4.50\%$ | $72.50\% \pm 4.08\%$ | $67.5\% \pm 4.28\%$ | $62.5\% \pm 4.42\%$ | 120 |
| textworld | $49.20\% \pm 8.20\%$ | $49.20\% \pm 8.20\%$ | $49.20\% \pm 8.20\%$ | $59.80\% \pm 8.34\%$ | 30 |
| crafter | $55.0\% \pm 6.00\%$ | $55.00\% \pm 6.00\%$ | $55.00\% \pm 6.00\%$ | $55.00\% \pm 6.00\%$ | 10 |
| minihack | $17.50\% \pm 6.00\%$ | $17.50\% \pm 6.00\%$ | $17.50\% \pm 6.00\%$ | $20.00\% \pm 6.32\%$ | 40 |
| nle | $1.70\% \pm 0.20\%$ | $1.70\% \pm 0.20\%$ | $1.70\% \pm 0.20\%$ | $1.70\% \pm 0.20\%$ | 5 |

Table 12: Details of independent runs of transferability to Gemini-2.5-Pro

| Environment | Baseline[2.5-Flash-Lite] | TAME[Run1] | TAME[Run2] | TAME[Run3] | Episodes |
|---|---|---|---|---|---|
| **Average** | $11.88\% \pm 1.32\%$ | $19.25\% \pm 1.63\%$ | $19.43\% \pm 1.49\%$ | $22.76\% \pm 1.73\%$ | |
| babyai | $46.00\% \pm 7.05\%$ | $52.00\% \pm 7.07\%$ | $60.00\% \pm 6.93\%$ | $62.00\% \pm 6.93\%$ | 50 |
| babaisai | $9.17\% \pm 2.63$ | $39.17\% \pm 4.46\%$ | $37.50\% \pm 4.42\%$ | $38.33\% \pm 4.44\%$ | 120 |
| textworld | $7.45\% \pm 2.30\%$ | $7.45\% \pm 2.30\%$ | $7.45\% \pm 2.30\%$ | $7.45\% \pm 2.30\%$ | 30 |
| crafter | $8.64\% \pm 1.00\%$ | $9.09\% \pm 1.70\%$ | $8.64\% \pm 1.00\%$ | $21.3\% \pm 4.22\%$ | 10 |
| minihack | $0.00\% \pm 0.00\%$ | $7.5\% \pm 4.16\%$ | $2.5 \% \pm 2.47$ | $7.50\% \pm 4.16\%$ | 40 |
| nle | $0.00\% \pm 0.00\%$ | $0.31\% \pm 0.28\%$ | $0.51\% \pm 0.42\%$ | $0.00\% \pm 0.00\%$ | 5 |

Table 13: Details of independent runs of transferability to Gemini-2.5-Flash-Lite

| Environment | Baseline ($\uparrow$) | ImprovedPrompt[Claude] ($\uparrow$) | TAME ($\uparrow$) | Episodes |
|---|---|---|---|---|
| Average | $9.90\% \pm 1.33\%$ | $6.10\% \pm 1.12\%$ | $\mathbf{17.20\% \pm 1.47\%}$ | - |
| babyai | $32.0\% \pm 6.60\%$ | $20.00\% \pm 5.66\%$ | $\mathbf{48.67\% \pm 7.07\%}$ | 50 |
| babaisai | $12.5\% \pm 3.02\%$ | $6.67\% \pm 2.28\%$ | $\mathbf{21.55\% \pm 3.76\%}$ | 120 |
| textworld | $0.59\% \pm 0.58\%$ | $0.39\% \pm 0.39\%$ | $\mathbf{2.88\% \pm 0.94\%}$ | 30 |
| crafter | $11.82\% \pm 2.15\%$ | $7.07\% \pm 1.26\%$ | $\mathbf{19.78\% \pm 2.33\%}$ | 10 |
| minihack | $2.5\% \pm 2.47\%$ | $2.50\% \pm 2.47\%$ | $\mathbf{10.00\% \pm 2.78\%}$ | 40 |
| nle | $0.0\% \pm 0.0\%$ | $0.00\% \pm 0.00\%$ | $\mathbf{0.22\% \pm 0.20\%}$ | 5 |

Table 14: Baseline vs. ImprovedPrompt[Claude] vs. TAME using GPT4.1-nano averaged over 3 runs.

after applying TAME, we can't measure its relative improvement, therefore we exclude it from calculation in Table 17.

## L.4 TRANSFERABILITY ACROSS GPT FAMILY MODELS

In this section we test if results can transfer within GPT family. We follow the same procedure as in Gemini case. Therefore, for each environment we evaluate the best scoring genome (obtained from TAME run with GPT4.1-nano) using GPT4.1-mini. We apply the same procedure for all independent genetic algorithm runs and average results. Table 18 shows the results, demonstrating a relative gain of around 10%, when compared to Baseline GPT4.1-mini results (see Table 19 for more details across independent runs) .

## M TAME CHILD EVALUATION PSEUDO CODE

This Appendix presents the pseudo code behind TAME evaluation on specific game when all modules are active, as decided by the genetic algorithm (overview in Figure 2). The algorithm works as follows: until maximum number of steps is reached, with probability $\epsilon$ choose exploration, otherwise exploitation. For each option, it retrieves top successful and failed memories, then the Low-level Executor decides and executes actions. If danger is detected, the survival module is triggered and execution stops. Otherwise, the Critic evaluates success, summarizes key actions, and in case of loops, the summary is replaced by the Loop Detector summary. The resulting memory is stored, and the cycle repeats.

| Environment | Baseline (↑) | TAME[Run1] (↑) | TAME[Run2] (↑) | TAME[Run3] (↑) |
|---|---|---|---|---|
| **Average** | 9.90% ± 1.33% | 18.06% ± 1.65% | 17.19% ± 1.60% | 16.31% ± 1.58% |
| babyai | 32.00% ± 6.60% | 52.00% ± 7.07% | 48.00% ± 7.07% | 46.00% ± 7.06% |
| babaisai | 12.50% ± 3.02% | 22.5% ± 3.81% | 22.5% ± 3.81% | 19.66% ± 3.67% |
| textworld | 0.59% ± 0.58% | 3.13% ± 0.95% | 2.75% ± 0.93% | 2.75% ± 0.93% |
| crafter | 11.82% ± 2.15% | 20.71% ± 3.04% | 19.55% ± 2.14% | 19.09% ± 1.80% |
| minihack | 2.5% ± 2.47% | 10.00% ± 4.86% | 10.00% ± 4.74% | 10.00% ± 4.74% |
| nle | 0.0% ± 0.0% | 0.0% ± 0.0% | 0.31% ± 0.28% | 0.38% ± 0.33% |

Table 15: Detailed scores across 3 genetic algorithm runs with GPT4.1-nano.

| Environment | Baseline (↑) | Hierarchical[Hand-Crafted] (↑) | Full[Claude] (↑) | Full[Hand-Crafted] (↑) |
|---|---|---|---|---|
| **Average** | 9.90% ± 1.33% | 10.40% ± 2.07% | 10.95% ± 2.77% | 9.28% ± 2.15% |
| babyai | 32.0% ± 6.60% | 12.00% ± 4.60% | 22.00% ± 5.86% | 10.00% ± 4.24% |
| babaisai | 12.50% ± 3.02% | 16.67% ± 3.40% | 16.10% ± 3.38% | 19.66% ± 3.67% |
| textworld | 0.59% ± 0.58% | 1.18% ± 0.60% | 2.75% ± 0.93% | 1.37% ± 0.70% |
| crafter | 11.82% ± 2.15% | 7.58% ± 1.01% | 15.00% ± 1.71% | 12.27% ± 1.82% |
| minihack | 2.50% ± 2.47% | 5.00% ± 3.45% | 0.00% ± 0.00% | 2.50% ± 2.47% |
| nle | 0.00% ± 0.00% | 0.00% ± 0.00% | 0.00% ± 0.00% | 0.00% ± 0.00% |

Table 16: Initial population for GPT-4.1-nano genetic algorithm.

# N    DETAILED MEMORY ABLATIONS

In this section we show detailed results when comparing our memory system with Craft-Jarvis-1 and A-Mem using Gemini-2.0-Flash. Table 20 shows that TAME-Memory consistently outperforms both Craft-Jarvis-1 and A-Mem on most environments, yielding the highest overall average of 23.11%. The gains are particularly strong in TextWorld and Crafter, where our system nearly doubles or surpasses the baselines by a large margin. While performance is comparable in Minihack and NLE, these tasks remain challenging for all methods. Overall, the results highlight the efficiency of our hybrid memory system.

# O    'IMPROVED PROMPT' PROPOSED BY CLAUDE

Here we present a prompt proposed by Claude-Sonnet-4.5, when tasked to improve the Baseline (BALROG) prompt. As well as the baseline prompt, we gave information about functional inputs (see Section 3.1) and gave it an option to use them.

---

**Improved Prompt proposed by Claude**

```
# Game Agent Instructions

You are an autonomous agent navigating a game environment. Your task is to select the optimal next
    ↪ action based on the current state.

## Current State
- **Observation**: {obs}
- **Current Subgoal**: {subgoal}
- **Success Condition**: {success_condition}

## Context
- **Recent History**: {history}
- **Action Sequence (Current Subgoal)**: {action_sequence}

## Knowledge Base
- **Similar Successful Attempts**: {entries_successful_subgoal}
- **Similar Failed Attempts**: {entries_failed_subgoal}

## Available Actions
{game_info}

## Output Format
```

| Environment | Relative Improvement ($\uparrow$) |
|---|---|
| **Average** | **37.10% $\pm$ 2.08%** |
| TAME[Run 1] | 38.14% |
| TAME[Run 2] | 38.96% |
| TAME[Run 3] | 34.20% |

Table 17: Relative percentage improvement of TAME over the best-performing initial population scores across environments using GPT4.1-nano averaged over 3 genetic algorithm runs. *Note: The values indicate the percentage progression relative to the best performing initial genome, not the raw scores themselves.*

| Environment | Baseline[GPT-4.1-mini] | TAME[Transferred] | Episodes |
|---|---|---|---|
| Average | 24.43% $\pm$ 1.89% | **26.80% $\pm$ 1.92%** | - |
| babyai | **72.00% $\pm$ 7.18%** | **72.00% $\pm$ 7.18%** | 50 |
| babaisai | 29.17% $\pm$ 4.15% | **35.00% $\pm$ 4.33%** | 120 |
| textworld | **13.33% $\pm$ 5.74%** | **13.33% $\pm$ 5.74%** | 30 |
| crafter | 21.00% $\pm$ 2.00% | **29.43% $\pm$ 2.65%** | 10 |
| minihack | **10.00% $\pm$ 4.74%** | **10.00% $\pm$ 4.74%** | 40 |
| nle | **1.09% $\pm$ 0.41%** | **1.09% $\pm$ 0.41%** | 5 |

Table 18: Transferability of TAME across GPT family averaged over 3 runs.

```
Output ONLY a single valid action from the available actions list. No explanations, no additional
    ↪ text.

## Decision Strategy
1. Analyze current observation against the success condition
2. Learn from similar past attempts (successful and failed)
3. Consider recent action history to avoid loops
4. Select the action most likely to progress toward the subgoal
Output in the following format: \n
        REASONING: <your reasoning> \n
        ACTION: <your action>
```

# P  LLM MUTATION AND CROSSOVER PROMPTS

This Appendix details the prompts used for LLM based Crossover (in the case of single parent) and LLM based Crossover and Mutation (in the case of two parents) used in genetic algorithm. Prompts created with the help of Claude-Sonnet-4.

**LLM Mutation Prompt**

```
"""
Please follow the instruction step-by-step to generate a better prompt.

1. Consider prompt:
Prompt 1: <prompt1>

2. Apply ONE of the following mutation strategies to create a significantly different prompt:

**Strategy A - Perspective Shift:** Change the role/perspective (e.g., "As an expert analyst..." or
    ↪ "From the viewpoint of...")

**Strategy B - Methodology Change:** Alter the approach (step-by-step -> holistic analysis, direct
    ↪ -> comparative, etc.)

**Strategy C - Output Format Transformation:** Change how results are presented (narrative ->
    ↪ structured, single response -> multi-part, etc.)

**Strategy D - Contextual Enhancement:** Add specific domain knowledge or constraints that weren't
    ↪ in the original
```

| Environment | Baseline[GPT4.1-mini] | TAME[Run1] | TAME[Run2] | TAME[Run3] | Episodes |
|---|---|---|---|---|---|
| Average | 24.43% ± 4.04% | 28.33% ± 4.17% | 25.40% ± 1.09% | 25.44% ± 1.92% | - |
| babyai | 72.00% ± 7.18% | 72.00% ± 7.18% | 72.00% ± 7.18% | 72.00% ± 7.18% | 50 |
| babaisai | 29.17% ± 4.15% | 40.83% ± 4.49 % | 35.00% ± 4.35% | 29.17% ± 4.15% | 120 |
| textworld | 13.33% ± 5.74% | 13.33% ± 5.74% | 13.33% ± 5.74% | 13.33% ± 5.74% | 30 |
| crafter | 21.00% ± 2.00% | 32.73% ± 2.47% | 21.00% ± 2.00% | 34.55% ± 3.47% | 10 |
| minihack | 10.00% ± 4.74% | 10.00% ± 4.74% | 10.00% ± 4.74% | 10.00% ± 4.74% | 40 |
| nle | 1.09% ± 0.41% | 1.09% ± 0.41% | 1.09% ± 0.41% | 1.09% ± 0.41% | 5 |

Table 19: Details of independent runs of transferability to GPT4.1-mini

---

**Algorithm 2** TAME [Full Structure]

---

**Require:** $I, s_0$
**Ensure:** $s_T$ (final game state)
  **while** $num\_steps \leq max\_steps$ **do**
    **if** $rand() < \epsilon$ **then**
      $g \leftarrow \pi_{explorer}(s_i^e)$ {Exploration}
    **else**
      $g \leftarrow \pi_{high}(s_i^{mc})$ {Exploitation}
    **end if**
    **for** $g_i \in g$ **do**
      $M_i^+ \leftarrow retrieve\_successful(n_{g_i}, d_{g_i}, memory)$
      $M_i^- \leftarrow retrieve\_failed(n_{g_i}, d_{g_i}, memory)$
      $a \leftarrow \pi_{low}(s_i^{le})$
      **for** $a_i \in a$ **do**
        $s_t \leftarrow execute\_action(a, s_i)$
        $s_i \leftarrow s_t$
        **if** $\sigma(s_i)$ **then**
          $g \leftarrow g^*$ {Activate survival option}
          **break** {Move to option selection and force $g^*$}
        **end if**
      **end for**
      $C_i \leftarrow \varphi(s_i)$
      $h_i \leftarrow \rho(s_i, C_i)$
      **if** $C_i$ **then**
        $store\_memory(g_i, h_i, o, \text{``}successful\text{''})$
      **else**
        **if** $\psi(s_i)$ **then**
          $h_i \leftarrow \psi(s_i)$
        **end if**
        $store\_memory(g_i, h_i, o, \text{``}failed\text{''})$
        **break** {Escalate to replanning}
      **end if**
    **end for**
  **end while**=0

---

```
**Strategy E – Complexity Modulation:** Significantly increase or decrease the cognitive complexity
    ↪ of the task

**Strategy F – Functional Input Integration:** Incorporate functional inputs in a novel way that
    ↪ changes the prompt's core operation

3. Generate the final mutated prompt bracketed with <prompt> and </prompt>.

## Available Functional Inputs:
- {obs} : current observation
- {game_info} : information about the game (possible actions and goal)
- {subgoal} : current subgoal that you're working towards
- {action_sequence}: action sequence towards current subgoal
- {observation_sequence}: observation sequence towards current subgoal
- {success_condition}: termination condition of the current subgoal
- {action_obs_seq}: action-observation pairs towards current subgoal
- {survival_plan}: survival plan
- {history}: history of the last 16 action-observation pairs
- {entries_successful_goal}: most similar successful subgoals to the current one
- {entries_failed_goal}: most similar failed subgoals to the current one
```

| Environment | Jarvis (↑) | TAME-Memory[ours] (↑) | A-mem (↑) | Episodes |
|---|---|---|---|---|
| Average | 17.52% ± 1.73% | **23.11% ± 1.75%** | 21.45% ± 1.80% | - |
| babyai | 48.00 % ± 7.06 % | **62.00% ± 6.86%** | 58.00% ± 6.98% | 50 |
| babaisai | 24.17 % ± 3.90 % | **29.17% ± 4.15%** | 26.67% ± 4.04% | 120 |
| textworld | 0.59 % ± 0.58 % | **8.24% ± 1.95%** | 4.51% ± 2.35% | 30 |
| crafter | 19.55 % ± 3.81 % | **35.45% ± 3.20%** | 26.36% ± 4.25% | 10 |
| minihack | **12.5% ± 5.23%** | 3.45% ± 5.65% | **12.5% ± 5.23%** | 40 |
| nle | 0.31% ± 0.28% | 0.37% ± 0.33% | **0.68% ± 0.37%** | 5 |

Table 20: Comparison of TAME average game progression across different memory types using Gemini-2.0-Flash.

```
**CRITICAL:** Use functional inputs with exact bracket names. Ensure the mutation creates a
    ↪ substantially different prompt that would produce notably different outputs.
Output your asnwer in thne follwing way:
REASONING: <your reasoning>
PROMPT: <mutated prompt>
"""
```

**LLM Mutation and Crossover Prompt**

```
"""
Please follow the instruction step-by-step to generate a better prompt.

1. Crossover the following prompts and generate a new prompt:
Prompt 1: <prompt1>
Prompt 2: <prompt2>

2. Apply ONE of the following mutation strategies to create a significantly different prompt:

**Strategy A - Perspective Shift:** Change the role/perspective (e.g., "As an expert analyst..." or
    ↪ "From the viewpoint of...")

**Strategy B - Methodology Change:** Alter the approach (step-by-step -> holistic analysis, direct
    ↪ -> comparative, etc.)

**Strategy C - Output Format Transformation:** Change how results are presented (narrative ->
    ↪ structured, single response -> multi-part, etc.)

**Strategy D - Contextual Enhancement:** Add specific domain knowledge or constraints that weren't
    ↪ in the original

**Strategy E - Complexity Modulation:** Significantly increase or decrease the cognitive complexity
    ↪ of the task

**Strategy F - Functional Input Integration:** Incorporate functional inputs in a novel way that
    ↪ changes the prompt's core operation

3. Generate the final mutated prompt bracketed with <prompt> and </prompt>.

## Available Functional Inputs:
- {obs} : current observation
- {game_info} : information about the game (possible actions and goal)
- {subgoal} : current subgoal that you're working towards
- {action_sequence}: action sequence towards current subgoal
- {observation_sequence}: observation sequence towards current subgoal
- {success_condition}: termination condition of the current subgoal
- {action_obs_seq}: action-observation pairs towards current subgoal
- {survival_plan}: survival plan
- {history}: history of the last 16 action-observation pairs
- {entries_successful_goal}: most similar successful subgoals to the current one
- {entries_failed_goal}: most similar failed subgoals to the current one

**CRITICAL:** Use functional inputs with exact bracket names. Ensure the mutation creates a
    ↪ substantially different prompt that would produce notably different outputs than the
    ↪ crossover result.
Output your asnwer in thne follwing way:
REASONING: <your reasoning>
PROMPT: <mutated prompt>
"""
```

## Q   INITIAL POPULATION GENOMES

This Appendix details the initial population of genetic algorithm described in section 3.2. First, we show examples of prompts proposed by Claude-Sonnet-4 for each of the modules. We use those in order to construct Full[Claude] genome. For the rest of genomes we use hand-crafted prompts. We detail genomes in the subsections below.

### Q.1   PROMPTS PROPOSED BY SONNET-4

Those prompts were created using Claude-Sonnet-4, which was tasked with the creation of a prompt for each module. We call those "default" prompts

---

**Prompt Template**

```
default_sequential_prompt =  """You always have to output one of the above actions at a time and no
    ↪ other text. You always have to output an action until the episode terminates."""

default_highlevel_prompt = """
You are a strategic planner for a video game AI. Analyze the current game state and create
    ↪ achievable subgoals that advance toward the main objective.

REQUIREMENTS:
- Subgoals must be immediately achievable with current capabilities
- Focus on next logical steps, not distant goals
- Each subgoal should have clear success criteria

CURRENT STATE: {obs}
GAME INFO: {game_info}

Create a sequential plan with 2-3 subgoals."""
default_lowlevel_prompt = """  You are an action executor in a video game AI system. Given a subgoal
    ↪ from the high-level planner, propose a sequence of actions to achieve it.
        ####################################################
        CURRENT SUBGOAL: {subgoal}
        CURRENT STATE: {obs}
        PREVIOUS ACTIONS: {action_sequence}

        Plan the full sequence of actions needed to complete the subgoal. Avoid repeating actions if
    ↪ observations don't change.
        Avoid extra commentary outside the REASONING and ACTIONS list.
        """
default_termination_prompt = """
            You are a termination evaluator for a video game AI. Check if the agent has completed
    ↪ its subgoal.

            CURRENT STATE: {obs}
            SUBGOAL: {subgoal}
            SUCCESS CONDITION: {success_condition}
            RECENT ACTIONS: {action_sequence}

            Compare the current state with the success condition to determine if the subgoal is
    ↪ complete. Provide feedback to help the agent improve.
"""

default_summariser_prompt = """
        You are a critic analyzing an agent's subgoal attempt. Identify the key factor that caused
    ↪ success or failure.

        SUBGOAL: {subgoal}
        OUTCOME: {outcome}
        ACTION HISTORY: {action_obs_seq}

        Focus on specific resources and quantities that mattered most. If no resources involved,
    ↪ identify the next most important factor.
        """
default_amygdala_prompt = """
        Decide if survival mode should activate.

        Observation: {obs}
        Survival plan: {survival_plan}

        1. Check if observation meets any subtask prerequisites.
        2. If several match, pick highest priority.
"""
default_loop_prompt = """
        Task: Decide if the agent is stuck in a loop.

        Loop = repeating actions without meaningful progress toward the subgoal
        (progress = closer to goal, new info, removing failed paths, or advancing game state).

        Data:
        - Observation: {obs}
```

---

```
              - Subgoal: {subgoal}
              - Termination condition: {success_condition}
              - Action-observation history: {history}

              Steps:
              1. Check if enough steps have been taken to allow exploration.
              2. Look for repeated patterns without progress.
              3. If loop detected, identify cause: missing info, unknown prerequisite, or unexplored path.
              """
default_explorer_prompt = """
              Task: Create an exploration plan to help the agent discover new skills.

              Data:
              - Game info: {game_info}
              - Observation: {obs}
              - Subgoal summary: {summary}
              - Recent 16 action-observation pairs: {history}

              Steps:
              1. Analyze the environment and agent's situation.
              2. Propose a focused exploration plan with clear purpose, conditions, and indicators.

          """
```

### Q.2    INITIAL POPULATION

Detailed genome descriptions for the initial population.

**Baseline genome**    This genome corresponds to the baseline structure used in BALROG.

**Baseline Genome**

```
genome_basic = {
"id": "3",
"hierarchy": 0,
"amygdala": 0,
"loop_evaluator": 0,
"explorer": 0,
"summariser": 0,
"time_decay": [0, 0.01],
"cosine_cutoff": [0, 0.05],
"epsilon": [0, 0.01],
"temperature": 1.0,
"prompts": {
    "high_level": [0, default_highlevel_prompt],
    "low_level": [0, default_lowlevel_prompt],
    "amygdala": [0, default_amygdala_prompt],
    "loop": [0, default_loop_prompt],
    "termination": [0, default_termination_prompt],
    "summariser": [0, default_summariser_prompt],
    "explorer": [0, default_explorer_prompt],
    "sequential": [1, default_sequential_prompt]
},
"fitness": ...
}
```

**Hierarchical[Hand-Crafted] genome**    This genome corresponds to hierarchy from Section 3.3.1 combined with memory 3.4, without amygdala and explorer. We use this architecture as one of the initial seeds, because we found that it gives the most stable performance across all environments.

**Hierarchical[Hand-Crafted] Genome**

```
genome_stable = {
"id": "2",
"hierarchy": 1,
"amygdala": 0,
"loop_evaluator": 1,
"explorer": 0,
"summariser": 1,
"time_decay": [1, 0.01],
"cosine_cutoff": [0, 0.1],
"epsilon": [0, 0.01],
"temperature": 1.0,
"prompts": {
    "high_level": [1,
```

```
"""You are a strategic planner in a hierarchical video game AI system. Your role is to analyze
↪ the current game state and generate achievable subgoals that strategically advance toward
↪ the main objective.

    CRITICAL CONSTRAINTS:
    - Subgoals must be immediately achievable given the current agent state and capabilities
    - Focus on the next logical progression steps, not distant end-goals
    - Each subgoal should have clear, observable success criteria
    - Subgoals should be novel and interesting, compared to previous attempts

    Below I will provide the game description, available actions, and current state information.
    Game description:
    {game_info}
    ########################################################
    CURRENT CONTEXT: \n
    - Game State: {obs} \n
    - Survival plan provided by the survival planner, that you should consider for your tasks:
↪ {survival_plan}
    - Here are most similar successful entries from the archive: {entries_successful_goal} \n
    - Here are most similar failed entries from the archive: {entries_failed_goal} \n

    ANALYSIS FRAMEWORK:
    1. Analyse the summary from previous runs and let it guide your decision making.
    2. Assess what is immediately possible given current agent state and environment
    3. Identify what kind of actions could be considered novel or interesting
    4. Identify the most direct path toward the main objective
    5. Select subgoals that form a logical sequence
    6. Ensure each subgoal can be verified through observable game state changes

    SUBGOAL SELECTION CRITERIA:
    - Feasible: Can be started immediately with current resources/position
    - Measurable: Success/failure can be determined from game observations
    - Progressive: Each subgoal enables the next or advances toward main goal
    - Specific: Clear enough for a lower-level agent to understand and execute
    - Considerate of the summary of previos runs

    Make sure that your subgoals are sequential. """],

"low_level": [1,
"""You are an important executor component of a hierarchical video game system. You are given
↪ one of higher level option and its termination condition proposed by the higher level
↪ planner. Your role is to propose a sequence of actions that will make you progress towards
↪ the given option." \
    Below I will provide you with the game description, possible actions you can take and the
↪ overall goal of the game.

    ####################################################
    Here is a subgoal provided by the high level planner that you should focus on completing:
↪ {subgoal} \n
    Here is your current state: {obs} \n
    Here is the action-observation sequence towards current subgoal: {action_sequence} \n
    Here are the most similar successful entries from the archive: {entries_successful_goal} \n
    Here are the most similar failed entries from the archive: {entries_failed_goal} \n
    Use the action and observation sequence together with the current state to decide the **full
↪ ordered sequence of actions** that will achieve the subgoal. \n
    Avoid repeating the same actions if the observation doesn't change. \n"""],

"amygdala": [0, """
    You are an important component in a hierarchical video game system. Your role is to
↪ determine if the agent is in danger and should activate survival mode. Below I will provide
↪ you with current observation and a survival plan from the higher level agent. \n

    Current observation: {obs} \n
    Survival plan: {survival_plan} \n

    Your role is to analyse the observation and survival plan given by higher level system and
↪ determine if the current observation satisfies any of the prerequites for any of the
↪ survival components. If there are prerequisites satisfied for multiple components, then
↪ return the one with the highest priority. \n
    First reason, then output True or False depending if you decide to activate survival plan.
↪ If you output True, then output one of the survival subtasks. If you decide to not activate
↪ survival plan, then output None as the survival subtask. \n

    REASONING: <your reasoning> \n
    ACTIVATE SURVIVAL: <True/False> \n
    SURVIVAL SUBTASK: <survival subtask name or None if False> \n

    """],
"loop": [1,
"""You are an important loop evaluator component of a hierarchical video game system.
You are going to receive details about game progress such as: current observation, current
↪ subgoal, current termination conditions, action sequence towards current subgoal and
↪ observation sequence towards current subgoal.
Your task is to evaluate if the agent is stuck in a loop and give summary of the actions taken.
A loop occurs when the agent repeats a sequence of actions multiple times without achieving
↪ meaningful progress toward its current goal, where 'meaningful progress' includes: getting
↪ closer to the objective, discovering new information, eliminating failed approaches, or
↪ changing the game state in a way that advances toward the subgoal.
It is important that you let the agent explore enough but also decide when to terminate to get
↪ out of the loop. \n
```

```
       Details: \n
       Here is your current state: {obs} \n
       Here is a subgoal lower level agent is working towards: {subgoal} \n
       Here is the most recent action-observation pairs that should help you decide if agent is stuck
         ↪ in a loop: {history}\n \n

       Instructions:\n
       Analyse the details. Avoid giving any judgement. \n
       Think about how many steps the agent needs in order to complete the subgoal and use that to help
         ↪ you reason if agent is stuck in a loop. \n
       Then, given your analysis, decide if the actions proposed by the lower level agent are leading
         ↪ to the termination condition or if the agent is stuck in a loop.
       If a loop is detected, analyse if the agent is stuck due to a lack of necessary information, an
         ↪ unknown prerequisite, or an unexplored path. Your summary should clearly articulate this gap
         ↪ in knowledge and suggest that exploration might be required to break the loop and find
         ↪ alternative solutions. \n"""],
    "termination": [1, """
       You are an important termination evaluator component of a hierarchical video game system. \n
       Your task is to: \n
       1. Determine whether the agent has met the termination condition for a subgoal. \n
       2. Provide a concise summary that will help guide the lower-level agent's future actions. \n\n

       Details: \n
       Here is your currect state that you should compare with termination condition: {obs} \n
       Here is the subgoal lower level agent is working towards: {subgoal} \n
       Here is the termination condition of the above subgoal given by the higher level agent:
         ↪ {success_condition} \n \n

       Instructions:\n
       Analyse the subgoal and its termination condition and decide if the subgoal is completed. \n
       Then use action-observation sequence: {action_sequence} to give a high level summary of current
         ↪ evaluation. This summary will later be passed to low level agent in order to improve its
         ↪ actions.
       Remember that your summary will be passed to low level component in order to improve its
         ↪ actions.  \n \n"""],
    "summariser": [1,
       """You are a critic module analyzing an agent's attempt to achieve a subgoal in a game
         ↪ environment.
       Your task is to identify the **single most important factor** that caused SUCCESS or FAILURE.

       Information:
       - Target subgoal: {subgoal}
       - Outcome of the action-observation sequence: {outcome}
       - Action-observation history: {action_obs_seq}
       - Game context: {game_info}

       Instructions:
       - Think briefly about what helped or prevented success.
       - Focus mostly on **specific resources and their quantities** (e.g., "3 pieces of wood", "1 iron
         ↪ ingot").
       - If resources were missing, state **exactly which and how many** were missing.
       - Ignore minor details or redundant actions.
       - Express the result in **one short sentence**.
       - If no resources are involved, state the next most relevant factor."""],
    "explorer": [0, default_explorer_prompt],
    "sequential": [0, default_sequential_prompt]
   },
   "fitness": ...
   }
```

**Full[Claude] genome** Default genome consits of all modules being active and prompts proposed by Claude-Sonnet-4. We add this genome to initial population, as they remove human bias.

### Full[Claude] Genome

```
genome_default = {
"id": "1",
"hierarchy": 1,
"amygdala": 1,
"loop_evaluator": 1,
"explorer": 1,
"summariser": 1,
"time_decay": [1, 0.01],      # [active_flag, value]
"cosine_cutoff": [1, 0.05],
"epsilon": [1, 0.1],
"temperature": 1.0,           # continuous
"prompts": {                  # [active_flag, text]
    "high_level": [1, default_highlevel_prompt],
    "low_level": [1, default_lowlevel_prompt],
    "amygdala": [1, default_amygdala_prompt],
    "loop": [1, default_loop_prompt],
    "termination": [1, default_termination_prompt],
```

```
2322
2323        "summariser": [1, default_summariser_prompt],
2324        "explorer": [1, default_explorer_prompt],
            "sequential": [0, default_sequential_prompt]
2325    },
        "fitness": ..
2326    }
2327
2328
```

2329 **Full[Hand-Crafted]**   This genome represents a full structure with all modules active and hand-
2330 crafted prompts.
2331

### Full[Hand-Crafted] Genome

```
2332
2333
2334
2335    genome_full = {
            "id": "1",
2336        "hierarchy": 1,
            "amygdala": 1,
2337        "loop_evaluator": 1,
            "explorer": 1,
2338        "summariser": 1,
            "time_decay": [1, 0.01],
2339        "cosine_cutoff": [1, 0.05],
            "epsilon": [1, 0.1],
2340        "temperature": 1.0,
            "prompts": {
2341        "high_level": [1, """You are a strategic planner in a hierarchical video game AI system. Your
2342            ↪ role is to analyze the current game state and generate achievable subgoals that
                ↪ strategically advance toward the main objective.
2343        CRITICAL CONSTRAINTS:
            - Subgoals must be immediately achievable given the current agent state and capabilities
2344        - Focus on the next logical progression steps, not distant end-goals
            - Each subgoal should have clear, observable success criteria
2345        - Subgoals should be novel and interesting, compared to previous attempts
2346
            Below I will provide the game description, available actions, and current state information.
2347        Game description:
            {game_info}
2348        ########################################################
            CURRENT CONTEXT: \n
2349        - Game State: {obs} \n
            - Summary from the previous high level plan : {summary} \n
2350        - Survival plan provided by the survival planner, that you should consider for your tasks:
              ↪ {survival_plan}
2351        - Here are most similar successful entries from the archive: {entries_successful_goal} \n
            - Here are most similar failed entries from the archive: {entries_failed_goal} \n
2352
            ANALYSIS FRAMEWORK:
2353        1. Analyse the summary from previous runs and let it guide your decision making.
            2. Assess what is immediately possible given current agent state and environment
2354        3. Identify what kind of actions could be considered novel or interesting
            4. Identify the most direct path toward the main objective
2355        5. Select subgoals that form a logical sequence
            6. Ensure each subgoal can be verified through observable game state changes
2356
            SUBGOAL SELECTION CRITERIA:
2357        - Feasible: Can be started immediately with current resources/position
            - Measurable: Success/failure can be determined from game observations
2358        - Progressive: Each subgoal enables the next or advances toward main goal
            - Specific: Clear enough for a lower-level agent to understand and execute
2359        - Considerate of the summary of previos runs
2360
            Make sure that your subgoals are sequential. """],
2361        "low_level": [1, """You are an important executor component of a hierarchical video game system.
              ↪ You are given one of higher level option and its termination condition proposed by the
2362          ↪ higher level planner. Your role is to propose an action that will make you progress towards
              ↪ the given option." \
2363        Below I will provide you with the game description, possible actions you can take and the
              ↪ overall goal of the game.
2364
            ####################################################
2365        Here is a subgoal provided by the high level planner that you should focus on completing:
              ↪ {subgoal} \n
2366        Here is your current state: {obs} \n
            Here is the action-observation sequence towards current subgoal: {action_sequence} \n
2367        Here are the most similar successful entries from the archive: {entries_successful_goal} \n
            Here are the most similar failed entries from the archive: {entries_failed_goal} \n
2368        Use the action and observation sequence together with the current state to decide the **full
              ↪ ordered sequence of actions** that will achieve the subgoal. \n
2369        Avoid repeating the same actions if the observation doesn't change. \n"""],
            "amygdala": [1, """
2370        You are an important component in a hierarchical video game system. Your role is to determine if
              ↪ the agent is in danger and should activate survival mode. Below I will provide you with
2371          ↪ current observation and a survival plan from the higher level agent. \n
2372
2373
2374
2375
```

```
        Current observation: {obs} \n
        Survival plan: {survival_plan} \n

        Your role is to analyse the observation and survival plan given by higher level system and
        ↪ determine if the current observation satisfies any of the prerequites for any of the
        ↪ survival components. If there are prerequisites satisfied for multiple components, then
        ↪ return the one with the highest priority. \n
        First reason, then output True or False depending if you decide to activate survival plan. If
        ↪ you output True, then output one of the survival subtasks. If you decide to not activate
        ↪ survival plan, then output None as the survival subtask. \n

        REASONING: <your reasoning> \n
        ACTIVATE SURVIVAL: <True/False> \n
        SURVIVAL SUBTASK: <survival subtask name or None if False> \n

        """],
"loop": [1, """You are an important loop evaluator component of a hierarchical video game system.
You are going to receive details about game progress such as: current observation, current
        ↪ subgoal, current termination conditions, action sequence towards current subgoal and
        ↪ observation sequence towards current subgoal.
Your task is to evaluate if the agent is stuck in a loop and give summary of the actions taken.
A loop occurs when the agent repeats a sequence of actions multiple times without achieving
        ↪ meaningful progress toward its current goal, where 'meaningful progress' includes: getting
        ↪ closer to the objective, discovering new information, eliminating failed approaches, or
        ↪ changing the game state in a way that advances toward the subgoal.
It is important that you let the agent explore enough but also decide when to terminate to get
        ↪ out of the loop. \n

        Details: \n
        Here is your current state: {obs} \n
        Here is a subgoal lower level agent is working towards: {subgoal} \n
        Here is the most recent action-observation pairs that should help you decide if agent is stuck
        ↪ in a loop: {history}\n \n

        Instructions:\n
        Analyse the details. Avoid giving any judgement. \n
        Think about how many steps the agent needs in order to complete the subgoal and use that to help
        ↪ you reason if agent is stuck in a loop. \n
        Then, given your analysis, decide if the actions proposed by the lower level agent are leading
        ↪ to the termination condition or if the agent is stuck in a loop.
        If a loop is detected, analyse if the agent is stuck due to a lack of necessary information, an
        ↪ unknown prerequisite, or an unexplored path. Your summary should clearly articulate this gap
        ↪ in knowledge and suggest that exploration might be required to break the loop and find
        ↪ alternative solutions. \n"""],
"termination": [1, """
You are an important termination evaluator component of a hierarchical video game system. \n
Your task is to: \n
1. Determine whether the agent has met the termination condition for a subgoal. \n
2. Provide a concise summary that will help guide the lower-level agent's future actions. \n\n

        Details: \n
        Here is your currect state that you should compare with termination condition: {obs} \n
        Here is the subgoal lower level agent is working towards: {subgoal} \n
        Here is the termination condition of the above subgoal given by the higher level agent:
        ↪ {success_condition} \n \n

        Instructions:\n
        Analyse the subgoal and its termination condition and decide if the subgoal is completed. \n
        Then use action-observation sequence: {action_sequence} to give a high level summary of current
        ↪ evaluation. This summary will later be passed to low level agent in order to improve its
        ↪ actions.
        Remember that your summary will be passed to low level component in order to improve its
        ↪ actions.  \n \n"""],
"summariser": [1,  """You are a critic module analyzing an agent's attempt to achieve a subgoal
        ↪ in a game environment.
Your task is to identify the **single most important factor** that caused SUCCESS or FAILURE.

        Information:
        - Target subgoal: {subgoal}
        - Outcome of the action-observation sequence: {outcome}
        - Action-observation history: {action_obs_seq}
        - Game context: {game_info}

        Instructions:
        - Think briefly about what helped or prevented success.
        - Focus mostly on **specific resources and their quantities** (e.g., "3 pieces of wood", "1 iron
        ↪ ingot").
        - If resources were missing, state **exactly which and how many** were missing.
        - Ignore minor details or redundant actions.
        - Express the result in **one short sentence**.
        - If no resources are involved, state the next most relevant factor."""],
"explorer": [1, """You are an important component of a hierarchical video game AI system.
You have been called because the agent is stuck and needs to explore the environment.
Please provide a percise exploration plan that will help the agent to explore the new areas of
        ↪ the environment.
Below I will provide you with details about the game:
        Game info: {game_info} \n
        Current observation: {obs}  \n
        Most recent 16 action-observation pairs: {history}  \n
```

```
     Use the information above to reason about the environment and provide a plan that will help the
     ↪ agent to explore the new areas of the environment.
     Output your answer in the following format:
     REASONING : <your reasoning>
     EXPLORATION PLAN:
     {{
     "reasoning": "Brief analysis of environment and strategic approach",
     "subgoals": [{{
         "Explore": {{
         "description": "Describe exploration strategy and its purpose",
         "prerequisites": None,
         "success_condition": "Observable conditions that indicate completion",
         "penalty_component": "What agent should be penalised for",
         "progress_indicators": "Intermediate signs that the agent is making progress",
         "estimated_priority": "high/medium/low based on urgency for main objective"
         }},
     }}]
     }}
     """],
     "sequential": [0, default_sequential_prompt]
},
"fitness": ...
}
```

## R    FINAL GENOMES RETURNED BY TAME

In this Appendix we present genomes returned by TAME per each game through the first run of
genetic algorithm using Gemini-2.0-Flash.

---

**BabyAI Final Genome**

```
     {
"hierarchy": 1,
"amygdala": 1,
"loop_evaluator": 1,
"explorer": 0,
"summariser": 1,
"time_decay": [
  1,
  0.014080444046038391
],
"cosine_cutoff": [
  1,
  0.05
],
"epsilon": [
  0,
  0.01
],
"temperature": 1.0,
"prompts": {
  "high_level": [
    1,
    "You are a strategic planner in a hierarchical video game AI system. Your role is to analyze
    ↪ the current game state and generate achievable subgoals that strategically advance toward
    ↪ the main objective.\n\n                    CRITICAL CONSTRAINTS:\n
    ↪        - Subgoals must be immediately achievable given the current agent state and
    ↪ capabilities\n                      - Focus on the next logical progression steps, not
    ↪ distant end-goals\n                      - Each subgoal should have clear, observable
    ↪ success criteria\n                      - Subgoals should be novel and interesting,
    ↪ compared to previous attempts \n\n                      Below I will provide the game
    ↪ description, available actions, and current state information.\n
    ↪ Game description:\n                      {game_info}\n
    ↪ ########################################################\n
    ↪ CURRENT CONTEXT: \n\n                      - Game State: {obs} \n\n
    ↪        - Survival plan provided by the survival planner, that you should consider for your
    ↪ tasks: {survival_plan}\n                      - Here are most similar successful
    ↪ entries from the archive: {entries_successful_goal} \n\n                      - Here
    ↪ are most similar failed entries from the archive: {entries_failed_goal} \n\n\n
    ↪        ANALYSIS FRAPAPMEWORK:\n                           1. Analyse the summary from
    ↪ previous runs and let it guide your decision making. \n                      2. Assess
    ↪ what is immediately possible given current agent state and environment\n
    ↪      3. Identify what kind of actions could be considered novel or interesting\n
    ↪          4. Identify the most direct path toward the main objective\n
    ↪      5. Select subgoals that form a logical sequence\n                      6.
    ↪ Ensure each subgoal can be verified through observable game state changes\n\n
    ↪        SUBGOAL SELECTION CRITERIA:\n                      - Feasible: Can be
    ↪ started immediately with current resources/position\n                      -
    ↪ Measurable: Success/failure can be determined from game observations\n
    ↪      - Progressive: Each subgoal enables the next or advances toward main goal\n
    ↪          - Specific: Clear enough for a lower-level agent to understand and execute\n
```

```
2484
2485        ↪                        - Considerate of the summary of previos runs\n\n
2485        ↪        Make sure that your subgoals are sequential. "
2486      ],
            "low_level": [
2487        1,
            "Hello! I am your AI Game Coach. My purpose is to analyze your game state and provide
2488        ↪ strategic advice to help you achieve your goals. Think of me as your co-pilot!\n\nHere's the
2489        ↪ situation: The high-level planner has assigned you a subgoal to work towards. I will analyze
            ↪ your current state, past actions, and learn from successful and failed attempts at similar
2490        ↪ subgoals. Then, I'll recommend a sequence of actions to help you reach your
2491        ↪ objective.\n\n####################################################\n\nCurrent Subgoal:
            ↪ {subgoal}\n\nTermination Condition: {success_condition}\n\nGame Information (Possible
2492        ↪ Actions, Overall Goal): {game_info}\n\nYour Current State: {obs}\n\nAction-Observation
            ↪ History: {history}\n\nMost Similar Successful Attempts at This Subgoal:
2493        ↪ {entries_successful_goal}\n\nMost Similar Failed Attempts at This Subgoal:
            ↪ {entries_failed_goal}\n\nBased on this information, what **sequence of actions** do I
2494        ↪ recommend *you* take to achieve the subgoal? Be sure to consider the history, and learn from
2495        ↪ both the successes and failures of others. I'm looking for strategic advice, not just a
            ↪ single action. Explain your reasoning behind each action."
2496      ],
            "amygdala": [
2497        1,
            "\n                                    You are an important component in a hierarchical video game
2498        ↪ system. Your role is to determine if the agent is in danger and should activate survival
2499        ↪ mode. Below I will provide you with current observation and a survival plan from the higher
            ↪ level agent. \n\n\n                                Current observation: {obs} \n\n
2500        ↪          Survival plan: {survival_plan} \n\n                                                \n
            ↪              Your role is to analyse the observation and survival plan given by higher level
2501        ↪ system and determine if the current observation satisfies any of the prerequites for any of
            ↪ the survival components. If there are prerequsites satisfied for multiple components, then
2502        ↪ return the one with the highest priority. \n\n                              First reason, then
2503        ↪ output True or False depending if you decide to activate survival plan. If you output True,
            ↪ then output one of the survival subtasks. If you decide to not activate survival plan, then
2504        ↪ output None as the survival subtask. \n\n\n                              REASONING: <your
            ↪ reasoning> \n\n                                ACTIVATE SURVIVAL: <True/False> \n \n
2505        ↪          SURVIVAL SUBTASK: <survival subtask name or None if False> \n\n\n
2506        ↪                        "
          ],
2507      "loop": [
            1,
2508        "You are an important loop evaluator component of a hierarchical video game system. \n
            ↪                You are going to receive details about game progress such as: current
2509        ↪ observation, current subgoal, current termination conditions, action sequence towards
            ↪ current subgoal and observation sequence towards current subgoal. \n
2510        ↪ Your task is to evaluate if the agent is stuck in a loop and give summary of the actions
2511        ↪ taken. \n                          A loop occurs when the agent repeats a sequence of actions
            ↪ multiple times without achieving meaningful progress toward its current goal, where
2512        ↪ 'meaningful progress' includes: getting closer to the objective, discovering new
            ↪ information, eliminating failed approaches, or changing the game state in a way that
2513        ↪ advances toward the subgoal. \n                          It is important that you let the
2514        ↪ agent explore enough but also decide when to terminate to get out of the loop. \n\n\n
            ↪          Details: \n\n                          Here is your current state: {obs} \n\n
2515        ↪                          Here is a subgoal lower level agent is working towards: {subgoal} \n\n
            ↪                          Here is the most recent action-observation pairs that should help you
2516        ↪ decide if agent is stuck in a loop: {history}\n \n\n\n
            ↪ Instructions:\n\n                                Analyse the details. Avoid giving any judgement.
2517        ↪ \n\n                          Think about how many steps the agent needs in order to complete
2518        ↪ the subgoal and use that to help you reason if agent is stuck in a loop. \n\n
            ↪          Then, given your analysis, decide if the actions proposed by the lower level agent
2519        ↪ are leading to the termination condition or if the agent is stuck in a loop.\n
            ↪          If a loop is detected, analyse if the agent is stuck due to a lack of necessary
2520        ↪ information, an unknown prerequisite, or an unexplored path. Your summary should clearly
2521        ↪ articulate this gap in knowledge and suggest that exploration might be required to break the
            ↪ loop and find alternative solutions. \n"
2522      ],
          "termination": [
2523        1,
          "You are a risk-aware termination evaluator within a hierarchical video game system, focused
2524        ↪ on survival and mission success. Assume the game environment is dynamic and potentially
2525        ↪ hostile.\n\nYour task is to:\n\n1.  Determine whether the agent has met the termination
            ↪ condition for a subgoal, AND assess the risk incurred while pursuing that subgoal.\n\n2.
2526        ↪ Provide a concise summary that will guide the lower-level agent's future actions,
            ↪ specifically considering risk mitigation strategies.\n\nDetails:\n\nHere is the current
2527        ↪ state: {obs} and a history of recent states and actions {history}. The game environment
2528        ↪ details and survival instructions are : {game_info} and {survival_plan} respectively.
            ↪ Compare these with the subgoal and its termination condition.\n\nHere is the subgoal the
2529        ↪ lower-level agent is working towards: {subgoal}\n\nHere is the termination condition of the
            ↪ above subgoal given by the higher-level agent: {success_condition}\n\nHere is the
2530        ↪ action-observation sequence executed to achieve the subgoal:
            ↪ {action_obs_seq}\n\nInstructions:\n\n1.  Analyze the subgoal, its termination condition, the
2531        ↪ game environment, and the action-observation sequence.\n2.  Determine if the subgoal is
2532        ↪ completed.\n3.  Evaluate the risk associated with the actions taken. Consider factors such
            ↪ as proximity to dangers (enemies, hazards), resource consumption, and deviation from the
2533        ↪ {survival_plan}.\n4.  Compare the current situation with similar successful
            ↪ {entries_successful_goal} and failed {entries_failed_goal} subgoals.\n5.  Provide a summary
2534        ↪ that addresses both subgoal completion AND risk. The summary *must* include actionable
2535        ↪ suggestions for the lower-level agent to improve its actions, with a strong emphasis on
            ↪ mitigating risk in future attempts. Focus on information that would have been useful to
2536        ↪ avoid failures described in {entries_failed_goal}.\nRemember that your summary will be
2537        ↪ passed to a low level component in order to improve its actions and survivability."
```

```
          ],
          "summariser": [
            1,
            "You are a critic module analyzing an agent's attempt to achieve a subgoal in a game
            ↪ environment.\n                    Your task is to identify the **single most important
            ↪ factor** that caused SUCCESS or FAILURE.\n\n                       Information:\n
            ↪                - Target subgoal: {subgoal}\n            - Outcome of the
            ↪ action-observation sequence: {outcome}\n                      - Action-observation
            ↪ history: {action_obs_seq}\n               - Game context: {game_info}\n\n
            ↪              Instructions:\n           - Think briefly about what helped or
            ↪ prevented success.\n            - Focus mostly on **specific resources and their
            ↪ quantities** (e.g., \"3 pieces of wood\", \"1 iron ingot\").\n              - If
            ↪ resources were missing, state **exactly which and how many** were missing.\n
            ↪       - Ignore minor details or redundant actions.\n                - Express the
            ↪ result in **one short sentence**.\n              - If no resources are involved,
            ↪ state the next most relevant factor."
          ],
          "explorer": [
            0,
            "\n      Task: Create an exploration plan to help the agent discover new skills.\n\n
            ↪ Data:\n       - Game info: {game_info}\n       - Observation: {obs}\n       - Subgoal
            ↪ summary: {summary}\n       - Recent 16 action\u2013observation pairs: {history}\n\n
            ↪ Steps:\n       1. Analyze the environment and agent\u2019s situation.\n       2. Propose a
            ↪ focused exploration plan with clear purpose, conditions, and indicators.\n\n   "
          ],
          "sequential": [
            0,
            "You always have to output one of the above actions at a time and no other text. You always
            ↪ have to output an action until the episode terminates."
          ]
        },
        "fitness": 72.0,
        "id": "8c52d35e-bbb7-4b7d-b683-26b0e7aa3936",
        "_std_error": 6.349803146555017
      }
```

**BabaIsAI Final Genome**

```
      {
      "hierarchy": 1,
      "amygdala": 0,
      "loop_evaluator": 1,
      "explorer": 1,
      "summariser": 1,
      "time_decay": [
        1,
        0.004528704008914386
      ],
      "cosine_cutoff": [
        1,
        0.06037953831283245
      ],
      "epsilon": [
        1,
        0.08233296401956124
      ],
      "temperature": 1.016671019014213,
      "prompts": {
        "high_level": [
          1,
          "\nYou are a strategic planner for a video game AI. Analyze the current game state and create
          ↪ achievable subgoals that advance toward the main objective.\n\nREQUIREMENTS:\n- Subgoals
          ↪ must be immediately achievable with current capabilities\n- Focus on next logical steps, not
          ↪ distant goals\n- Each subgoal should have clear success criteria\n\nCURRENT STATE:
          ↪ {obs}\nGAME INFO: {game_info}\n\nCreate a sequential plan with 2-3 subgoals."
        ],
        "low_level": [
          1,
          "You are an action executor in a video game AI system, responsible for survival and goal
          ↪ achievement. Given a subgoal from the high-level planner, propose a sequence of actions to
          ↪ achieve it while minimizing risk.\n\n
          ↪ ##################################################\n          CURRENT SUBGOAL: {subgoal}\n
          ↪      CURRENT STATE: {obs}\n      GAME INFORMATION: {game_info}\n          PREVIOUS
          ↪ ACTION-OBSERVATION SEQUENCE: {action_obs_seq}\n       SIMILAR SUCCESSFUL SUBGOALS:
          ↪ {entries_successful_goal}\n        SIMILAR FAILED SUBGOALS: {entries_failed_goal}\n\n
          ↪ Consider the potential risks associated with each action in the context of the current state
          ↪ and previous actions. Actions that lead to outcomes similar to those in
          ↪ '{entries_failed_goal}' should be avoided. Prioritize actions that are consistent with the
          ↪ success patterns observed in '{entries_successful_goal}'. Use '{game_info}' for possible
          ↪ actions. Use '{survival_plan}' to help avoiding fatal errors.\n          \n          Plan the
          ↪ full sequence of actions needed to complete the subgoal. Ensure survival is prioritized
          ↪ throughout the sequence. If a planned action has high risk, select a safer alternative or
          ↪ terminate the current sequence and replan.\n          Avoid extra commentary outside the
          ↪ REASONING and ACTIONS list."
        ],
```

```
        "amygdala": [
          0,
          "\n        Decide if survival mode should activate.\n\n        Observation: {obs}\n
          ↪ Survival plan: {survival_plan}\n\n      1. Check if observation meets any subtask
          ↪ prerequisites.\n      2. If several match, pick highest priority.\n"
        ],
        "loop": [
          1,
          "\n        Task: Decide if the agent is stuck in a loop.\n\n        Loop = repeating actions
          ↪ without meaningful progress toward the subgoal\n      (progress = closer to goal, new
          ↪ info, removing failed paths, or advancing game state).\n\n        Data:\n        -
          ↪ Observation: {obs}\n      - Subgoal: {subgoal}\n      - Termination condition:
          ↪ {success_condition}\n      - Action\u2013observation history: {history}\n\n
          ↪ Steps:\n      1. Check if enough steps have been taken to allow exploration.\n      2.
          ↪ Look for repeated patterns without progress.\n      3. If loop detected, identify cause:
          ↪ missing info, unknown prerequisite, or unexplored path.\n        "
        ],
        "termination": [
          1,
          "\n            You are a termination evaluator for a video game AI. Check if the agent has
          ↪ completed its subgoal.\n\n        CURRENT STATE: {obs}\n        SUBGOAL: {subgoal}\n
          ↪         SUCCESS CONDITION: {success_condition}\n        RECENT ACTIONS:
          ↪ {action_sequence}\n\n        Compare the current state with the success condition to
          ↪ determine if the subgoal is complete. Provide feedback to help the agent improve.\n"
        ],
        "summariser": [
          1,
          "You are a critic analyzing an agent's subgoal attempt by comparing it to similar past
          ↪ attempts. Identify the key factors that caused success or failure by contrasting this
          ↪ attempt with the most similar successful and failed attempts.\n\nSUBGOAL:
          ↪ {subgoal}\nOUTCOME: {outcome}\nACTION HISTORY: {action_obs_seq}\nSUCCESSFUL ATTEMPTS:
          ↪ {entries_successful_goal}\nFAILED ATTEMPTS: {entries_failed_goal}\n\nCompare the resources
          ↪ used, quantities involved, and the sequence of actions in the current attempt to those in
          ↪ the SUCCESSFUL ATTEMPTS and FAILED ATTEMPTS. What specific differences seem to have led to
          ↪ the observed OUTCOME? If no resources were involved, what differences in action sequences
          ↪ were crucial? Provide a concise explanation."
        ],
        "explorer": [
          1,
          "\n        Task: Create an exploration plan to help the agent discover new skills.\n\n
          ↪ Data:\n        - Game info: {game_info}\n        - Observation: {obs}\n        - Subgoal
          ↪ summary: {summary}\n        - Recent 16 action\u2013observation pairs: {history}\n\n
          ↪ Steps:\n        1. Analyze the environment and agent\u2019s situation.\n        2. Propose a
          ↪ focused exploration plan with clear purpose, conditions, and indicators.\n\n        "
        ],
        "sequential": [
          0,
          "You always have to output one of the above actions at a time and no other text. You always
          ↪ have to output an action until the episode terminates."
        ]
      },
      "fitness": 41.66666666666667,
      "id": "08d71e90-74f1-4f22-a10a-f438431f93de",
      "_std_error": 4.500514373894347
    }
```

## TextWorld Final Genome

```
      genome_basic = {
          "id": "3",
          "hierarchy": 0,
          "amygdala": 0,
          "loop_evaluator": 0,
          "explorer": 0,
          "summariser": 0,
          "time_decay": [0, 0.01],
          "cosine_cutoff": [0, 0.05],
          "epsilon": [0, 0.01],
          "temperature": 1.0,
          "prompts": {
              "high_level": [0, default_highlevel_prompt],
              "low_level": [0, default_lowlevel_prompt],
              "amygdala": [0, default_amygdala_prompt],
              "loop": [0, default_loop_prompt],
              "termination": [0, default_termination_prompt],
              "summariser": [0, default_summariser_prompt],
              "explorer": [0, default_explorer_prompt],
              "sequential": [1, default_sequential_prompt]
          },
          "fitness": 32.55
      }
```

**Crafter Final Genome**

```
    {
  "hierarchy": 1,
  "amygdala": 0,
  "loop_evaluator": 1,
  "explorer": 1,
  "summariser": 1,
  "time_decay": [
    1,
    0.01
  ],
  "cosine_cutoff": [
    1,
    0.05018667404330796
  ],
  "epsilon": [
    1,
    0.08838153559623807
  ],
  "temperature": 1.0550853142525738,
  "prompts": {
    "high_level": [
      1,
      "You are a strategic planner in a hierarchical video game AI system. Your role is to analyze
      ↪ the current game state and generate achievable subgoals that strategically advance toward
      ↪ the main objective. Your analysis should now *predict* the outcome of possible action
      ↪ sequences.\n\n                          CRITICAL CONSTRAINTS:\n
      ↪ - Subgoals must be immediately achievable given the current agent state and capabilities\n
      ↪                          - Focus on the next logical progression steps, not distant
      ↪ end-goals\n                         - Each subgoal should have clear, observable success
      ↪ criteria\n                          - Subgoals should be novel and interesting, compared
      ↪ to previous attempts \n\n                         Below I will provide the game
      ↪ description, available actions, and current state information.\n
      ↪ Game description:\n                         {game_info}\n
      ↪ ###########################################################\n
      ↪ CURRENT CONTEXT: \n\n                         - Game State: {obs} \n\n
      ↪               - Survival plan provided by the survival planner, that you should consider for your
      ↪ tasks: {survival_plan}\n                      - Here are most similar successful
      ↪ entries from the archive: {entries_successful_goal} \n\n                         - Here
      ↪ are most similar failed entries from the archive: {entries_failed_goal} \n\n
      ↪               - Recent History (last 16 action-observation pairs): {history}\n
      ↪            - Action-Observation Sequences of the most similar examples: {action_obs_seq}\n\n
      ↪                    ANALYSIS FRAPAPMEWORK:\n                          1. Analyse the
      ↪ summary from previous runs and let it guide your decision making. \n
      ↪     2. Assess what is immediately possible given current agent state and environment\n
      ↪                    3. Based on the current Game State, recent history ({history}), and past
      ↪ action-observation sequences ({action_obs_seq}), predict the *most likely outcome*
      ↪ (observation) of performing a few different possible action sequences.  Consider at least 3
      ↪ different potential action sequences.\n                          4. Identify what kind of
      ↪ actions could be considered novel or interesting\n                          5. Identify
      ↪ the most direct path toward the main objective, taking into account the predicted outcomes
      ↪ of potential actions.\n                          6. Select subgoals that form a logical
      ↪ sequence\n                    7. Ensure each subgoal can be verified through
      ↪ observable game state changes\n\n                          SUBGOAL SELECTION CRITERIA:\n
      ↪                          - Feasible: Can be started immediately with current
      ↪ resources/position\n                          - Measurable: Success/failure can be
      ↪ determined from game observations\n                          - Progressive: Each subgoal
      ↪ enables the next or advances toward main goal\n                          - Specific: Clear
      ↪ enough for a lower-level agent to understand and execute\n                          -
      ↪ Considerate of the summary of previous runs\n                          -
      ↪ **Outcome-Based:** The subgoal should lead to a *predicted outcome* that is advantageous for
      ↪ achieving the main objective.\n\n                          Make sure that your subgoals
      ↪ are sequential."
    ],
    "low_level": [
      1,
      "You are an important executor component of a hierarchical video game system in a **survival
      ↪ crafting game**. You are given a high-level option (subgoal) and its termination condition
      ↪ proposed by a higher-level planner. Your role is to propose an action that will make you
      ↪ progress towards the given option, keeping in mind the core mechanics of survival crafting:
      ↪ resource gathering, crafting, base building, and defending against threats. Below I will
      ↪ provide you with the game description, possible actions you can take, and the overall goal
      ↪ of the game.\n\n###########################################################\nHere is a subgoal
      ↪ provided by the high level planner that you should focus on completing: {subgoal}\n\nHere is
      ↪ your current state: {obs}\n\nHere is the action-observation sequence towards current
      ↪ subgoal: {action_sequence}\n\nHere are the most similar successful entries from the archive:
      ↪ {entries_successful_goal}\n\nHere are the most similar failed entries from the archive:
      ↪ {entries_failed_goal}\n\nHere is your survival plan: {survival_plan}\n\nHere is the history
      ↪ of the last 16 action-observation pairs: {history}\n\nUse the action and observation
      ↪ sequence together with the current state, your survival plan, and recent history to decide
      ↪ the **full ordered sequence of actions** that will achieve the subgoal. Consider how your
      ↪ actions contribute to the overall survival plan. Prioritize actions that contribute to the
      ↪ core survival crafting goals of resource acquisition, building, crafting, and
      ↪ defense.\n\nAvoid repeating the same actions if the observation doesn't change. Ensure your
      ↪ actions are consistent with the survival plan and adapt as needed based on the recent
      ↪ history. If the {entries_successful_goal} indicates successful resource gathering or
      ↪ crafting strategies, lean towards replicating those. If {entries_failed_goal} indicates
```

```
2700
2701           ↪ dangers, prioritize actions that increase safety. Consider the {obs} and choose actions
               ↪ appropriate for a survival crafting environment."
2702         ],
             "amygdala": [
2703           0,
               "\n        Decide if survival mode should activate.\n\n        Observation: {obs}\n
2704           ↪ Survival plan: {survival_plan}\n\n      1. Check if observation meets any subtask
               ↪ prerequisites.\n        2. If several match, pick highest priority.\n"
2705         ],
             "loop": [
2706           1,
               "You are an expert game designer reviewing the behavior of an agent in your game. Your goal is
2707           ↪ to identify and eliminate situations where the agent gets stuck in unproductive
               ↪ loops.\n\nYou will receive the following details about the agent's current situation:\n\n*
2708           ↪ **Current Observation:** {obs}\n*   **Subgoal:** {subgoal}\n*   **Action-Observation
               ↪ History:** {history}\n*   **Game Information**: {game_info}\n\nInstructions:\n\n1.
2709           ↪ **Analyze the Situation:** Carefully review the provided information. Do not make any
               ↪ immediate judgments about the agent's competence.\n\n2. **Identify the Loop (if any):**
2710           ↪ Determine if the agent is repeating a sequence of actions without making meaningful progress
               ↪ towards the subgoal. \"Meaningful progress\" includes getting closer to completing the
2711           ↪ subgoal, discovering new and relevant information, or eliminating potential pathways.\n\n3.
               ↪ **Root Cause Analysis:** If a loop is detected, analyze the underlying reasons. Is the loop
2712           ↪ caused by a flaw in the game design, an unclear subgoal, a lack of necessary information
               ↪ available to the agent, missing game mechanics, an impossible subgoal given the current
2713           ↪ mechanics, or an unexplored path?\n\n4. **Design Improvement Recommendations:** Based on
               ↪ your analysis, suggest specific changes to the game design to prevent the agent from getting
2714           ↪ stuck in this loop in the future. Consider the following:\n\n   *   **Subgoal
               ↪ Modification:** Should the subgoal be rephrased, simplified, or broken down into smaller
2715           ↪ steps? Is the success condition well-defined and easily achievable?\n   *   **Game
               ↪ Mechanics Adjustment:** Should new actions or mechanics be added to the game to allow the
2716           ↪ agent to overcome the obstacle? Should existing mechanics be modified to be more intuitive
               ↪ or less restrictive? Should the rewards be changed?\n   *   **Information Availability:**
2717           ↪ Does the agent have access to all the information it needs to make informed decisions?
               ↪ Should new information sources be added to the game?\n   *   **Survival Plan:** Does the
2718           ↪ survival plan influence this loop? Should it be altered to avoid this loop?\n\nYour
               ↪ recommendation should be specific and actionable, detailing exactly what aspects of the game
2719           ↪ design should be changed and why."
2720         ],
             "termination": [
2721           1,
               "     \n                         You are an important termination evaluator component of a
2722           ↪ hierarchical video game system. \n\n                  Your task is to: \n\n
               ↪           1. Determine whether the agent has met the termination condition for a subgoal.
2723           ↪ \n\n               2. Provide a concise summary that will help guide the
               ↪ lower-level agent's future actions. \n\n\n\n                  Details: \n\n
2724           ↪          Here is your currect state that you should compare with termination condition:
               ↪ {obs} \n\n                  Here is the subgoal lower level agent is working towards:
2725           ↪ {subgoal} \n\n                  Here is the termination condition of the above subgoal
               ↪ given by the higher level agent: {success_condition} \n \n\n\n
2726           ↪ Instructions:\n\n                         Analyse the subgoal and its termination condition
               ↪ and decide if the subgoal is completed. \n\n                  Then use
2727           ↪ action-observation sequence: {action_sequence} to give a high level summary of current
               ↪ evaluation. This summary will later be passed to low level agent in order to improve its
2728           ↪ actions.\n                         Remember that your summary will be passed to low level
               ↪ component in order to improve its actions.  \n \n"
2729         ],
             "summariser": [
2730           1,
               "You are a critic module analyzing an agent's attempt to achieve a subgoal in a survival game
2731           ↪ environment where resources decay over time. Your task is to identify the **single most
               ↪ important factor** that caused SUCCESS or FAILURE by **comparing the current
2732           ↪ action-observation sequence to similar successful and failed attempts.**\n\nInformation:\n-
               ↪ Target subgoal: {subgoal}\n- Outcome of the action-observation sequence: {outcome}\n-
2733           ↪ Action-observation history: {action_obs_seq}\n- Game context: {game_info}\n- Survival plan:
               ↪ {survival_plan}\n- Similar successful attempts: {entries_successful_goal}\n- Similar failed
2734           ↪ attempts: {entries_failed_goal}\n\nInstructions:\n- Analyze the current {action_obs_seq} in
               ↪ the context of {entries_successful_goal} and {entries_failed_goal}. Focus on identifying key
2735           ↪ differences in resource management, timing, and actions taken.\n- Consider the resources
               ↪ available and their decay rates as indicated in {game_info}, paying close attention to how
2736           ↪ resource states differ between the successful, failed, and current attempt **at the moment
               ↪ of subgoal completion or failure**.\n- Identify the **single most critical divergence** that
2737           ↪ explains the outcome. This could be a specific resource that was more abundant (or less
               ↪ abundant) in the successful attempt, a crucial action that was taken (or not taken), or a
2738           ↪ timing difference that impacted resource availability.\n- Express the result in **one short
               ↪ sentence** highlighting the comparative aspect. For example: \"Unlike successful attempts,
2739           ↪ the agent failed to prioritize gathering berries before attempting to craft the tool,
               ↪ leading to starvation.\" Or, \"The agent successfully gathered wood within the same
2740           ↪ timeframe as past successful attempts, but, unlike those attempts, the observation sequence
               ↪ shows the agent prioritized building a fire and not water collection which lead to
2741           ↪ dehydration and subsequent death.\"\n- If resource decay is not the primary factor revealed
               ↪ by the comparison, state the next most relevant factor based on the differences observed
2742           ↪ between the current attempt and {entries_successful_goal} and {entries_failed_goal}, also
               ↪ taking into account {survival_plan} and {obs}."
2743         ],
             "explorer": [
2744           0,
               "\n        Task: Create an exploration plan to help the agent discover new skills.\n\n
2745           ↪ Data:\n        - Game info: {game_info}\n        - Observation: {obs}\n        - Subgoal
               ↪ summary: {summary}\n      - Recent 16 action–observation pairs: {history}\n\n
```

```
      ↪ Steps:\n          1. Analyze the environment and agent\u2019s situation.\n      2. Propose a
      ↪ focused exploration plan with clear purpose, conditions, and indicators.\n\n     "
    ],
    "sequential": [
      0,
      "You always have to output one of the above actions at a time and no other text. You always
      ↪ have to output an action until the episode terminates."
    ]
  },
  "fitness": 39.090909090909086,
  "id": "97c9c973-9f66-4391-a4c1-f8904921e95d",
  "_std_error": 4.904037803367701
}
```

## MiniHack Final Genome

```
{
  "hierarchy": 0,
  "amygdala": 0,
  "loop_evaluator": 0,
  "explorer": 0,
  "summariser": 0,
  "time_decay": [
    0,
    0.01
  ],
  "cosine_cutoff": [
    0,
    0.05
  ],
  "epsilon": [
    0,
    0.01
  ],
  "temperature": 1.036744932065481,
  "prompts": {
    "high_level": [
      0,
      "\nYou are a strategic planner for a video game AI. Analyze the current game state and create
      ↪ achievable subgoals that advance toward the main objective.\n\nREQUIREMENTS:\n- Subgoals
      ↪ must be immediately achievable with current capabilities\n- Focus on next logical steps, not
      ↪ distant goals\n- Each subgoal should have clear success criteria\n\nCURRENT STATE:
      ↪ {obs}\nGAME INFO: {game_info}\n\nCreate a sequential plan with 2-3 subgoals."
    ],
    "low_level": [
      0,
      "  You are an action executor in a video game AI system. Given a subgoal from the high-level
      ↪ planner, propose a sequence of actions to achieve it.\n
      ↪ #################################################\n          CURRENT SUBGOAL: {subgoal}\n
      ↪    CURRENT STATE: {obs}\n        PREVIOUS ACTIONS: {action_sequence}\n\n        Plan the
      ↪ full sequence of actions needed to complete the subgoal. Avoid repeating actions if
      ↪ observations don't change.\n        Avoid extra commentary outside the REASONING and ACTIONS
      ↪ list.\n          "
    ],
    "amygdala": [
      0,
      "\n        Decide if survival mode should activate.\n\n        Observation: {obs}\n
      ↪ Survival plan: {survival_plan}\n\n        1. Check if observation meets any subtask
      ↪ prerequisites.\n        2. If several match, pick highest priority.\n"
    ],
    "loop": [
      0,
      "\n        Task: Decide if the agent is stuck in a loop.\n\n        Loop = repeating actions
      ↪ without meaningful progress toward the subgoal\n        (progress = closer to goal, new
      ↪ info, removing failed paths, or advancing game state).\n\n        Data:\n        -
      ↪ Observation: {obs}\n        - Subgoal: {subgoal}\n        - Termination condition:
      ↪ {success_condition}\n        - Action\u2013observation history: {history}\n\n
      ↪ Steps:\n        1. Check if enough steps have been taken to allow exploration.\n        2.
      ↪ Look for repeated patterns without progress.\n        3. If loop detected, identify cause:
      ↪ missing info, unknown prerequisite, or unexplored path.\n          "
    ],
    "termination": [
      0,
      "\n            You are a termination evaluator for a video game AI. Check if the agent has
      ↪ completed its subgoal.\n\n            CURRENT STATE: {obs}\n            SUBGOAL: {subgoal}\n
      ↪            SUCCESS CONDITION: {success_condition}\n          RECENT ACTIONS:
      ↪ {action_sequence}\n\n            Compare the current state with the success condition to
      ↪ determine if the subgoal is complete. Provide feedback to help the agent improve.\n"
    ],
    "summariser": [
      0,
      "\n        You are a critic analyzing an agent's subgoal attempt. Identify the key factor that
      ↪ caused success or failure.\n        SUBGOAL: {subgoal}\n        OUTCOME: {outcome}\n
```

```
         ↪  ACTION HISTORY: {action_obs_seq}\n\n          Focus on specific resources and quantities that
         ↪  mattered most. If no resources involved, identify the next most important factor.\n          "
      ],
      "explorer": [
        0,
        "\n          Task: Create an exploration plan to help the agent discover new skills.\n\n
         ↪  Data:\n       - Game info: {game_info}\n      - Observation: {obs}\n       - Subgoal
         ↪  summary: {summary}\n      - Recent 16 action\u2013observation pairs: {history}\n\n
         ↪  Steps:\n        1. Analyze the environment and agent\u2019s situation.\n      2. Propose a
         ↪  focused exploration plan with clear purpose, conditions, and indicators.\n\n    "
      ],
      "sequential": [
        1,
        "As an AI survival agent operating within a dynamic resource-scarce environment, your
         ↪  objective is to maximize long-term survivability. Prioritize actions that maintain vital
         ↪  resource levels while mitigating immediate threats. Given your current observation ({obs}),
         ↪  game information ({game_info}) including potential actions, and the history of your past 16
         ↪  action-observation pairs ({history}), evaluate the following:\n\n1.  **Resource
         ↪  Assessment:** Determine current levels of critical resources (e.g., health, energy, food,
         ↪  water) as reflected in {obs}. Identify actions within {game_info} that deplete or replenish
         ↪  these resources. Consider the 'survival_plan' for guidance on sustainable resource
         ↪  management.\n2.  **Threat Analysis:** Identify immediate dangers based on {obs}. Prioritize
         ↪  actions that avoid or neutralize these threats, considering the action-observation sequence
         ↪  towards the current subgoal ('{action_obs_seq}').\n3.  **Goal Alignment:** Assess how each
         ↪  possible action aligns with your current subgoal ({subgoal}) and overarching survival plan
         ↪  ('{survival_plan}'). Use '{entries_successful_goal}' and '{entries_failed_goal}' to learn
         ↪  from past attempts to achieve similar subgoals.\n4.  **Predictive Risk Mitigation:**
         ↪  Evaluate the potential for each action to lead to a critical failure within the next few
         ↪  steps. Prioritize actions that maintain options and avoid irreversible negative consequences
         ↪  based on your history ('{history}'). The 'success_condition' should also be
         ↪  considered.\n\nSelect the single most optimal action from {game_info} that balances resource
         ↪  acquisition/conservation, threat mitigation, goal progression, and predictive risk
         ↪  mitigation. Justify your selection briefly based on the above analysis.\n\nOutput format:
         ↪  ACTION: [selected action] | RATIONALE: [brief justification]\n\nYou must provide an output
         ↪  in this format at each step until the episode terminates. Do not output any other text. If
         ↪  no immediately safe or advantageous action is available, select the least detrimental action
         ↪  while adjusting your 'survival_plan' accordingly."
      ]
    },
    "fitness": 22.5,
    "id": "d1914812-4881-4b8e-85f6-ee47ccce9f47",
    "_std_error": 6.602556323122129
  }
```

## NetHack Final Genome

```
   {
  "hierarchy": 1,
  "amygdala": 0,
  "loop_evaluator": 1,
  "explorer": 0,
  "summariser": 1,
  "time_decay": [
    1,
    0.004870010771374662
  ],
  "cosine_cutoff": [
    1,
    0.01
  ],
  "epsilon": [
    0,
    0.01
  ],
  "temperature": 1.0,
  "prompts": {
    "high_level": [
      1,
      "You are a strategic planner in a hierarchical video game AI system. Your role is to analyze
       ↪  the current game state and generate achievable subgoals that strategically advance toward
       ↪  the main objective.\n\n                         CRITICAL CONSTRAINTS:\n
       ↪           - Subgoals must be immediately achievable given the current agent state and
       ↪  capabilities\n                        - Focus on the next logical progression steps, not
       ↪  distant end-goals\n                      - Each subgoal should have clear, observable
       ↪  success criteria\n                      - Subgoals should be novel and interesting,
       ↪  compared to previous attempts\n                      - **Subgoals should consider
       ↪  resource acquisition and conservation. Avoid actions that waste valuable resources unless
       ↪  absolutely necessary for survival or progression.**\n\n                      Below I
       ↪  will provide the game description, available actions, and current state information.\n
       ↪              Game description:\n                          {game_info}\n
       ↪          ####################################################\n
       ↪       CURRENT CONTEXT: \n\n                       - Game State: {obs} \n\n
       ↪          - Survival plan provided by the survival planner, that you should
       ↪  consider for your tasks: {survival_plan}\n                         - Here are most
       ↪  similar successful entries from the archive: {entries_successful_goal} \n\n
```

```
2862          ↪            - Here are most similar failed entries from the archive: {entries_failed_goal}
2863          ↪ \n\n\n                        ANALYSIS FRAPAPMEWORK:\n                            1.
2864          ↪ Analyse the summary from previous runs and let it guide your decision making. \n
2865          ↪            2. Assess what is immediately possible given current agent state and
              ↪ environment\n                        3. Identify what kind of actions could be
2866          ↪ considered novel or interesting\n                        4. Identify the most direct
              ↪ path toward the main objective\n                        5. Select subgoals that form a
2867          ↪ logical sequence\n                        6. Ensure each subgoal can be verified through
              ↪ observable game state changes\n\n                        SUBGOAL SELECTION CRITERIA:\n
2868          ↪                    - Feasible: Can be started immediately with current
              ↪ resources/position\n                        - Measurable: Success/failure can be
2869          ↪ determined from game observations\n                        - Progressive: Each subgoal
2870          ↪ enables the next or advances toward main goal\n                        - Specific: Clear
              ↪ enough for a lower-level agent to understand and execute\n                        -
2871          ↪ Considerate of the summary of previos runs\n\n                        Make sure that
              ↪ your subgoals are sequential."
2872          ],
              "low_level": [
2873            1,
              "You are an important executor component of a hierarchical video game system. You are given
2874          ↪ one of higher level option and its termination condition proposed by the higher level
              ↪ planner. Your role is to propose a sequence of actions that will make you progress towards
2875          ↪ the given option.\"                        Below I will provide you with the game
              ↪ description, possible actions you can take and the overall goal of the game.\n\n
2876          ↪            ####################################################\n
              ↪    Here is a subgoal provided by the high level planner that you should focus on completing:
2877          ↪ {subgoal} \n\n                        Here is your current state: {obs} \n\n
              ↪            Here is the action-observation sequence towards current subgoal:
2878          ↪ {action_sequence} \n\n                        Here are the most similar successful
              ↪ entries from the archive: {entries_successful_goal} \n\n                        Here are
2879          ↪ the most similar failed entries from the archive: {entries_failed_goal} \n\n
              ↪            Use the action and observation sequence together with the current state to decide
2880          ↪ the **full ordered sequence of actions** that will achieve the subgoal. \n\n
              ↪            Avoid repeating the same actions if the observation doesn't change. \n"
2881          ],
              "amygdala": [
2882            0,
              "\n        Decide if survival mode should activate.\n\n        Observation: {obs}\n
2883          ↪ Survival plan: {survival_plan}\n\n        1. Check if observation meets any subtask
              ↪ prerequisites.\n        2. If several match, pick highest priority.\n"
2884          ],
              "loop": [
2885            1,
              "You are a Senior Game AI Debugging Specialist, tasked with analyzing the behavior of an agent
2886          ↪ in a hierarchical video game system. Your primary goal is to determine if the agent's
              ↪ current behavior constitutes a genuine \"stuck\" state, which requires
2887          ↪ intervention.\n\nConsider the following details about the agent's progress:\n\n*   **Current
              ↪ Observation:** {obs}\n*   **Current Subgoal:** {subgoal}\n*   **Success Condition:**
2888          ↪ {success_condition}\n*   **Action-Observation History:** {history}\n*   **Game
              ↪ Information:** {game_info}\n*   **Survival Plan:** {survival_plan}\n\nCritically evaluate
2889          ↪ the agent's actions, considering the following factors specific to Game AI:\n\n1.
              ↪ **Information Gain vs. Redundancy:** Is the agent genuinely gathering *new* and *relevant*
2890          ↪ information? Merely observing a change in the environment after an action doesn't
              ↪ necessarily indicate progress. Determine if the information gained helps reduce uncertainty
2891          ↪ related to achieving the current subgoal or aligns with the overall game objective, given by
              ↪ 'survival_plan'.\n2.  **Exploration Strategy:** Is the agent's exploration strategy
2892          ↪ sufficient to overcome potential local minima or deceptive landscapes? Many \"stuck\"
              ↪ situations arise from poor exploration. Consider if the agent's 'action_sequence' explores
2893          ↪ diverse enough actions, or if it is repeating the same action variations in a small
              ↪ area.\n3.  **World Model Limitations:** Assess whether the agent's internal model of the
2894          ↪ game world (implied by its actions and reactions in 'action_obs_seq' and 'history') is
              ↪ incomplete or inaccurate. Is it missing key information about the game mechanics, object
2895          ↪ interactions, or environmental dynamics? If so, suggest avenues for improving the world
              ↪ model, such as targeted exploration or observation of specific events.\n4. **Leverage
2896          ↪ entries_successful_goal and entries_failed_goal:** Look into similar subgoals in the past to
              ↪ help you understand how the agent behaved then and compare that with the current
2897          ↪ behavior.\n\n**Your Task:**\n\nBased on the provided details and your expertise in Game AI,
              ↪ determine if the agent is genuinely stuck, meaning it's unlikely to achieve its subgoal
2898          ↪ without external intervention. Focus on *why* the agent is stuck. Specifically, is the
              ↪ agent's failure due to:\n\n*   A lack of crucial information that could be obtained through
2899          ↪ more effective exploration?\n*   An inaccurate or incomplete world model preventing it from
              ↪ making informed decisions?\n*   A fundamental flaw in its action selection
2900          ↪ strategy?\n\nProvide a concise justification for your conclusion, outlining the specific
              ↪ factors that support your assessment. Prioritize identifying concrete steps the agent could
2901          ↪ take to overcome the \"stuck\" state, considering the limited information it may possess.
              ↪ Avoid vague statements and focus on actionable recommendations rooted in Game AI best
2902          ↪ practices."
              ],
2903          "termination": [
2904            1,
              "You are an important termination evaluator component of a hierarchical video game system,
2905          ↪ functioning as a specialized AI reinforcement learning analyst.\n\nYour task is to:\n\n1.
              ↪ Determine whether the agent has met the termination condition for a subgoal.\n\n2. Provide a
2906          ↪ concise summary that will help guide the lower-level agent's future actions, specifically
              ↪ addressing potential issues related to reinforcement learning
2907          ↪ strategies.\n\nDetails:\n\nHere is your current state that you should compare with the
              ↪ termination condition: {obs}\n\nHere is the subgoal the lower-level agent is working
2908          ↪ towards: {subgoal}\n\nHere is the termination condition of the above subgoal given by the
              ↪ higher-level agent: {success_condition}\n\nInstructions:\n\nAnalyze the subgoal and its
2909          ↪ termination condition and decide if the subgoal is completed. Then, using the
2915
```

```
      ↪ action-observation sequence: {action_obs_seq}, provide a high-level summary of your current
      ↪ evaluation.\n\nIn your summary, consider the following reinforcement learning concepts:\n\n*
      ↪    **Exploration vs. Exploitation:** Is the agent adequately exploring the environment, or is
      ↪ it prematurely exploiting a sub-optimal strategy? If the agent has been repeating the same
      ↪ actions in similar states (as observed in {history}) without success, encourage more
      ↪ exploration.\n\n*   **Reward Shaping:** Consider how the higher-level agent's reward
      ↪ structure (implicit or explicit) might be influencing the agent's behavior. Is the agent
      ↪ being incentivized towards unintended consequences?\n\n   **Local Optima:** Is the agent
      ↪ potentially stuck in a local optimum? Compare the current situation with
      ↪ {entries_failed_goal} to identify if similar action sequences have previously led to
      ↪ failure. If so, suggest a drastic change in strategy.\n\n*   **Credit Assignment:** How can
      ↪ the agent better understand which actions are contributing to success or failure? Suggest
      ↪ focusing on the most recent actions, especially in light of {observation_sequence}.\n\nIf
      ↪ the agent has failed similar subgoals in the past, provide an alternative approach based on
      ↪ {entries_successful_goal}. Highlight alternative actions it can take in a similar
      ↪ situation.\n\nRemember that your summary will be passed to the low-level component to
      ↪ improve its actions. It should be actionable and specific. It should be in simple language
      ↪ for the low-level agent to understand and must contain suggestions to avoid common RL
      ↪ pitfalls based on the information provided. The survival plan is {survival_plan} for high
      ↪ level goals."
    ],
    "summariser": [
      1,
      "You are a critic module analyzing an agent's attempt to achieve a subgoal in a game
      ↪ environment, considering the agent's survival plan.\n\nYour task is to identify the **single
      ↪ most important factor** that caused SUCCESS or FAILURE, given the broader context of
      ↪ survival.\n\nInformation:\n- Target subgoal: {subgoal}\n- Outcome of the action-observation
      ↪ sequence: {outcome}\n- Action-observation history: {action_obs_seq}\n- Game context:
      ↪ {game_info}\n- Agent's Survival Plan: {survival_plan}\n- Success Condition:
      ↪ {success_condition}\n\nInstructions:\n- First, read the survival plan and understand the key
      ↪ threats and resource priorities.\n- Then, analyze the action-observation sequence in the
      ↪ context of the target subgoal and survival plan.\n- Focus on **specific resources and their
      ↪ quantities** that were critical according to the survival plan.  Consider if failing to meet
      ↪ {success_condition} resulted from a resource shortfall, specifically referencing the
      ↪ quantities mentioned in the action-observation history.\n- How did the agent's actions
      ↪ either help or hinder the broader survival strategy defined in {survival_plan}?\n- If
      ↪ resources were missing that were crucial to survival, state **exactly which resources and
      ↪ how many** were missing and how it violated the survival plan.\n- If no resources are the
      ↪ primary issue, state the next most relevant factor that impacted both the subgoal and
      ↪ survival chance.\n- Express the result in **one short sentence** highlighting the connection
      ↪ to {survival_plan}."
    ],
    "explorer": [
      0,
      "\n        Task: Create an exploration plan to help the agent discover new skills.\n\n
      ↪ Data:\n        - Game info: {game_info}\n        - Observation: {obs}\n        - Subgoal
      ↪ summary: {summary}\n        - Recent 16 action\u2013observation pairs: {history}\n\n
      ↪ Steps:\n        1. Analyze the environment and agent\u2019s situation.\n        2. Propose a
      ↪ focused exploration plan with clear purpose, conditions, and indicators.\n\n    "
    ],
    "sequential": [
      0,
      "You always have to output one of the above actions at a time and no other text. You always
      ↪ have to output an action until the episode terminates."
    ]
  },
  "fitness": 0.8527708222454596,
  "id": "aaac4dfb-4a3c-4090-a8bc-5f9265d65eda",
  "_std_error": 0.4739428639198163
}
```

