# OpenReview forum: "TAME the BALROG: Task-Adaptative Modular Emergent framework for Game Agents"
_ICLR.cc/2026/Conference — Submitted to ICLR 2026_

### Official Review · Reviewer_dxNh · 2025-10-29

**Soundness:** 3
**Presentation:** 2
**Contribution:** 3
**Rating:** 6
**Confidence:** 3

**Summary:**

This work proposes TAME, a framework that uses a genetic algorithm to evolve modular architecture for LLM agents in gaming environments. Each agent (member) is equipped with varying number of activated modules and the population of agents evolves to keep the best ones. The experiment results on BALROG benchmark show that TAME outperform BALROG's original method on various tasks.

**Strengths:**

1. Using genetic algorithms for dynamic configuration of agent architectures is interesting.
2. The experiments show significant improvements of TAME's performances on various gaming tasks.

**Weaknesses:**

1. The writing is hard to follow. Many terms are interchangeably used in different contexts (like "modole", "framework").
2. Lacking crucial implementation details for the genetic algorithm, such as the precise encoding of the genome, the specific mechanics of genetic operators (selection, crossover, mutation), and the calculation of diversity metrics.
3. TAME framework resembles "evolutionary hyperparameter optimization", but it regards the number and composition of modules of an agent as part of the parameters. This core idea is just using more trails (agents with differnt genomes) to find the "optimal hyperparameter setting". It seems to be resource-heavy and difficult to apply in production.

**Questions:**

1. How does this work's core idea differ from existing relevant work? The authors are suggested to provide detailed comparison and connection between this work and existing work in Section 2 (Related Work) to better position their novelty.
2. What are the details of evolutionary algorithm?
3. The authors are suggested to present the workflow of TAME framework more clearly in Figure 2. It would be better to show the initial population where each member is one agent with different modules, and how the population and parameters evolves in each generation.
4. The authors are suggested to polish the writing to avoid ambiguity, particularly by distinguishing a single agent’s framework and modular architecture from TAME’s evolutionary framework. In many places, these two concepts are used interchangeably.

---

> ### Author Response · Authors · 2025-11-27
> **Response to Reviewer dxNh (1/2)**
>
> We thank the reviewer for their time and valuable feedback. Please find our responses below:
>
> ### Presentation & Clarity
>
> Following your feedback, we consolidated the naming of framework when referring to TAME and module only when referring to individual components, e.g., memory module, critic, etc. We also updated the introduction, related work and conclusion and added more detailed references to the appendix through the experiments section to improve clarity following the feedback from other reviewers.
>
> ### Missing Implementation Details for the Genetic Algorithm
>
> Due to space constraints, these details are deferred to Appendix H and D. In the updated version we included a more explicit reference to the Appendices in Section 3.2 when presenting the genetic approach. In Appendix H we confirm the implementation specifics for the genetic algorithm, including parameters (population size, number of generations, children per generation), the mechanics of genetic operation, prompt operations, parent selection, diversity measures  as well as pseudo code. We also provide the exact prompts for each module detailed in the Appendix Q, and final genomes in the Appendix R.
>
> We are happy to add any additional details if required.
>
> ### “This core idea [of TAME] is just using more trails (agents with different genomes) to find the "optimal hyperparameter setting""
>
> We appreciate that TAME resembles at the meta-level to evolutionary hyperparameter optimisation (EHO). While TAME leverages a genetic algorithm at the meta-level, similar to other EHO approaches, the core distinction lies in the nature and complexity of the elements being optimised. EHO typically optimises a fixed set of numerical hyperparameters (e.g., learning rate, batch size, …) or the topology and weights of a neural network operating within a numeric or sub-symbolic computational domain. In contrast, TAME operates on a higher level, evolving agentic structure and behavior at LLM level. The TAME genome does not represent a parameter vector; instead, it encodes the blueprint of the architecture itself, specifying which high-level modules (like Explorer, Critic, Loop Detector,...) exist, their parameters and the semantic instructions that govern their behavior, including prompt and input. Because these modules are specified using natural-language prompts, the search operates over a mixed symbolic–semantic design space rather than a numeric or sub-symbolic one, with both continuous and discrete genes. As a result, mutations and crossovers can alter entire functional components, routing logic, or prompt-level behavioral specifications, leading to qualitatively different agent behaviors that adapt their functional components to highly variable environments.
>
> We revised the related work section to address the reviewer’s comment.
>
> ### “It seems to be resource-heavy and difficult to apply in production.”
>
> We understand the reviewer's concern regarding the efficiency of TAME given that we use an evolutionary approach. We use a genetic algorithm as they enable parallelisation across hardware. Crucially our results evidence that with a very restricted evolutionary budget (4 generations and 5 children only) we obtain 25.27% (see Table 9 in the Appendix K.4) and 37.10% (see Table 17 in Appendix L.3) relative performance improvements with Gemini-Flash-2.0 and GPT4.1-nano, respectively.
> Moreover, we present a novel long-term memory approach that matches the performance of SOTA method A-Mem, while requiring three times less LLM calls, with the corresponding benefits in  compute time and cost from requiring less instructions with the LLM.
>
> [1] Xu, Wujiang, et al. "A-mem: Agentic memory for llm agents.". 39th Conference on Neural Information Processing Systems, 2025

---

> > ### Author Response · Authors · 2025-11-27
> > **Response to Reviewer dxNh (2/2)**
> >
> > For your questions:
> >
> > **Q1: How does this work's core idea differ from existing relevant work? The authors are suggested to provide detailed comparison and connection between this work and existing work in Section 2 (Related Work) to better position their novelty.**
> >
> > Following similar comments of Reviewers 5 and 3 we included additional related works in Section 2 and clarified on novelty:
> > While evolutionary algorithms have been explored for agent optimisation, our work addresses a distinct problem space.
> > AFLOW [1] and ADAS [2] focus on optimising agent workflows—the sequential flow and coordination of processing steps. TAME operates on a fixed workflow architecture (see  Figure 2) and instead optimises individual module implementations and configurations. These approaches are complementary but address different aspects of agent design.
> > EvoAgent [4] and MaAS [5] use evolutionary search to construct multi-agent systems, where multiple distinct agents are coordinated with specialised roles and inter-agent communication patterns. TAME is fundamentally a single-agent system that evolves the internal implementations of fixed functional modules (memory, planning, action) within one agent. Multi-agent optimisation focuses on role distribution and collaborative strategies, while TAME focuses on intra-agent module optimisation for gaming environments.
> >
> > Last, AgentSquare [3] similarly proposes a single-agent framework like TAME but with several key shortcomings over our approach. AgentSquare only allows evolution through a predefined set of modules  (planning, reasoning, tool use, memory), and requires the user to manually define the tools to use for every new domain. TAME includes modules for planning, reasoning, tool use and memory along with exploration, hierarchical reasoning, loop detection, critic feedback and survival, which as can be noted from Figure 15 in the Appendix I, all play a critical role in at least one game. Moreover, TAME doesn’t require any user input to adapt to different games. Additionally, AgentSquare memory module uses the same embedding-based memory system that we demonstrated in Table 3 is less effective than TAME’s proposed long-term memory system.
> >
> >
> > [1] Zhang J, Xiang J, Yu Z, et al. Aflow: Automating agentic workflow generation[J]. arXiv preprint arXiv:2410.10762, 2024.
> > [2] Hu S, Lu C, Clune J. Automated design of agentic systems[J]. arXiv preprint arXiv:2408.08435, 2024.
> > [3] Shang Y, Li Y, Zhao K, et al. Agentsquare: Automatic llm agent search in modular design space[J]. arXiv preprint arXiv:2410.06153, 2024.
> > [4]Yuan S, Song K, Chen J, et al. Evoagent: Towards automatic multi-agent generation via evolutionary algorithms[J]. arXiv preprint arXiv:2406.14228, 2024.
> > [5] Zhang G, Niu L, Fang J, et al. Multi-agent architecture search via agentic supernet[J]. arXiv preprint arXiv:2502.04180, 2025.
> >
> > **Q2: What are the details of the evolutionary algorithm?**
> >
> > As noted in our response above, details of the genetic algorithm can be found in Appendix H. We modified the main text to more clearly guide the reader to this information.
> >
> > **Q3: The authors are suggested to present the workflow of TAME framework more clearly in Figure 2. It would be better to show the initial population where each member is one agent with different modules, and how the population and parameters evolves in each generation.**
> >
> > Figure 2 is intentionally designed to show the full capabilities of the agent and how the different modules interact with each other. We believe Figure 1(TAME Framework Overview) already captures the evolutionary workflow: showing the Pool of Modules, the creation of the initial population ($P_0$), the iterative GAME LOOP leading to Selection & Cross Breeding ($P_i \to P_{i+1}$), and the final result (FINAL GENOME).
> > To address the reviewer's concern, we modified the captions and accompanying text for both figures to clearly articulate their focus. We also revised Figure 1's description to more clearly indicate that $P_0$, $P_i$, and $P_{i+1}$ are populations of diverse agents (genomes), each having a different subset of modules from the pool.
> > Please let us know if you have any additional questions about Figure 1.
> >
> > **Q4: The authors are suggested to polish the writing to avoid ambiguity, particularly by distinguishing a single agent’s framework and modular architecture from TAME’s evolutionary framework. In many places, these two concepts are used interchangeably.**
> >
> > As per our comment above, we addressed the naming issue in our revised version.
> >
> > Thank you again for your valuable feedback. Please let us know if you have any further questions or concerns.

---

### Official Review · Reviewer_mETx · 2025-10-30

**Soundness:** 3
**Presentation:** 2
**Contribution:** 3
**Rating:** 4
**Confidence:** 4

**Summary:**

The authors present a new evolutionary method for augmenting the capabilities of LLMs to play videogames in the BALROG benchmark. The authors define a set of high-level modules (e.g. long-term memory or explicit exploration) and use an evolutionary approach to select which modules are deployed (along with relevant hyperparameters and modifications to the prompt) for each game in the benchmark. The authors demonstrate that this approach significantly improves the performance of the Gemini 2.0-Flash model. In addition, the authors show that adapting the best genotype found for the Gemini 2.0-Flash model to other models results in zero-shot performance gains, including a new SOTA for the BALROG benchmark.

**Strengths:**

The core method of the paper (evolutionary optimization to determine which modules are most appropriate for a given game) seems both novel and reasonable. The paper is also quite thorough, with a variety of ablations and hyperparameters -- I feel that the reproducibility of the experiments is high. Barring the caveats described below, I think the impact of the paper could also be high.

**Weaknesses:**

My primary concern with this paper is in the comparisons to baselines. The performance gains are indeed impressive, but at present it’s somewhat difficult to tell how much of the improvement is attributable to the evolutionary search procedure and how much is the result of the various modules simply causing the LLMs to “reason more” than the baseline prompts. The prompts in the original BALROG paper appear to be quite simple (i.e. just stating that the LLM is a player and enumerating the valid actions). I think the paper would benefit from an additional baseline which introduces more reasoning but perhaps without the fully decomposed module structure (or an explanation of why the original BALROG prompts act as a fair baseline).

I also think that the clarity of the paper could also be improved. There are a few technical terms which are used but not introduced (e.g. “wheel selection” or options). I also found Table 1 confusing at first -- the “Full Pop. Score” column seems like it could be referring to the gains of the whole TAME population over the baseline LLM instead of the gain of the TAME + adaptation model over the TAME[full] model, since “full” is a somewhat overloaded term. I also think it’s more common to state the performance gain in terms of percentage points (i.e. 34.7 - 27.2 = 7.5%) as opposed to percent improvement (i.e. 34.7/27.2 ~= 1.28). Relatedly, it’s not clear if the “+12.18%” gain of TAME+adaptation over TAME[full] is a raw percentage point increase or another percent improvement measure and it would be good to clarify (perhaps by simply including the raw performance of the TAME[full] model).

While these points affect my rating, I would be happy to increase my score if they are addressed.

**Questions:**

- How much of the gain in performance over the BALROG baseline is attributable to more reasoning or longer prompts as opposed to the specific modules selected?
- Line 240: what is “wheel selection”?
- There are two different citations to Eureka -- (Ma et al. 2023) and (Ma et al. 2024)

---

> ### Author Response · Authors · 2025-11-27
> **Response to Reviewer mETx (1/2)**
>
> We thank the Reviewer for their kind and constructive comments and for taking the time to review our paper.
>
> ### “How much of the improvement is attributable to the evolutionary search procedure and how much is the result of the various modules simply causing the LLMs to “reason more”
>
> Table 9 demonstrated in the Appendix K.4 reports the performance improvement over initial population achieved by the evolutionary algorithm. When compared to the best scoring genomes from the initial population, we obtained a 25.27% ± 3.05% average relative progression improvement across the 6 games using Gemini-2.0-Flash. We agree with the reviewer that in the previous version this was not presented clearly. In Tables 8 and 16 in the Appendix, we demonstrate performances of initial population, showing scores of different types of module combinations and different sets of prompts. We also have significantly revised our terminology: (the initial naive architecture from Balrog is still “Baseline”, hierarchical structure using the full llm-generated prompt as “Full[Claude]” and a human crafted prompt as “Full [Hand-crafted]”).
>
> Moreover, in the revised version Tables 6 and 14 in the Appendix we present comparisons of Baseline structure (BALROG), ImprovedPrompt[Claude] and TAME with an optimised set of modules. We find that while the “ImprovedPrompt[Claude]” prompt improves the performance over “Baseline” prompt in some games, it hurts the performance in others. This aligns with our earlier findings that made us pivot towards the evolutionary adaptation approach for TAME. That is, most changes made to benefit performance in one game hurts the performance in other games, and some prompt changes that helped one LLM model would require a human prompting again for a newer model.  Instead, TAME removes this problem by adapting the agent (modules, their inputs, hyperparameters and prompts) to both the game and model.
>
> ### Clarity & Presentation
>
> Thank you for pointing out the presentation and clarity issues. We changed our wordings as per your comments.
>
> By “wheel selection” we meant roulette wheel selection which is a common technique used in Evolutionary Algorithms for selecting individuals (parents) from the current population to create the next generation. It operates like a roulette wheel where the selection probability for each individual is proportional to its fitness score. Fitter individuals are assigned larger "slices" on the wheel, giving them a higher chance of being selected, while still ensuring that all individuals have a non-zero probability of being chosen.
>
> Formally, the probability $P(g_i)$  of selecting an individual genome $\mathbf{g}_i$ is calculated as:
>
> $$ P(g_i) = \frac{f(g_i)}{\sum_{j=1}^{|P|} f(g_j)} $$
>
> where $f(\mathbf{g}_i)$ is the fitness score of genome $\mathbf{g}_i$ and $|\mathcal{P}|$ is the total size of the population.
>
> We modified the paper to clarify it in Section 3.2 and we present the mathematical formulation in the Appendix H.
>
> Regarding confusing metrics in Table 1, we removed the last column and only left the main results. Instead, as suggested, in Table 9 of the Appendix K.4 we present the average relative improvement over the initial population, showing improvement of TAME framework. We also make sure to clarify whether we mean relative percentage improvement or raw game score improvement where necessary.

---

> ### Author Response · Authors · 2025-11-27
> **Response to Reviewer mETx (2/2)**
>
> ### For your questions:
>
> **Q1: How much of the gain in performance over the BALROG baseline is attributable to more reasoning or longer prompts as opposed to the specific modules selected?**
>
> We refer here to our detailed response on baseline comparisons in the above section. In summary, to isolate the effect of "more reasoning" versus the evolutionary search, we introduced a strong 'ImprovedPrompt[Claude]' prompt baseline (engineered by Claude to maximise reasoning without TAME). While this ImprovedPrompt[Claude]' prompt improves on certain environments (e.g. BabyAI with Gemini, see Table 6 in Appendix K.1), TAME consistently delivers higher relative performance by 50% when compared to ImprovedPrompt[Claude] using Gemini-2.0-Flash (see Table 6 in Appendix K.1) and by 182% when using GPT4.1-nano (see Table 14 in Appendix L.1). This confirms that the optimisation of specific module selection and structure via evolution is the main reason for the performance gain, not just the inclusion of more reasoning steps.
>
> **Q2: Line 240: what is “wheel selection”?**
>
> As stated above, we clarified “wheel selection” to “roulette wheel selection” and briefly described in Section 3.2 of the Tame Framework and pointed to the mathematical formulation in Appendix H.
>
> **Q3: There are two different citations to Eureka -- (Ma et al. 2023) and (Ma et al. 2024)**
>
> Thank you for spotting the double citation of the Eureka paper. We have removed the redundant entry from the references in the revised manuscript.
>
> We believe we addressed all the points the reviewer raised. Please let us know if there are any other standing concerns.

---

### Official Review · Reviewer_s3bn · 2025-11-01

**Soundness:** 3
**Presentation:** 3
**Contribution:** 2
**Rating:** 2
**Confidence:** 4

**Summary:**

This paper addresses the poor performance of Large Language Models in complex, interactive gaming environments, such as those in the BALROG benchmark. The authors introduce the Task-Adaptive Modular Emergence framework, which employs a genetic algorithm to automatically discover effective, environment-specific agentic structures. TAME evolves a genome that specifies which human-designed modules to activate, their hyperparameters, and their prompts. The core contributions are: (1) the TAME framework itself; (2) a novel, efficient long-term memory system; (3) achieving SOTA performance on the BALROG benchmark by improving a baseline Gemini 2.0-Flash score from 27.15% to 34.77% ; and (4) demonstrating that these evolved structures are transferable, enabling a Gemini 2.5-Pro model to achieve a new SOTA score (47.65%).

**Strengths:**

1. The paper is well-organized and uses easy-to-understand language.

2. The paper provides rich implementation details in the appendix, which is commendable and crucial for reproducibility.

3. Rigorous ablation and analysis.

**Weaknesses:**

1. Limited novelty: Evolutionary algorithms have already been applied in the field of agent optimization/search, for example, in AgentSquare[3] and EvoAgent[4].

2. Poor scope/generalizability of the framework: The framework proposed in this paper is applied to the domain of interactive games. In contrast, related agent optimization/search works, such as Aflow[2] and MaAS[5], can be applied across multiple domains.

3. Insufficient discussion of related work: The core ideas of this paper, including agent evolution, evolutionary algorithms, merge components, and mutation modules, are all highly related to works like ADAS[1], AFLOW[2], AgentSquare[3], and MaAS[5], yet the paper does not discuss them.

4. Limited experiments: The paper is only evaluated on the BALROG benchmark. To my knowledge, other benchmarks for interactive games exist, such as Minecraft.

Reference

[1]Zhang J, Xiang J, Yu Z, et al. Aflow: Automating agentic workflow generation[J]. arXiv preprint arXiv:2410.10762, 2024.

[2]Hu S, Lu C, Clune J. Automated design of agentic systems[J]. arXiv preprint arXiv:2408.08435, 2024.

[3]Shang Y, Li Y, Zhao K, et al. Agentsquare: Automatic llm agent search in modular design space[J]. arXiv preprint arXiv:2410.06153, 2024.

[4]Yuan S, Song K, Chen J, et al. Evoagent: Towards automatic multi-agent generation via evolutionary algorithms[J]. arXiv preprint arXiv:2406.14228, 2024.

[5]Zhang G, Niu L, Fang J, et al. Multi-agent architecture search via agentic supernet[J]. arXiv preprint arXiv:2502.04180, 2025.

**Questions:**

1. Can experiments be conducted on other benchmarks?

2. Can experiments be conducted on models other than the Gemini series (e.g., the GPT series, open-source models)?

3. Can the authors provide a detailed explanation of the differences from related work?

---

> ### Author Response · Authors · 2025-11-27
> **Response to the Reviewer s3bn (1/2)**
>
> We thank the reviewer for the feedback they provided on our submission.
>
> ### Limited novelty: Evolutionary algorithms have already been applied in the field of agent optimization/search, for example, in AgentSquare[3] and EvoAgent[4].
>
> We agree with the reviewer that evolutionary approaches have proven successful in agentic optimisation and, as highlighted in the original manuscript, Eureka [1], a framework that employs evolutionary algorithms, was a key inspiration of our framework. However, we believe our work has multiple meaningful novel components as described in the updated contributions paragraph and that we add here:
>
> (1) We introduce TAME, the first emergent game-agentic framework that enables LLMs to evolve modular structures tailored to gaming environments and achieve state-of-the-art performance in the BALROG benchmark; (2) We present a genetic approach key for the functioning of this framework, introducing a set of modules for general gaming capabilities and a genome that captures modules, hyperparameters and prompts enabling TAME to adapt the agent to the game with a low evolutionary budget (4 generations and 5 children); (3) We demonstrate that TAME can be used to find effective agentic configurations with smaller models that directly transfer and improve the performance of larger and reasoning models of the same family; (4) We propose a novel and effective long-term memory system that combines embedding-based retrieval with LLM-augmented semantic memory matching the performance of state-of-the-art memory methods requiring three times less LLM calls.
>
> [1] Ma, Yecheng Jason, et al. ‘Eureka: Human-Level Reward Design via Coding Large Language Models’. The Twelfth International Conference on Learning Representations, 2024
>
>
> ### Scope & Generalization
>
> We present an agentic framework designed to play and adapt to any game -provided we can transform the game input to text-, as we noticed a significant gap in agentic frameworks applied to gaming environments. As we detail below, related works like Aflow and MaAS   are designed to solve very different kinds of problems and are not applicable here. Games, alongside maths, test comprehension and logic, are crucial benchmarks for AI models as they test different objectives such as: survival skills, long-term memory, management of resources, survival, exploration and long-horizon skill decomposition. Other tasks and frameworks, while still important, are oblivious to many of these skills.
>
>
> ### Related Work
>
> While evolutionary algorithms have been explored for agent optimisation, our work addresses a distinct problem space:
> AFLOW and ADAS focus on optimising agent workflows—the sequential flow and coordination of processing steps. TAME operates on a fixed workflow architecture (Figure 2) and instead optimises individual module implementations and configurations. These approaches are complementary but address different aspects of agent design.
> EvoAgent and MaAS use evolutionary search to construct multi-agent systems, where multiple distinct agents are coordinated with specialised roles and inter-agent communication patterns. TAME is fundamentally a single-agent system that evolves the internal implementations of fixed functional modules (memory, planning, action) within one agent. Multi-agent optimisation focuses on role distribution and collaborative strategies, while TAME focuses on intra-agent module optimisation for gaming environments.
> Last, AgentSquare similarly proposes a single-agent framework like TAME but with several key shortcomings over our approach. AgentSquare only allows evolution through a predefined set of modules  (planning, reasoning, tool use, memory), and requires the user to manually define the tools to use for every application. TAME includes modules for planning, reasoning, tool use and memory along with exploration, hierarchical reasoning, loop detection, critic feedback and survival. Notably, all the proposed modules play a critical role in at least one game, as can be noted from Figure 15 in the Appendix I. Moreover, TAME doesn’t require the user to craft new modules/tools to adapt to different games. Additionally, AgentSquare memory module uses the same embedding-based memory system that we demonstrated in Table 3 is less effective than TAME’s proposed long-term memory system.
> Nevertheless, we agree with the reviewer that these works are relevant related literature and expanded the existing paragraph on Agentic framework’s related literature to include all of them.

---

> ### Author Response · Authors · 2025-11-27
> **Response to the Reviewer s3bn (2/2)**
>
> ### The paper is only evaluated on the BALROG benchmark. To my knowledge, other benchmarks for interactive games exist, such as Minecraft.
>
> We want to clarify that BALROG is not a game (like Minecraft), but a challenging benchmark that gathers together six very different games (BabyAI, TextWorld, Crafter, Baba Is AI, MiniHack, NLE) of varying difficulty -from BabyAI (takes seconds to learn by humans) to Nethack (years)- and allows us to test both basic competence and more advanced long-horizon capabilities. Moreover, one of them (Crafter) is very similar to the game Minecraft. We also highlight that BALROG includes NLE (Nethack), a game where, different from Minecraft[1], LLM agents still struggle to make progress [2]. Our experiments show how TAME gets frontier models like Gemini flash 2.0 from not making any progress on the game to become a top 10 model, beating even newer thinking versions like Gemini flash 2.5 [3].
>
> [1] Wang G, Xie Y, Jiang Y, et al. Voyager: An Open-Ended Embodied Agent with Large Language Models. Transactions on Machine Learning Research. Published online 2024.
> [2] Paglieri, Davide, et al. "Balrog: Benchmarking agentic llm and vlm reasoning on games." The Thirteenth International Conference on Learning Representations, 2025
> [3] BALROG. balrogai.com. Accessed on the 20/11/2025.
>
>
> ### For your questions:
> **Q1: Can experiments be conducted on other benchmarks?**
> As mentioned above, BALROG is already a diverse and well acclaimed benchmark for testing LLMs on a wide range of games. Therefore, we will not be including additional benchmarks.
>
> **Q2: Can experiments be conducted on models other than the Gemini series (e.g., the GPT series, open-source models)?**
> Yes, the updated version now includes experiments both for learning and transferability with GPT models showing the good performance of TAME in that family of models as well.
>
> **Q3: Can the authors provide a detailed explanation of the differences from related work?**
> We provided the detailed explanation above and updated the paper accordingly. Please let us know if you have any further concerns.
>
> Thank you again for your time. Please let us know if you have any further concerns or questions.

---

> > ### Comment · Reviewer_s3bn · 2025-11-28
> >
> > I thank the authors for their detailed response, which has successfully addressed my concerns. I appreciate the effort the authors have put into the rebuttal, particularly in clarifying the distinctions from related works and providing additional context.
> >
> > After carefully reviewing the authors' response, re-examining the related literature, and considering the comments from other reviewers, I have decided to raise my score to 6. Additionally, I will lower my confidence score to reflect my reassessed position.

---

> > > ### Author Response · Authors · 2025-11-28
> > > **Thank you and score/confidence failed to update**
> > >
> > > We want to thank the Reviewer for rapidly engaging with our work again. Their feedback was very helpful to strengthen the paper.
> > >
> > > We also want to note that while the Reviewer's scores for the contribution were updated (soundness, presentation and contribution), the Rating and Confidence scores failed to update to the new values stated in their response.

---

### Official Review · Reviewer_NH6Y · 2025-11-05

**Soundness:** 3
**Presentation:** 3
**Contribution:** 2
**Rating:** 6
**Confidence:** 4

**Summary:**

This paper presents TAME (Task-Adaptive Modular Emergence), a framework based on genetic algorithms that evolves modular architectures for large language model (LLM) agents in interactive games. Each agent configuration comprises module combinations, hyperparameters and prompts, which are optimised through evolutionary selection and mutation. When evaluated on the BALROG benchmark, TAME was found to significantly improve LLM performance (e.g. Gemini-2.0-Flash: 27.16% to 34.78%), and demonstrate cross-model transferability. This study contributes a novel task-adaptive agent framework and an efficient long-term memory design, as well as providing empirical evidence that modular adaptation can enhance LLMs' reasoning and planning abilities in complex environments.

**Strengths:**

S1. The paper introduces a modular emergence framework driven by genetic algorithms that automatically configures LLM agent architectures based on task environments. This innovative approach combines evolutionary search, modular agent design, and prompt optimisation, offering a novel way to extend LLM adaptation beyond prompt tuning or static scaffolds.
S2. The paper presents a thorough analysis of experiments conducted on the BALROG benchmark, demonstrating consistent enhancements across various game environments and models. Demonstrating cross-model transferability, where evolved architectures generalise from Gemini-2.0-Flash to Gemini-2.5-Pro, supports the robustness and practical value of the approach.
S3. The paper clearly articulates the design of each module and the evolutionary process, supported by well-structured figures and ablation studies. Its findings make a significant contribution to the emerging field of LLM-based agent architecture search by offering insights into how adaptive structural configurations can improve reasoning and planning in dynamic environments.

**Weaknesses:**

W1. Limited conceptual novelty: Although the paper presents TAME as an emergent intelligence framework, its core mechanism essentially involves evolutionary search over prompt and module configurations without introducing any new fundamental learning principles. Therefore, the originality lies more in system integration than in theoretical advancement.

W2. Dependence on hand-crafted modules and uncertain generalisation: TAME' s adaptive capacity is limited by a modular library designed by humans. Each component (e.g. memory, exploration and the amygdala) contains pre-defined functional assumptions. Consequently, its success may hinge on the designer's prior knowledge and understanding of the domain. While this approach is effective in game-based environments, it remains unclear whether such a manually curated framework can be applied to other games and other reasoning or planning domains beyond games.

W3. Evaluation metrics focus narrowly on progression scores: The empirical evaluation primarily relies on game progression percentages as the fitness signal. While this metric captures task success, it does not capture reasoning quality, sample efficiency or adaptation dynamics. Consequently, it is unclear whether the observed performance gains reflect genuine improvement in reasoning or merely the exploitation of heuristic patterns.

**Questions:**

Q1. The paper frequently refers to 'modular emergence'. How do the authors formally define or measure emergence in this context, besides the performance improvements discovered through genetic search?
Q2. What is the approximate computing budget required for one evolutionary run on BALROG? Are there any efficiency-improving mechanisms, such as early stopping, surrogate evaluation or population pruning?
Q3. Could the authors provide qualitative or behavioural analyses demonstrating why certain module combinations outperform others?
Q4. What were the reasons for the genetic algorithm being selected over other architecture or prompt optimisation approaches?
Q5. Will the authors release the full codebase? And the configuration files?

---

> ### Author Response · Authors · 2025-11-27
> **Response to Reviewer NH6Y (1/2)**
>
> We appreciate the reviewer's interest in our work and their valuable time and feedback in their review. Our detailed responses are provided below:
>
> ### Limited Novelty
>
> We believe our work is highly novel. We introduce a novel agentic framework that, to the best of our knowledge, is the first one designed to effectively adapt to different games, propose a new and improved long-term memory mechanism which uniquely synthesises the merits of prior state-of-the-art methods, and are the first to put forth a set of general gaming skills for game agents that are proved to work well with different models and in diverse games. We add an extra paragraph - Agentic frameworks- in the Related Works section in order to position the novelty of our paper better.
>
>
> ### Dependence on hand-crafted modules and uncertain generalisation
>
> Literature has often shown that providing human priors can significantly improve the ability of agentic frameworks [1,2]. Moreover, domain-specific methods outperform general approaches with limited computational budgets (for example in code generation [3], medical [4], finance [5] and legal domains [6]). Thus, as part of our contributions we provide a set of human-crafted priors that, as demonstrated in the experiments, helps the framework to find relevant configurations within a limited number of genome generations. Future work could explore hybrid approaches that combine our method with automatic module generation and workflow optimisation techniques. We have updated the introduction and conclusions to better highlight these points.
>
> [1] Ma, Yecheng Jason, et al. ‘Eureka: Human-Level Reward Design via Coding Large Language Models’. The Twelfth International Conference on Learning Representations, 2024
> [2] Klissarov, M., et al. MaestroMotif: Skill Design from Artificial Intelligence Feedback. 2025.
> [3] Guo, Daya, et al. "DeepSeek-Coder: When the Large Language Model Meets Programming--The Rise of Code Intelligence." arXiv preprint arXiv:2401.14196 (2024).
> [4] Wang, Zifeng, et al. "A perspective for adapting generalist ai to specialized medical ai applications and their challenges." NPJ Digital Medicine 8.1 (2025): 429.
> [5] Lee, Jean, et al. "A survey of large language models in finance (finllms)." arXiv preprint arXiv:2402.02315 (2024).
> [6] Singh, Amrita, et al. "LLMs for Law: Evaluating Legal-Specific LLMs on Contract Understanding." arXiv preprint arXiv:2508.07849 (2025).
>
> ### Evaluation Metrics
>
> We adopted the progression-percentage metric as our primary performance measure to maintain consistency with the benchmark. This metric quantifies task completion success - for instance, whether specific objectives were accomplished or, in NetHack, the probability of winning after reaching a particular level. We used the same metrics as defined in the original benchmark (See our detailed description in Appendices B,C and E).
> Nevertheless, through early research, we created supplementary metrics to evaluate specific framework components, including memory retrieval through episodes (Appendix F) and threat response analysis (Appendix G1). We also tracked the number of steps before loop detection and steps needed to reach achievements with and without the critic. We excluded the loop detector and critic metrics from the Appendix because they were primarily tested only through early stages. We found these metrics gave us development signals, adding more noise than value to the main analysis.

---

> > ### Author Response · Authors · 2025-11-27
> > **Response to Reviewer NH6Y (2/2)**
> >
> > For your questions:
> >
> > **Q1: The paper frequently refers to 'modular emergence'. How do the authors formally define or measure emergence in this context, besides the performance improvements discovered through genetic search?**
> >
> > We employed the term “modular emergence" as an intuition that given the initial human-crafted architecture, the framework will automatically reconfigure which models are active, the input they use, their prompts and hyperparameters to automatically discover effective configurations and prompts for each game. We explicitly quantify measure emergence, instead we measure the diversity of the genomes to ensure the population keeps exploring different while interesting solutions.
> >
> > **Q2: What is the approximate computing budget required for one evolutionary run on BALROG? Are there any efficiency-improving mechanisms, such as early stopping, surrogate evaluation or population pruning?**
> >
> > We evaluate BALROG on 30 cpu cores, fully parallelising where possible. The runtime was mostly influenced by the inference speed of the model and the games that had the longest loops (Nethack). The computing time running the evolutionary algorithm was slightly below two days per run to train the agent on all 6 games, and required about 15B input tokens. In our work we employ population pruning: after each evaluation, we keep 5 best scoring genomes and up to 5 most diverse but scoring at least 70% of the best genome (diversity is measured through embeddings, as most of the genome is represented by the natural language description). Moreover, for NetHack (only during the genetic algorithm), we additionally implement a 1000-step timeout for agents that have not made progress, as we observe that such episodes rarely result in further advancement.
> >
> > **Q3:  Could the authors provide qualitative or behavioural analyses demonstrating why certain module combinations outperform others?**
> >
> > The module combination is highly dependent on the game (which is the idea behind this framework), however we provide an intuition behind why certain modules might be better for certain environments. For example, TextWorld performs best when using baseline structure, as actions that the agent can take are dependent on current observation, therefore an agent can’t plan it ahead and doesn’t require additional modules. In contrast, other environments like BabyAI, BabaIsAI, Crafter and NetHack benefit from hierarchical structure as they require planning. We find that MiniHack benefits from both, however the best genome is returned by baseline structure, possibly because it is a simplified version of NetHack. Similarly, only certain environments benefit from specialised amygdala (survival skills) and explorer (exploration) and in some those are not needed. All environments apart from TextWorld (for the same reasons as above) benefit from long-term memory. We found that, in general, Critic and Loop Detector are beneficial when using hierarchical structure (and not applicable when using baseline), as they add extra information about good/bad decisions (Critic) and prevent from looping behaviour (Loop Detector).
> >
> >
> > **Q4: What were the reasons for the genetic algorithm being selected over other architecture or prompt optimisation approaches?**
> >
> > The reason for choosing a genetic algorithm is the fact that it is easily parallelisable across hardware when evaluating different children. Prompt optimisation is indeed a part of TAME (See Section 3.2, where we highlight now more this point). Architecture optimisation is an interesting line of future work, however in this work we focus on optimisation of modular components. Future work could investigate optimising the connection of flow of information between modules, as well as emergent, new modules.
> >
> > **Q5: Will the authors release the full codebase? And the configuration files?**
> >
> > Yes. We plan to release the whole codebase and configuration files upon acceptance.
> >
> > We appreciate your feedback once more. Please feel free to reach out if you have any additional concerns or questions.

---

### Official Review · Reviewer_eUZK · 2025-11-05

**Soundness:** 2
**Presentation:** 3
**Contribution:** 2
**Rating:** 4
**Confidence:** 4

**Summary:**

This paper presents **TAME (Task-Adaptive Modular Emergence)**, a method that uses a genetic/evolutionary search over a pre-designed module space (e.g., hierarchical planner, explorer, long-term memory, loop-detection) to discover an agent “genome” — a combination of modules, prompts, and hyperparameters — tuned for a given game environment. The approach is evaluated on the BALROG benchmark; the authors report improvements over a baseline LLM (e.g., Gemini-2.0-Flash) and claim a transferred genome produces SOTA results when applied to a stronger model (Gemini-2.5-Pro). The paper also provides ablations on the memory module (vs. Jarvis / A-MEM) and includes evolved genome JSONs in the appendix.

**Strengths:**

1. **The research is practically relevant**. Automating discovery of modular agent architectures for interactive game tasks is a worthwhile goal for agentic LLM research.
2. **The engineering is reasonably complete**, proposing a concrete genome encoding (modules + hyperparams + prompts), evolutionary procedure.
3. The paper **includes targeted ablations** (memory module comparisons) rather than only reporting final aggregate scores, indicating attempts to analyze component contributions.

**Weaknesses:**

1. **Stability and statistical rigor are weak.** The evolutionary search uses very small budgets (ngen = 4, nchild = 5) and some evaluations use few episodes (e.g., NetHack/NLE evaluated with 5 episodes), making results noisy and potentially non-robust. The paper lacks multiple independent runs, confidence intervals, and statistical tests.
2. **The claim of transferability is under-supported.** The transfer experiment that moves a genome from Gemini-2.0 to Gemini-2.5-Pro and claims SOTA lacks tests on models from different families; the result could be specific to closely related models rather than generally transferable. Full leaderboard context and uncertainty estimates are missing.
3. **Missing budget-matched baselines.** The paper does not compare against budget-matched RL agents or other automated search methods (Bayesian Optimization, Population-Based Training, Evolution Strategies) under comparable compute budgets, leaving open whether evolutionary search is the best choice.
4. The paper **claims benefits from long-term memory** for long-horizon tasks, but provides **no qualitative retrieval to decision case study or horizon-sensitive ablation** to show how memory changes behavior.

**Questions:**

1. How does performance scale with search budget? Report results for at least two larger budgets (e.g., ngen ∈ {4, 8, 16} and nchild ∈ {5, 10}) and state whether performance consistently improves, saturates, or degrades.
2. Can you evaluate transfer across model families, applying the same evolved genome to at least one model from a **different family** (e.g., an OpenAI, Anthropic, or an open-source LLM). Report per-environment performance and compare to the original target model.
3. Show per-method learning curves and final performance ± CI under a matched budget. If Genetic Algorithm still performs best, explain WHY (e.g., better parallelism, robustness) rather than relying on single-number superiority.
4. Please include one retrieval→decision trace and a simple ablation by episode length.

---

> ### Author Response · Authors · 2025-11-27
> **Response to Reviewer eUZK (1/2)**
>
> We want to first thank the reviewer for their time and valuable comments. We incorporated your feedback to strengthen our contribution and provide a detailed responses below:
>
> ### Additional independent runs
>
> The reviewer raises a very good point that the experiments would benefit from additional independent runs. We incorporated additional runs and statistical confidence in the updated version for all gemini-based training and transfer experiments. We will update the version with additional runs for the newer GPT experiments shortly.
>
> ### Some evaluations use few episodes (e.g., NetHack/NLE evaluated with 5 episodes)
>
> We use the evaluation runs as in BALROG’s official leaderboard on Nethack. This allows for a fair comparison with the solutions there. We also want to highlight that the games on this  benchmark are of very diverse difficulty and length (a human can understand and play some of the games in seconds, but others require years) and that is why the official leaderboard employs different numbers of evaluations per game.
>
> ### The evolutionary search uses very small budgets (ngen = 4, nchild = 5)
>
> The reviewer rightly points out that our evolutionary budget is low, but as can be appreciated in Table 2 of the updated version, even this small budget yields very strong empirical results, improving the relative score on average by 25.27% +- 3.05% (also see Table 9 in the Appendix K.4 ) over initial population using Gemini-2.0-Flash and by 37.10% +- 2.08% using GPT4.1-nano (see Table 17 in Appendix L.3), highlighting the strengths of our proposed framework.
>
> ### The claim of transferability is under-supported.
>
> This comment from the reviewer was particularly insightful. While our original experiments demonstrated we achieved the best performance of the leaderboard, we agree with the reviewer that it was uncertain if such transferability was due to being models of the same family. Thus, for the updated version we repeated the experiments with GPT 4.1 models, and found that the genomes from one family of models (Gemini) do not transfer well to other models (GPT). Nevertheless, we found that as with gemini models, genomes discovered with a smaller model (GPT-4.1-nano) also improve the relative performance by ~10% (see Table 18 in the Appendix L.4) of a larger model of the same family (GPT-4.1-mini). Above is the updated Table 2 from Section 4.2 of the manuscript. We observe an ~74% relative improvement when comparing TAME against the GPT-4.1-nano baseline (see Table 1). Moreover, genomes transferred from GPT-4.1-nano to GPT-4.1-mini improve the relative score by ~10% (see Appendix  L.4), further strengthening our transferability claims.
>
> ### The paper does not compare against budget-matched RL agents or other automated search methods (Bayesian Optimization, Population-Based Training, Evolution Strategies)
>
> We do compare reasoning agent models with unlimited thinking budgets. In this work, we are presenting the first framework for finding game-focused agentic architectures/scaffolds. We agree that it can be interesting in future work to explore whether other automated search methods can strengthen TAME, but it is out of the scope of this paper. We employed a genetic algorithm due to easy parallelisation across hardware, and for it being a default choice in many state-of-the art works [1, 2, 3]. We included these clarifications in the Evolutionary Strategies paragraph in the Related Work section,
>
> [1] Ma, Yecheng Jason, et al. ‘Eureka: Human-Level Reward Design via Coding Large Language Models’. The Twelfth International Conference on Learning Representations, 2024
> [2] Rosser, J., and Jakob Nicolaus Foerster. "AgentBreeder: Mitigating the AI Safety Impact of Multi-Agent Scaffolds via Self-Improvement." arXiv preprint arXiv:2502.00757 (2025).
> [3] Sarkar, Bidipta, et al. "Evolution Strategies at the Hyperscale." arXiv preprint arXiv:2511.16652 (2025).

---

> ### Author Response · Authors · 2025-11-27
> **response to Reviewer eUZK (2/2)**
>
> ### The paper claims benefits from long-term memory for long-horizon tasks, but provides no qualitative retrieval to decision case study or horizon-sensitive ablation
>
> Thank you for your comment and for highlighting the importance of evaluating long-term memory in long-horizon settings. We agree with the reviewer on the importance of these experiments and would like to clarify that our paper does include horizon-sensitive evaluations specifically designed to test the contribution of the long-term memory system.
> In particular, Appendix F (Testing the Retrieval Mechanism – Long-Term Memory System) presents the exact case study. In the mentioned case study we test memory retrieval and action mechanism. The case study tasks the agent with a very complex task of creating an iron sword (multi-step objective that we noticed the agent always fails to perform across all crafter runs we inspected). At the beginning of the episode we add hand-crafted ‘memories’ of simpler tasks that lead to creating an iron sword. In the appendix we show which memories are activated at different time steps, showing the ability to retrieve memory and then act upon them. In the subsection F.2 in the Appendix we provide additional examples of the same tasks.
>
> ### For your questions:
>
> **Q1: How does performance scale with search budget? Report results for at least two larger budgets (e.g., ngen ∈ {4, 8, 16} and nchild ∈ {5, 10}) and state whether performance consistently improves, saturates, or degraesd.**
>
> Due to resource constraints we are unable to present results with the requested larger budgets. Nevertheless, as stated in our response above about the search budgets, we obtain large performance gains from the limited iterations. We also included additional figures (Figure 19 and 20 in the Appendix K.5)  of how the performance is not giving signs of saturation on the given budget.
>
>
> **Q2: Can you evaluate transfer across model families, applying the same evolved genome to at least one model from a different family (e.g., an OpenAI, Anthropic, or an open-source LLM). Report per-environment performance and compare to the original target model.**
>
> We believe that our response above addresses the question, we are happy to respond to any follow ups if the reviewer has additional questions.
>
> **Q3: Show per-method learning curves and final performance ± CI under a matched budget. If Genetic Algorithm still performs best, explain WHY (e.g., better parallelism, robustness) rather than relying on single-number superiority.**
>
> For reasons mentioned above, this question is out of the scope of this paper. We believe that it is an interesting direction for future work.
>
>
> **Q4: Please include one retrieval→decision trace and a simple ablation by episode length.**
>
> We refer here to our answer in the above section. In Appendix F, we evaluate the mechanism on the complex "iron sword" task. By injecting hand-crafted memories, the agent demonstrates a successful retrieval to decision trace, activating specific memories at the precise time steps required to solve the objective.
>
> Thank you again for your feedback. Let us know if you have any further questions.

---

### Author Response · Authors · 2025-11-27
**General Response**

We thank all the reviewers for their time and feedback. We have integrated it into an improved version of our work. A general overview of the changes follows now:

- We included **additional runs of our main experiments of Gemini** to evidence the statistical robustness of our approach.
- We included **new experiments with GPT models**, evidencing the strength of our approach with different families of models.
- We added **genome transferability experiments across different families of models**, finding that genomes transfer without additional optimisation between models of the same family, but we find that they don’t transfer well between different families. We observed this is caused due to the prompt nuanced biases that different models have.
- Included **experimental comparison with enhanced handcrafted prompts for non-TAME architectures**. Results further illustrate that manually adjusting prompts can enhance performance on some games but at the cost of the performance in other games, highlighting the need of adaptability that TAME provides.
- We included additional **figures to illustrate the effect of the evolutionary algorithm** on the agentic performance.
- **Additional related works** discussed, **addressed multiple clarification and detail concerns** raised by the reviewers, **renamed the different agentic version** to better illustrate the objective of our ablations.

Below we include the **results with the new models and transferability experiments**. Results with [TAME] indicate that the agentic architecture using that model was discovered using TAME with that model. [Transferred] is using architectures discovered with other models without performing evolutionary optimisation.  In the case of Gemini models the architecture transferred is from the best genome in Gemini-2.0-Flash and for GPT it is from GPT-4.1-nano.


| Group | Method | Score (↑) | BALROG Rank (↓) |
| :---: | :--- | :---: | :---: |
| 🔵 | **Gemini-2.5-Pro [Transferred]** | **47.57% ± 2.72%** | **(1) ↑ 1** |
| ⚪ | Grok-4 | 43.60% ± 2.20% | 1 |
| 🔵 | Gemini-2.5-Pro [Baseline] | 43.35% ± 2.3% | 2 |
| 🟢 | Gemini-2.0-Flash [TAME] | 35.05% ± 2.24% | (3) ↑ 9 |
| 🟢 | Gemini-2.0-Flash [Baseline] | 27.16% ± 2.12% | (12) |
| 🟡 | GPT-4.1-mini [Transferred] | 26.80% ± 1.92%| (12) |
| 🟡 | GPT-4.1-mini [Baseline] | 24.43% ± 1.89% | (12) |
| 🟠 | Gemini-2.5-Flash-Lite [Transferred] | 20.48% ± 0.91% | (14) ↑ 9 |
| 🔴 | GPT-4.1-nano [TAME] | 17.20% ± 1.47% | (18) ↑ 7 |
| 🟠 | Gemini-2.5-Flash-Lite [Baseline] | 11.87% ± 1.32% | (23) |
| 🔴 | GPT-4.1-nano [Baseline] | 9.91% ± 1.33% | (25) |


Full leaderboard available at: https://balrogai.com/index.html

Thus, we updated our contributions as  follows:
 (1) We introduce TAME, the first emergent game-agentic framework that enables LLMs to evolve modular structures tailored to gaming environments and achieve state-of-the-art performance in the BALROG benchmark; (2) We present a genetic approach key for the functioning of this framework, introducing a set of modules for general gaming capabilities and a genome that captures modules, hyperparameters and prompts enabling TAME to adapt the agent to the game with a low evolutionary budget (4 generations and 5 children); (3) We demonstrate that TAME can be used to find effective agentic configurations with smaller models that directly transfer and improve the performance of larger and reasoning models of the same family; (4) We propose a novel and effective long-term memory system that combines embedding-based retrieval with LLM-augmented semantic memory matching the performance of state-of-the-art memory methods requiring three times less LLM calls.

~~For the final version, we will be including additional runs from the GPT models to keep statistical consistency in our experiments. Sadly the experiments on Nethack take very long to run and we won’t have them until closer to the end of the rebuttal period. Nevertheless, from the results we are obtaining on all the other games we see this won’t require any changes on the text or the findings and claims of our work.~~
**UPDATE:** We finalised our experiments with all three runs from the GPT family and updated the Table above and in the revised version of our manuscript accordingly. We also updated results from GPT models in responses to reviewers.


We provide detailed individual responses to all your comments on the individual responses.

---

### Author Response · Authors · 2025-12-03
**Summary of rebuttal period**

We thank all the reviewers for their time and feedback. We have now integrated the feedback into an improved version of our work.

To conclude, we would like to summarize the key contributions and major changes made to our manuscript following the review process.

We present the key strengths and weaknesses raised by reviewers and our rebuttal process in the table below. We mark comments resolved by extra experiments or material in green(🟢), misunderstandings (e.g. material already included) in orange (🟠), otherwise addressed concerns or questions in purple(🟣), and out-of-scope requests in blue(🔵).

| Reviewer (rating / confidence) | eUZK (4/4) | NH6Y (6/4) | s3bn (2→6, 4→?) | mETx (4/4) | dxNh (6/3) |
| :--- | :--- | :--- | :--- | :--- | :--- |
| **Strengths** | Practically relevant | Innovative approach | Well-organized and easy-to-understand | Novel and reasonable method | Interesting use of genetic algorithm |
| | Engineering is reasonably complete | Thorough analysis of experiments | Rich implementation details | Variety of ablations and hyperparameters | Significant improvement |
| | Targeted ablations | Clearly articulated design | Rigorous ablation and analysis | Reproducibility of the experiments is high | |
| | | Significant contribution | | Possibly high impact | |
| **Weaknesses** | 🟢 Small budget | 🟢 Limited conceptual novelty | 🟢 Limited novelty | 🟢 How much of improvement is due to genetic approach? | 🟢 Writing is hard to follow |
| | 🟢 One family of models | 🟣 Dependence on hand-crafted modules and uncertain generalisation | 🟣 Poor scope/generalizability | 🟢 Clarity | 🟠 Lacking details of the genetic algorithm |
| | 🔵 Missing budget-matched baselines | 🟣 Metrics focus on progression scores | 🟢 Insufficient discussion of related work | | 🟣 Resembles "evolutionary hyperparameter optimization" |
| | 🟠 No memory case study | | 🟢 Limited experiments | | 🟣 Resource-heavy and difficult to apply in production |
| **Questions** ("-" if similar to weaknesses) | - | 🟣 Measure of emergence | 🔵 Other benchmarks | | 🟢 More discussion on relevant work |
| | | 🟣 Approximate computing budget | | | 🟢 Improve clarity of figures |
| | | 🟣 Behavioural analyses of module combinations | | | |
| | | 🟣 Why genetic algorithm? | | | |
| | | 🟣 Will authors release code? | | | |

**We summarise the key changes made during the rebuttal period.**
-  **Additional runs of our main experiment with Gemini** to evidence the statistical robustness of our approach.
-  **New experiments with GPT models with different families of models,** evidencing the strength of our approach.
-  **Additional transferability experiments across different families of models**, finding that genomes transfer without additional optimisation between models of the same family, but we find that they don’t transfer well between different families. We observed this is caused due to the prompt nuanced biases that different models have.
- **Additional experimental comparison with enhanced handcrafted prompts for non-TAME architectures**. Results further illustrate that manually adjusting prompts can enhance performance on some games but at the cost of the performance in other games, highlighting the need for adaptability that TAME provides.
- **Additional figures to illustrate the effect of the evolutionary algorithm on the agentic performance**.
- **Additional related works and discussions** as requested by the reviewers.
-  **Improved clarity & presentation** as we addressed multiple clarification and detail concerns raised by the reviewers and renamed the different agentic versions to better illustrate the objective of our ablations.

We believe to have addressed most of the concerns to the best of our abilities. Following our response:
- Reviewer **s3bn** agreed to increase their score from 2 to 6. Unfortunately, due to the cancellation of the rebuttal, they were not able to edit their score.
- The other reviewers were unable to respond to our clarifications in time. However, we note that Reviewer **mETx** indicated in their initial review that they would be happy to raise their score if their questions were adequately addressed - we believe we have comprehensively responded to all their concerns.

In light of the cancelled rebuttal phase, we kindly request the Area Chair to carefully consider the detailed responses and key changes summarized above before taking a final decision on our manuscript.

---

### Meta-Review · Area_Chair_kfBp · 2025-12-29

**Summary:**

The reviews ranged between weak accept and weak reject, highlighting limited novelty, lack of rigour, poor writing, and limited scope amongst other criticism. While some concerns were addressed during the rebuttal, the paper is still not ready for publication.

**Reviewer Concerns:**

no concerns

**Reviewer Scores:**

not relevant

---

### Decision · Program_Chairs · 2026-01-26

Reject